# Distinct pathways drive anterior hypoblast specification in the implanting human embryo

Bailey A. T. Weatherbee [1,7,10], Antonia Weberling [1,8,9,10], Carlos W. Gantner [1,10], Lisa K. Iwamoto-Stohl[1], Zoe Barnikel[2], Amy Barrie[2], Alison Campbell[2], Paula Cunningham[2], Cath Drezet[2], Panagiota Efstathiou[2], Simon Fishel[2], Sandra Gutiérrez Vindel[2], Megan Lockwood[2], Rebecca Oakley[2], Catherine Pretty[2], Nabiha Chowdhury[3], Lucy Richardson[3], Anastasia Mania[4], Lauren Weavers[4], Leila Christie[5], Kay Elder [5], Phillip Snell[5] & Magdalena Zernicka-Goetz [1,6] ✉

Development requires coordinated interactions between the epiblast, which generates the embryo proper; the trophectoderm, which generates the placenta; and the hypoblast, which forms both the anterior signalling centre and the yolk sac. These interactions remain poorly understood in human embryogenesis because mechanistic studies have only recently become possible. Here we examine signalling interactions post-implantation using human embryos and stem cell models of the epiblast and hypoblast. We find anterior hypoblast specification is NODAL dependent, as in the mouse. However, while BMP inhibits anterior signalling centre specification in the mouse, it is essential for its maintenance in human. We also find contrasting requirements for BMP in the naive pre-implantation epiblast of mouse and human embryos. Finally, we show that NOTCH signalling is important for human epiblast survival. Our findings of conserved and species-specific factors that drive these early stages of embryonic development highlight the strengths of comparative species studies.

Many human pregnancies fail during implantation[1,2], yet this stage is difficult to study in vivo. Successful development beyond implantation stages requires the specification of the three major lineages: the epiblast, which gives rise to the embryo proper, and two extra-embryonic tissues: the trophectoderm, that generates the placenta, and the hypoblast, which forms the yolk sac. Interactions between extra-embryonic and epiblast tissues are essential to embryo implantation, survival and patterning, and coordinate to set the stage for gastrulation[3,4].

Although some of the mechanisms underlying early embryo development differ between human and mouse[3–5], several events, including the formation of the pro-amniotic cavity within the polarized epiblast, concomitant exit from naive pluripotency[6–8], and specification of a

[1]Mammalian Embryo and Stem Cell Group, Department of Physiology, Development and Neuroscience, Mammalian Embryo and Stem Cell Group, University of Cambridge, Cambridge, UK. [2]CARE Fertility, Nottingham, UK. [3]Herts & Essex Fertility Centre, Bishops College, Cheshunt, UK. [4]King's Fertility, Denmark Hill, London, UK. [5]Bourn Hall Fertility Clinic, Bourn, UK. [6]Stem Cells Self-Organization Group, Division of Biology and Biological Engineering, California Institute of Technology, Pasadena, CA, USA. [7]Present address: Center for Stem Cell and Organoid Medicine, Perinatal Institute, Division of Developmental Biology, Cincinnati Children's Hospital Medical Center, Cincinnati, OH, USA. [8]Present address: All Souls College, Oxford, UK. [9]Present address: Nuffield Department of Women's and Reproductive Health, Women's Centre, John Radcliffe Hospital, University of Oxford, Oxford, UK. [10]These authors contributed equally: Bailey A. T. Weatherbee, Antonia Weberling, Carlos W. Gantner. ✉e-mail: mz205@cam.ac.uk

subset of hypoblast (primitive endoderm in mouse)-derived visceral endoderm cells into the anterior visceral endoderm (AVE), or anterior hypoblast, are conserved across mammalian species. The anterior hypoblast signalling centre secretes antagonists of the BMP, WNT and NODAL pathways, which protect the adjacent epiblast from primitive streak formation[9-14]. In the mouse, the AVE first appears at the distal tip of the egg cylinder in response to synergistic NODAL induction between the visceral endoderm and epiblast[15-17]. Trophectoderm-derived extra-embryonic ectoderm-secreted Bmp4 couples morphogen gradients to morphogenesis by repressing AVE specification until the embryo reaches the appropriate size[16,18,19]. The presence of the analogous anterior hypoblast signalling centre has also been observed in non-human primate[20,21] and in vitro cultured human embryos[10]. However, the role of morphogens in driving anterior hypoblast specification during human development and their conservation with the mouse is unknown.

In this Article, we sought to determine the signalling dynamics underpinning the pre-to-post implantation transition, focusing specifically on anterior hypoblast specification in human embryos. Functional testing in human embryos, as well as stem cell models of the hypoblast and epiblast, revealed crucial roles for NODAL, BMP and NOTCH during human peri-implantation development.

## Results

### Signalling dynamics during human embryo implantation

Recent advancements adapting culture conditions originally developed for the mouse to the in vitro culture of human embryos beyond implantation have allowed us and others to investigate this critical window of development for the first time[22-25]. We, and other groups, have utilized these approaches to generate large single-cell RNA sequencing (scRNA-seq) datasets of early primate development[10,26-29].

Here we used existing scRNA-seq datasets to guide a cross-species study of the conserved and divergent signalling dynamics underpinning anterior hypoblast specification. To identify candidate signalling pathways, we first integrated existing scRNA-seq datasets spanning 5–14 days post-fertilization (days 5–14) in the human embryo, as well as equivalent stages in cynomolgus macaque (days 6–17) and mouse embryos (embryonic day (E)3.5–E7.5) (Fig. 1a–c and Extended Data Figs. 1a–c and 2a–c).

In our integrated, species-specific datasets, we could identify key clusters on the basis of conserved expression patterns, including POU5F1, SOX2 and NANOG in the epiblast clusters, GATA6 and SOX17 in the hypoblast/primitive and visceral endoderm clusters and GATA3, GATA2 and TFAP2C in the trophectoderm/trophoblast/extra-embryonic ectoderm clusters (Fig. 1d,e, Extended Data Figs. 1d and 2d, and Supplementary Tables 1–6). Two terminally differentiated CGA-positive trophoblast (a post-implantation derivative of the trophectoderm) lineages could also be detected in the human: SDC1-positive syncytiotrophoblast and CGB1-positive extravillous trophoblast[10,30-33]. In the macaque, a population of GATA6, COL6A1, VIM, POSTN, LUM-positive extra-embryonic mesenchyme was also detected, which was not observed in in vitro cultured human embryos[21,34,35] (Extended Data Fig. 1d).

Next, we leveraged our integrated dataset to identify signalling dynamics during human in vitro development, focusing on stages when the anterior hypoblast is specified. Module scoring (using WikiPathways gene lists, Supplementary Table 7) for genes associated with key signalling pathways identified enrichment within and between different cell lineages (Fig. 1f and Extended Data Fig. 3a,b). Module scoring was also carried out in mouse and macaque datasets (Extended Data Figs. 1e and 2e).

Our module scoring closely aligned with patterns previously described in mammalian embryos, including enrichment of NODAL in the epiblast and hypoblast[15,36,37], WNT pathway enrichment in the epiblast before gastrulation[38,39], upregulation of FGFR (FGFR1, FGFR3

and FGFR4) modules across lineages at implantation[10] and Hedgehog signalling module enrichment in the syncytiotrophoblast[40] (Fig. 1f). The NODAL module was noticeably enriched in the blastocyst-stage epiblast in human and peri-implantation-stage epiblast in macaque, in contrast to mouse, where enrichment in genes associated with NODAL signalling occurred upon implantation (Fig. 1f and Extended Data Figs. 1e and 2e). Likewise, the mouse extra-embryonic ectoderm was enriched for the BMP module following implantation, but this pattern was predominantly absent in macaque and human trophoblast, where the BMP module is enriched in the hypoblast and extra-embryonic mesenchyme instead (Fig. 1f and Extended Data Figs. 1e, 2e and 3a,b). Overall, this analysis provided a global insight into the gene expression patterns associated with specific signalling pathways during implantation of the human embryo.

To dissect potential crosstalk between lineages of the peri-implantation human embryo, we employed the computational tool CellPhoneDB, which utilizes a curated heteromeric ligand–receptor repository to provide a tissue-to-tissue interaction map[41] (Fig. 1g,h and Extended Data Fig. 3c,d). We found that epiblast-secreted NODAL was predicted to be received by all lineages of the blastocyst. In contrast, the predicted NODAL receptivity was enriched in the hypoblast upon implantation (Fig. 1g,h). In agreement with our global analysis, BMP signalling interactions were predicted to be present in the hypoblast, which expressed both BMP ligands and receptors. Notably, upon implantation, predicted epiblast-to-epiblast NOTCH signalling was specifically enriched.

Using our combined transcriptomic analysis as a guide, we next sought to identify signalling pathways important for human anterior hypoblast signalling centre specification. We selected candidate pathways on the basis of three criteria: (1) lineage-specific expression; (2) temporal expression changes during implantation; and (3) predicted role in anterior hypoblast formation and/or epiblast maturation based on other model organisms. This analysis led us to select two TGFb superfamily members: NODAL and BMP; and the NOTCH pathway.

### NODAL signalling is essential for the anterior hypoblast

To quantify inferred NODAL activity, we examined the NODAL effector SMAD2.3. by calculating the nuclear, cytoplasmic and nuclear-to-cytoplasmic ratio (n/c) of total SMAD2.3 in individual cells (Extended Data Fig. 4a–c)[42,43]. To accurately capture early blastocyst development, we further classified blastocysts at day 5 or 6 as 'segregating' (inner cell mass co-expressing epiblast and hypoblast markers) or 'segregated' (inner cell mass with mutually exclusive epiblast and hypoblast protein expression) (Extended Data Fig. 4d,e)[37,44,45].

We found that the trophectoderm, epiblast and hypoblast all expressed NODAL receptors at the blastocyst stage in human (Extended Data Fig. 4f–h). In segregating blastocysts, the n/c SMAD2.3 was significantly higher in trophectoderm cells than inner cell mass cells, though raw nuclear and cytoplasmic signals were lower, similar to the E3.5 early mouse blastocyst (Fig. 2a,b and Extended Data Fig. 5a–c). Concomitant with lineage segregation, GATA6-positive hypoblast cells exhibited increased n/c SMAD2.3 compared with epiblast cells (Fig. 2c). The hypoblast (human) and visceral endoderm (mouse) maintained these higher levels of n/c SMAD2.3 compared with the epiblast at implantation stages (Fig. 2d–f and Extended Data Fig. 5a,d–f), consistent with our module scoring and CellPhoneDB predictions from the scRNA-seq datasets. The n/c SMAD2.3 decreased over time in both human and mouse epiblasts (Fig. 2g and Extended Data Fig. 5a,g). However, levels within the inner cell mass in human embryos appeared higher and subsequently showed a larger decrease following lineage segregation. In contrast, high n/c SMAD2.3 was maintained in the human hypoblast upon implantation but increased in the visceral endoderm upon implantation in mouse (Fig. 2h and Extended Data Fig. 5a,h). The mouse extra-embryonic ectoderm showed a decrease in n/c Smad2.3 upon implantation (Extended Data Fig. 5i). Strikingly, on

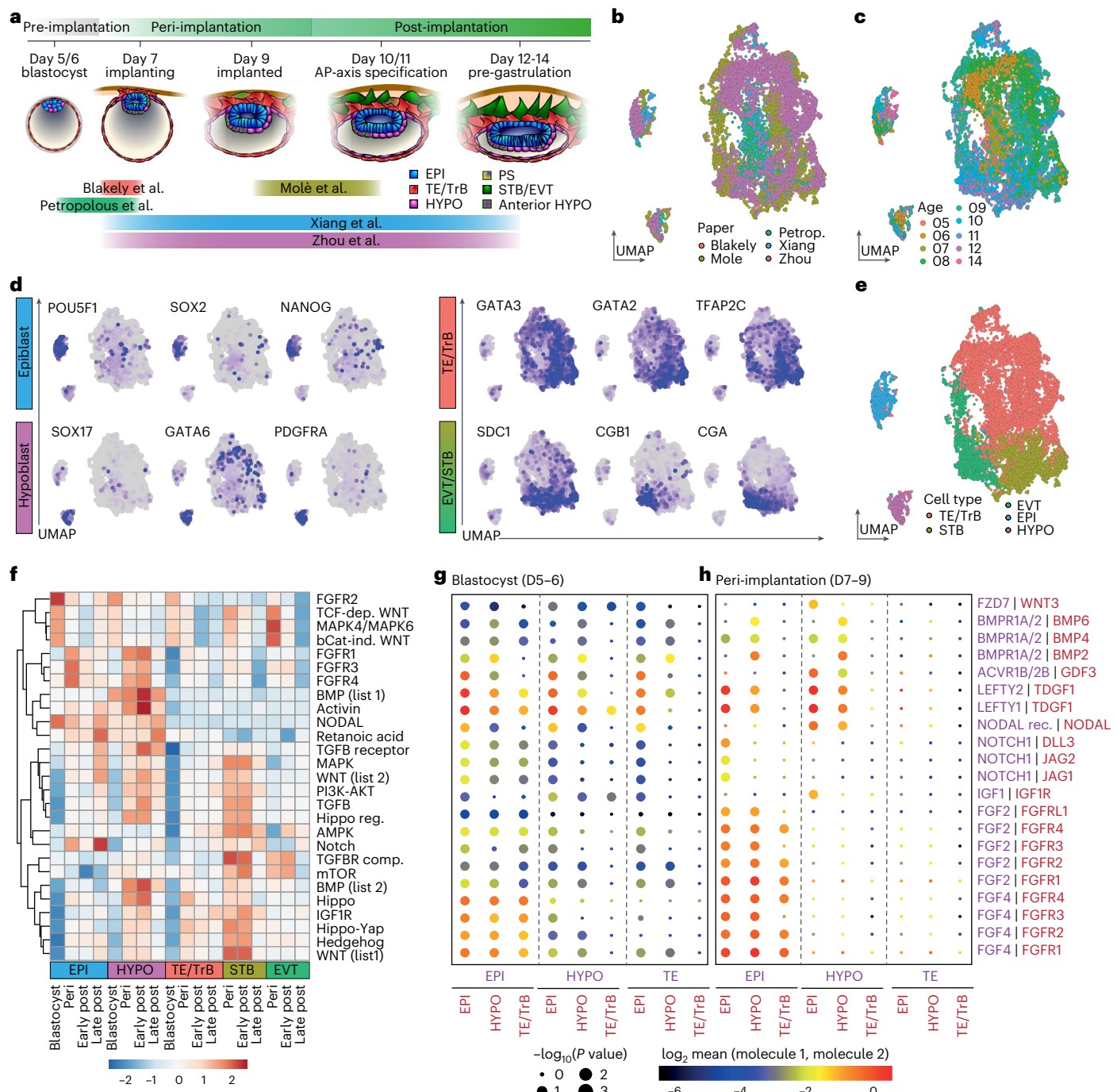

**Fig. 1 | Generation and analysis of a combined human sequencing dataset.**
**a**, Diagram depicting human development from blastocyst through pre-gastrulation stages. The datasets integrated to generate our combined object are presented below. **b**, Uniform manifold approximation projection (UMAP) of datasets coloured by original publication. *N* = 9,862 single cells. **c**, UMAP of human datasets coloured by embryo age. **d**, UMAPs of human datasets coloured by a gradient of canonical marker gene expression for epiblast (EPI), hypoblast (HYPO), trophectoderm/trophoblast (TE/TrB) and terminal extravillous trophoblast (EVT) or syncytiotrophoblast (STB) lineages. **e**, UMAP coloured by assigned cell type. **f**, Heatmap of WikiPathways signalling pathway module score for each developmental stage of each individual cell type. Visualized value is calculated average scaled score from individual single-cell module scores. **g**,**h**, CellPhoneDB dotplots for blastocyst (**g**) and peri-implantation (**h**) stages depicting predicted activity of individual receptor–ligand pairs. Each ligand–receptor pair is colour matched to the expressing lineage on the *x* axis. *N* = 9,862 single cells. AP, anterior-posterior; dep, dependent; ind, independent; reg, regulation; comp, complex.

day 9 in the human embryo, n/c SMAD2.3 was higher in hypoblast cells in direct contact with the overlying epiblast (Fig. 2i,j), supporting predictions by CellPhoneDB that the epiblast is the primary NODAL source.

To investigate the function of NODAL signalling across peri-implantation stages, we used Activin-A or the ALK4/5/7 inhibitor A83-01 to modulate activation of this pathway at two stages of human

embryogenesis: in the human blastocyst from day 5 to day 7, when the inner cell mass segregates and the anterior hypoblast marker CER1 is first expressed; and from day 7 to day 9, which encompasses in vitro implantation, exit from naive pluripotency and regionalization of the hypoblast for anterior–posterior axis specification[10]. As amnion and extra-embryonic mesenchyme do not form robustly during in vitro

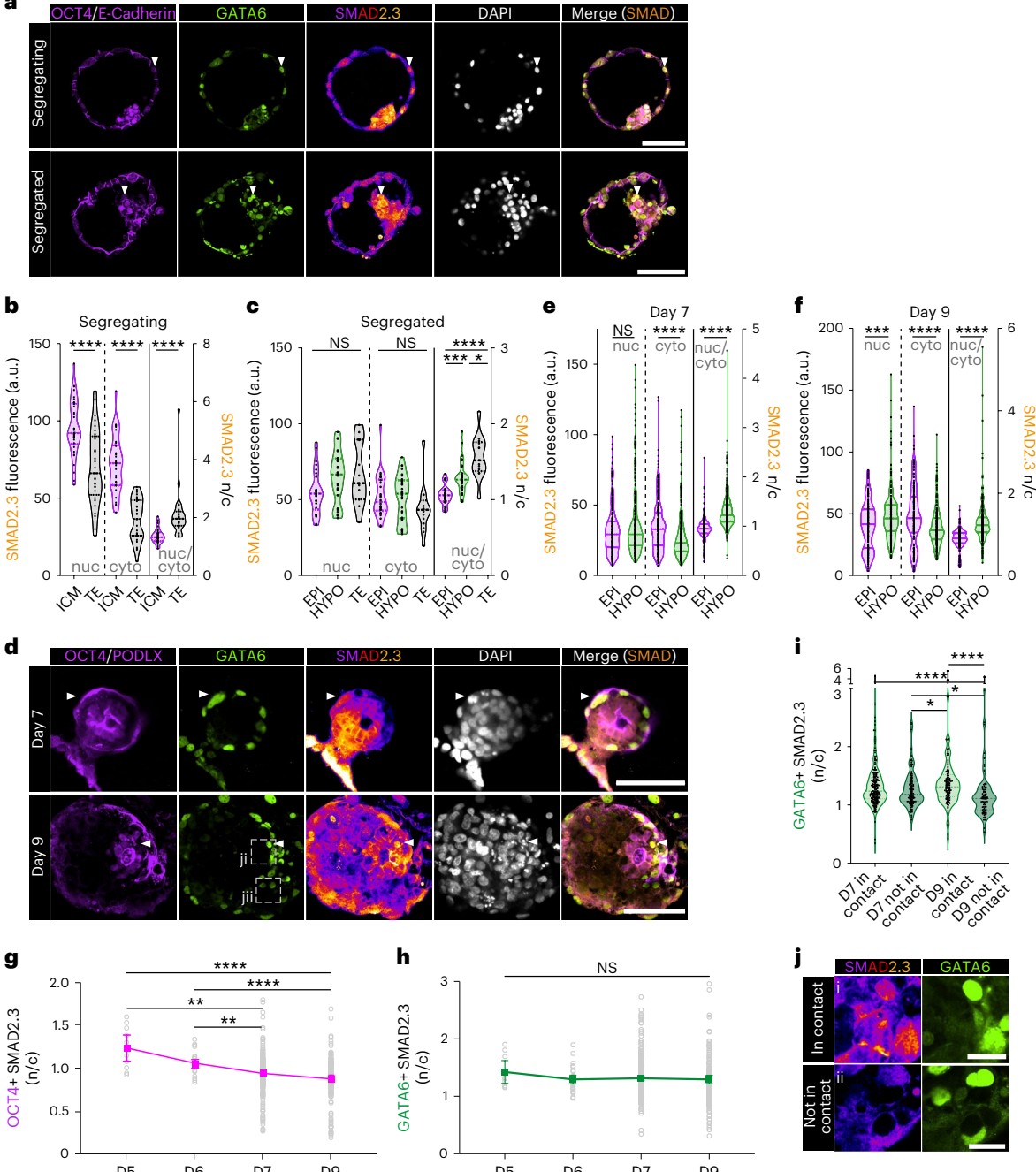

**Fig. 2 | Characterization of SMAD2.3 dynamics during human in vitro implantation. a**, Immunofluorescence images of segregating (*N* = 3 embryos) and segregated human blastocysts (*N* = 3 blastocysts) stained for OCT4/E-CADHERIN, GATA6, total SMAD2.3 and DAPI. **b**,**c**, Quantification of nuclear and cytoplasmic SMAD2.3 fluorescence and their ratio in inner cell mass (ICM, *n* = 34 cells) and trophectoderm (TE, *n* = 31 cells) in segregating blastocysts and between epiblast (EPI, *n* = 26 cells), hypoblast (HYPO, *n* = 31 cells) and TE (n = 13 cells) of segregated blastocysts. **d**, Immunofluorescence images of day 7 (*N* = 17 embryos) and day 9 (*N* = 13 embryos) human embryos following in vitro implantation stained for OCT4/PODLX, GATA6, SMAD2.3 and DAPI. **e**,**f**, Quantification of nuclear and cytoplasmic SMAD2.3 fluorescence and their ratio in EPI and HYPO at day 7 (EPI *n* = 299 cells, HYPO *n* = 322 cells) and day 9 (EPI *n* = 297 cells, HYPO *n* = 214 cells). **g**,**h**, Plot of n/c SMAD2.3 over time in EPI (**g**; *n* = 11

day 5, 26 day 6, 299 day 7 and 197 day 9 cells) and HYPO (**h**; *n* = 10 day 5, 31 day 6, 322 day 7 and 214 day 9 cells). **i**, Quantification of n/c SMAD2.3 in HYPO cells at day 7 and day 9 depending on contact with overlying EPI. **j**, High-power images of SMAD2.3 in day 9 HYPO cells either contacting or not contacting the EPI. Regions shown here are outlined with a dashed box in **d**. For violin plots, central dotted line denotes median and dotted lines mark the 25th and 75th quartiles. Error bars denote standard error (**g** and **h**). Statistical tests: two-sided unpaired *t*-test (**b** (nuclear (nuc) and cytoplasmic (cyto))), two-sided Mann–Whitney (**b** (n/c), **e** and **f**), one-way analysis of variance with Tukey–Kramer post-hoc (**c** (nuc)), Kruskal–Wallis with Dunn's post-hoc (**c** (cyto and n/c) and **g**–**i**); ****P* < 0.0001, ***P* < 0.001, ***P* < 0.01, **P* < 0.05. Unmarked pairwise comparisons are not significant (NS). Exact *P* values presented in Supplementary Table 8. Scale bars: 25 μm (**a**), 100 μm (**b** and **e**) and 20 μm (**k**).

culture of human embryos[10,24–27,46], we limited our experimentation to time frames preceding their expected differentiation in vivo to mitigate signalling defects associated with the lack of these tissues.

We validated Activin-A-mediated activation and A83-01-mediated inhibition of NODAL signalling in human embryonic stem (ES) cells (Extended Data Fig. 6a,b).

Activin-A or A83-01 treatment did not affect total cell numbers of OCT4-positive epiblast or GATA6-positive hypoblast (Fig. 3a–j and Extended Data Fig. 6c,d). In contrast, treating human embryos from days 5–7 or days 7–9 with A83-01 decreased the number of CER1-positive hypoblast cells (Fig. 3f,j and Extended Data Fig. 6c,d). NODAL is essential for the formation of the AVE at equivalent peri-implantation stages in the mouse[15,16,47]. In stage-matched in vitro cultured mouse embryos, A83-01 treatment between E3.5 + 48 h did not affect epiblast or hypoblast cell numbers (Extended Data Fig. 6e–g). A83-01 treatment between E5.0 + 36 h decreased epiblast size and AVE formation, as observed previously[16] (Extended Data Fig. 6h–k).

To further examine the role of NODAL in the anterior hypoblast, we generated yolk sac-like cells (YSLCs) from human ES cells, which express CER1 and resemble extra-embryonic endoderm[48] (Fig. 3k). Treatment with A83-01 decreased both GATA6 and CER1 expression (Fig. 3l,m), as expected given the use of exogenous Activin-A to differentiate YSLCs. We have previously shown that YSLCs secrete signalling antagonists and that YSLC conditioned medium protects three-dimensional (3D) epiblast-like human ES cell spheroids from BMP-induced differentiation[48]. Conditioned medium generated from YSLCs treated with A83-01 exhibited impaired capacity to protect human ES cell spheroids, demonstrated by increased pSMAD1.5.9 expression (Fig. 3n,o). These data demonstrate that specification and maintenance of the anterior hypoblast are NODAL dependent.

A83-01 is commonly used to differentiate naïve human ES cells towards trophectoderm[49–51]. Therefore, we also assessed the response of the trophectoderm to NODAL perturbation from day 5 to day 7 post-fertilization. A83-01 treatment did not increase trophectoderm cell number. However, Activin-A treatment decreased the number of GATA3-positive trophectoderm cells (Extended Data Fig. 6l–p). Further analysis demonstrated that this reduction occurs specifically in the GATA3/GATA6-double positive population, which marks pre-implantation trophectoderm surrounding the blastocoel cavity[52] (Extended Data Fig. 6o,p). Further, Activin-A treated embryos at day 7 showed higher rates of attachment and implantation-like morphology compared with A83-01 treated embryos (Extended Data Fig. 6q). The transition towards predominantly GATA3-positive/GATA6-negative trophoblast cells was reminiscent of the transition between day 7 and day 9 in control conditions (Extended Data Fig. 6r,s). Thus, our results show that NODAL signalling positively regulates trophectoderm maturation, in agreement with recent reports that epiblast-secreted signals drive local maturation of the trophectoderm[51].

## BMP signalling is active in the human naive epiblast

Next, we characterized the activity of BMP signalling by examining the expression of its nuclear effector pSMAD1.5.9 (refs. 43,53) (Extended Data Fig. 7a–c). In segregating blastocysts, the trophectoderm was enriched for pSMAD1.5.9 (Fig. 4a,b and Extended Data Fig. 7d) despite low predicted activity/module scoring of BMP. However, both BMPR1A and BMPR2 are expressed in pre-implantation trophectoderm and downregulated at implantation (Extended Data Fig. 3b). In contrast, there was no difference in pSMAD1.5.9 intensity between inner cell mass and trophectoderm lineages of stage-matched mouse embryos (Extended Data Fig. 7e,f). In segregated blastocysts, pSMAD1.5.9 was enriched in the hypoblast, and expression in the trophectoderm was maintained (Fig. 4c and Extended Data Fig. 7d). Comparatively, despite a global increase in pSMAD1.5.9 levels, we observed no lineage-specific enrichment in E4.5 late mouse blastocysts (Extended Data Fig. 7g). pSMAD1.5.9 was higher in the hypoblast than epiblast at day 7 in human embryos (Fig. 4d,e and Extended Data Fig. 7d), consistent with the module scoring, but levels were similar in the hypoblast and epiblast in human embryos at day 9 (Fig. 4f,g and Extended Data Fig. 7d). In the mouse, following implantation, only the visceral endoderm sustained high levels of BMP activity (Extended Data Fig. 7j–l).

pSMAD1.5.9 expression decreased over time in the human epiblast. Similarly, the human hypoblast exhibited a decrease in pSMAD1.5.9 upon implantation (Fig. 4h,i). Interestingly, at day 9, pSMAD1.5.9 expression was higher in hypoblast cells in contact with the epiblast than in those distal to the epiblast, despite no observable increase in epiblast expression of BMP ligands at this stage, implicating an additional regulator of pSMAD1.5.9 in the hypoblast (Fig. 4j).

To interrogate the role of BMP signalling in the embryo, we first examined the expression of BMP ligands. BMP2/BMP4/BMP6 expression was enriched in the human hypoblast (Extended Data Fig. 8a and Supplementary Table 4). In macaque, BMP2/BMP4 expression could also be observed in the extra-embryonic mesenchyme, while in mouse Bmp4/Bmp8b were enriched in the extra-embryonic ectoderm and Bmp2 was expressed in the visceral endoderm (Supplementary Tables 5 and 6). Addition of BMP2, BMP4 or BMP6 efficiently increased phosphorylation of SMAD1.5.9 in human ES cells (Extended Data Fig. 8b,c). We utilized BMP6 for subsequent experiments because it (1) showed specific upregulation upon implantation, (2) did not induce TBXT/BRACHYURY-positive primitive streak-like cell differentiation in vitro, and (3) was hypoblast specific in macaque. Additionally, we treated human embryos with the BMP receptor inhibitor LDN193189 (LDN; see Methods for details) to antagonize BMP signalling.

BMP inhibition with LDN from day 5 to day 7 decreased the number of epiblast cells (Fig. 5a–d and Extended Data Fig. 8d). In contrast, LDN treatment from day 7 to day 9 did not affect epiblast cell numbers, but conversely, the addition of BMP6 resulted in a reduction (Fig. 5e–h and Extended Data Fig. 8e). In contrast, mouse epiblast size decreased with LDN treatment in post- but not pre-implantation, as reported previously[54] (Extended Data Fig. 8f–l). Together, these results suggest

---

**Fig. 3 | NODAL signalling is essential for anterior hypoblast specification and maintenance. a**, Schematic of experimental design for human embryo treatment from day 5 to day 7 to modulate NODAL signalling. **b**, Immunofluorescence images of embryos cultured from day 5 to day 7 in control (N = 44 embryos), 2 μM A83-01 (N = 12 embryos) or 25 ng ml⁻¹ Activin-A (N = 14 embryos) conditions. **c–e**, Quantification of the number of epiblast (**c**), hypoblast (**d**) and CER1-positive hypoblast cells (**e**) at day 7. **f**, Schematic of experimental design for human embryo treatment antagonists from day 7 to day 9 to modulate NODAL signalling. **g**, Immunofluorescence images of embryos cultured from day 7 to day 9 in control (N = 39 embryos), 2 μM A83-01 (N = 12 embryos) or 25 ng ml⁻¹ Activin-A (N = 15 embryos) conditions. **h–j**, Quantification of the number of epiblast (**h**), hypoblast (**i**) and CER1-positive hypoblast cells (**j**) at day 9. **k**, Schematic and immunofluorescence images of YSLC differentiation ± 2 μM A83-01 for 48 h during YSLC specification or maturation. **l**, Quantification of GATA6 fluorescence levels relative to DAPI and normalized to control. D2–D4: control n = 4,085; A83-01 n = 4,255. D4–D6: control n = 4,139; A83-01 n = 4,180. N = 2 experiments. **m**, Percentage of GATA6-positive cells that are CER1 positive. D2–D4: control n = 2,502; A83-01 n = 2,347. D4–D6: control n = 2,500; A83-01 n = 2,013. N = 2 experiments. **n**, Schematic and immunofluorescence images of pSMAD1.5.9 expression in human (h)ES cell-derived spheroids cultured in mTeSR+ medium (mTeSR+ n = 1,060 cells) or mTeSR+ medium conditioned on either control YSLC (YSLC CM n = 1,836 cells) or on YSLC differentiated with 48 h of A83-01 treatment (YSLC + A83 CM n = 1,654 cells). N = 3 experiments. **o**, Quantification of normalized pSMAD1.5.9 fluorescence from **n**. For box plots, box encompasses 25th and 75th quartiles with median marked by central line. Minimum and maximum and denoted by whiskers. Plus symbol marks the mean. Error bars denote standard error (**m–o**). Statistical tests: two-sided Mann–Whitney test (**c–e** and **h–j**); two-sided unpaired t-test (**l** and **m**); one-way analysis of variance with Tukey–Kramer post-hoc (**o**). ****P < 0.0001, **P < 0.01, *P < 0.05. Unmarked pairwise comparisons are not significant (NS). Exact P values presented in Supplementary Table 8. Scale bars: 100 μm (**b** and **g**) and 50 μm (**k** and **n**). Embryos in **b** and **g** were re-stained for GATA6, and re-orientated second-stain images are placed on a grey background for transparency and to match the orientation of the merged image.

a switch in the requirement of BMP during pre- and post-implantation as well as a divergence between the mouse and human patterns.

To validate the role of BMP in the human epiblast, we utilized human and mouse ES cells across the pluripotency spectrum (that is naive or primed). We found that LDN treatment increased apoptosis in naive human ES cells but conversely decreased apoptosis in primed human ES cells (Extended Data Fig. 8m). These data suggest that pluripotency states may underscore the difference in pre- and post-implantation BMP dependence in the embryo. Examination of the correlation of BMP response genes *ID1–3* (ref. 55) with naive

pluripotency markers in the pre-implantation epiblast showed positive co-expression in human, but not mouse (Fig. 5i and Extended Data Fig. 8n). Similarly, pSMAD1.5.9 expression was enriched in naive compared with primed human ES cells, but this pattern was reversed in mouse ES cells (Fig. 5j,k). This difference in BMP activity appears functionally important, as LDN treatment decreased the human-specific naive marker AP2γ in human ES cells, while in mouse ES cells it decreased the mouse-specific primed marker Otx2 (Extended Data Fig. 8o,p). Overall, these data support a role for BMP in human pre-implantation epiblast and suggest that stage-dependent

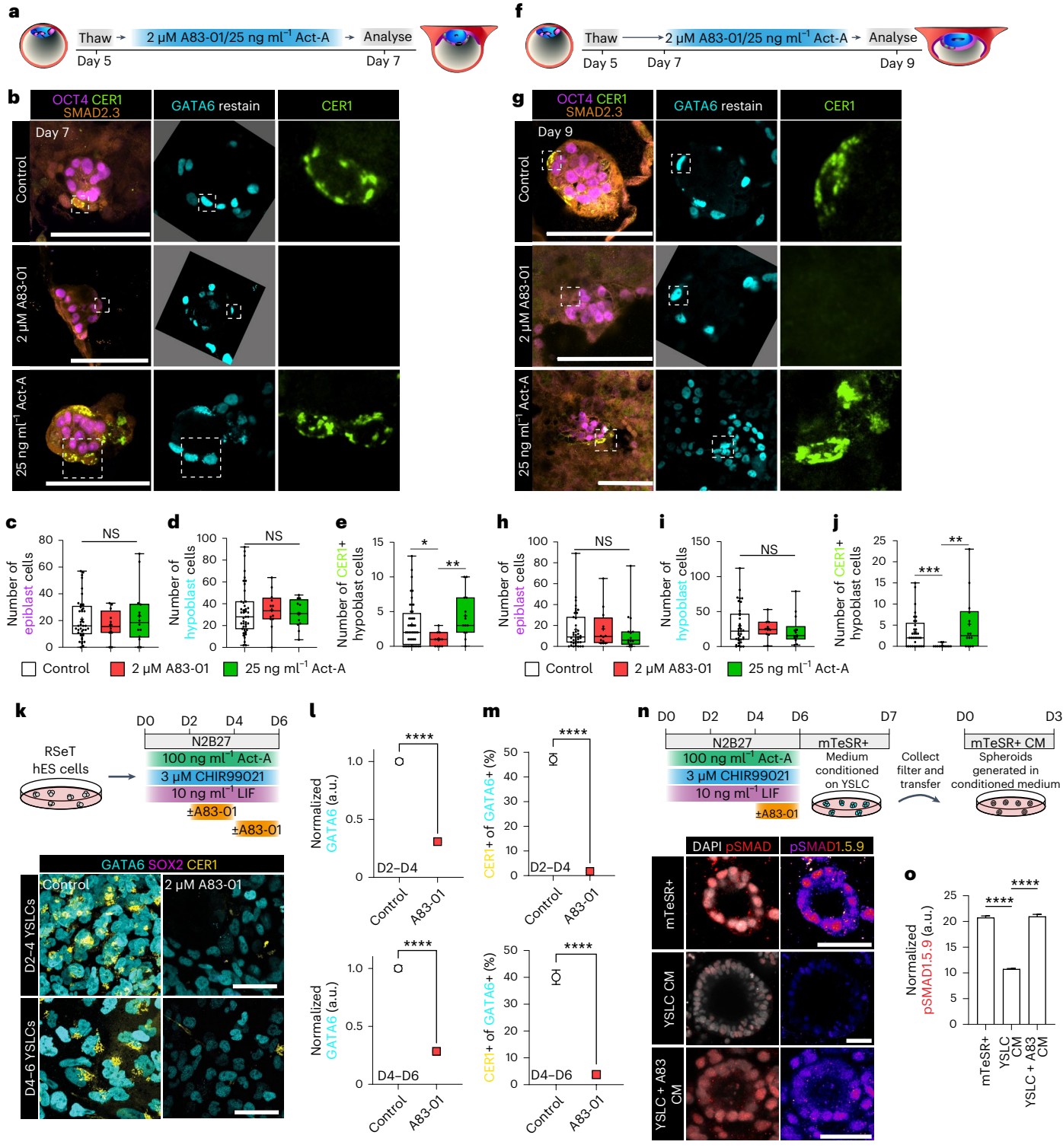

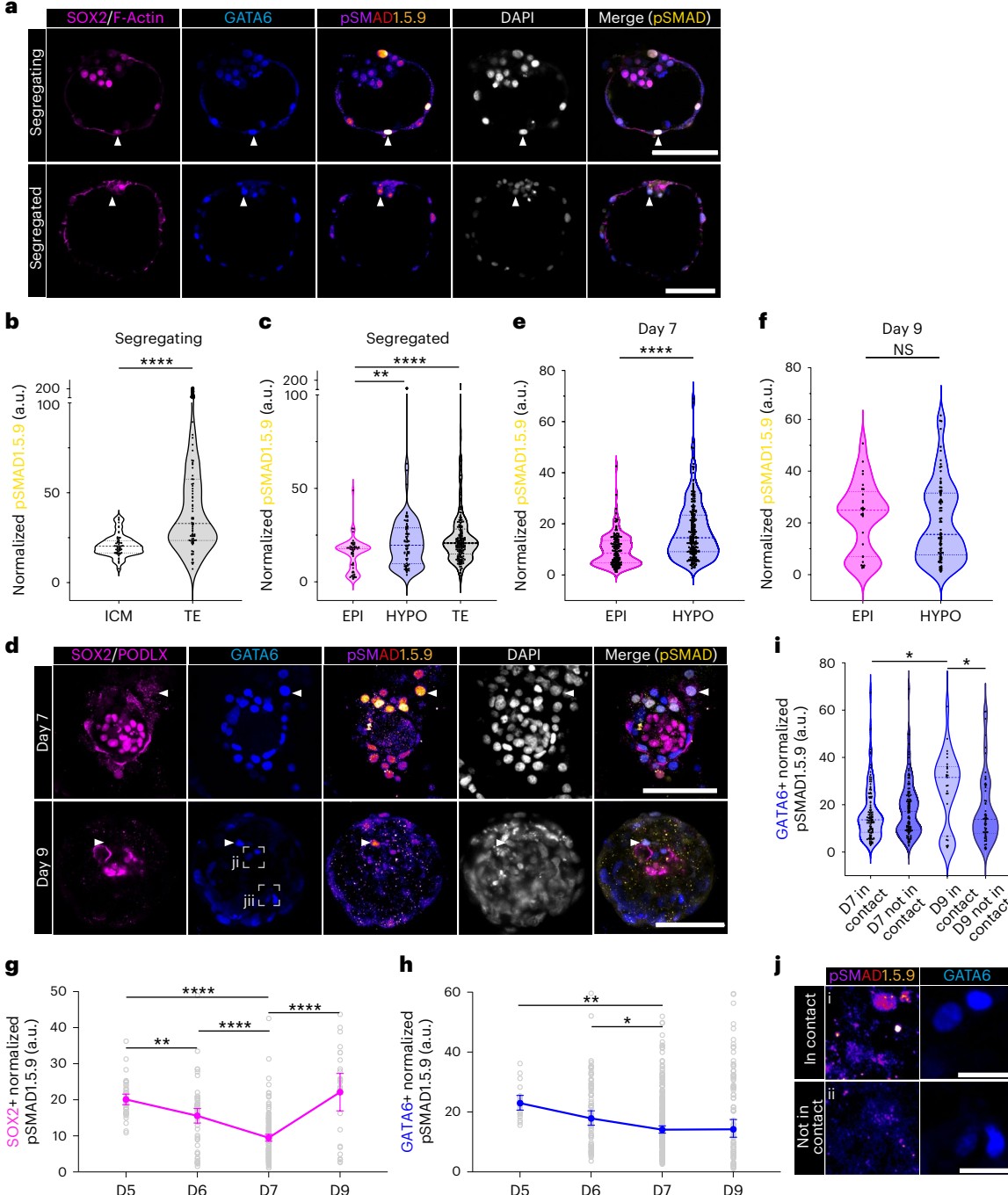

**Fig. 4 | Characterization of SMAD1.5.9 dynamics during human in vitro implantation. a**, Immunofluorescence images of segregating (*N* = 4 blastocysts) and segregated human blastocysts (*N* = 9 blastocysts) stained for SOX2/F-Actin, GATA6, phosphorylated (p)SMAD1.5.9 and DAPI. **b,c**, Quantification of the normalized fluorescence of pSMAD1.5.9 in inner cell mass (ICM, *n* = 59) versus trophectoderm (TE, *n* = 80 cells) in segregating blastocysts and between epiblast (EPI, *n* = 77 cells), hypoblast (HYPO, *n* = 142 cells) and TE (*n* = 258 cells) of segregated blastocysts. **d**, Immunofluorescence images of day 7 (*N* = 9 embryos) and day 9 (*N* = 4 embryos) human embryos following in vitro implantation stained for SOX2/PODLX, GATA6, pSMAD1.5.9 and DAPI. **e,f**, Quantification of normalized fluorescence of pSMAD1.5.9 in EPI versus HYPO at day 7 (EPI *n* = 162 cells, HYPO *n* = 236 cells) and day 9 (EPI *n* = 29 cells, HYPO *n* = 87 cells). **g,h**, Plot

of normalized pSMAD1.5.9 fluorescence over time in EPI (**g**; *n* = 44 day 5, 68 day 6, 162 day 7 and 29 day 9 cells) and HYPO (**h**; *n* = 20 day 5, 90 day 6, 236 day 7 and 87 day 9 cells). **i**, Quantification of normalized pSMAD1.5.9 fluorescence in HYPO cells at day 7 and day 9 depending on contact with EPI. **j**, High-power images of pSMAD1.5.9 in day 9 HYPO cells either contacting or not contacting the EPI. Regions shown here are outlined with a dashed box in **d**. For violin plots, central dotted line denotes median and dotted lines mark the 25th and 75th quartiles. Error bars denote standard error (**g** and **h**). Statistical tests: two-sided Mann–Whitney test (**b**, **e** and **f**); Kruskal–Wallis with Dunn's post-hoc (**c** and **g**–**i**). ****$P$ < 0.0001, ***$P$ < 0.001, **$P$ < 0.01, *$P$ < 0.05. Exact $P$ values presented in Supplementary Table 8. Scale bars: 25 μm (**a**), 100 μm (**b** and **e**) and 20 μm (**k**).

differences in BMP dependence between mouse and human are related to divergences in the pluripotency regulatory network during exit from the naive state.

## BMP signalling is required for human anterior hypoblast

Strikingly, BMP inhibition by LDN treatment of human embryos from day 7 to day 9 ablated the anterior hypoblast, despite no change in total

hypoblast cell numbers (Fig. 5e–h), revealing an essential role for BMP signalling in anterior hypoblast maintenance following implantation. In mouse, LDN treatment conversely increased the formation of the AVE (Extended Data Fig. 8i,j), in line with an inhibitory role of Bmp4. Further, we observed that LDN treatment during differentiation of YSLC from ES cells resulted in a decrease in GATA6 at all points in differentiation (Fig. 5l,m and Extended Data Fig. 8q). The addition of LDN to YSLC also reduced the proportion of CER1-positive cells but only at the endpoint of differentiation (Fig. 5n). This phenotype recapitulated the requirement for BMP in anterior hypoblast maintenance but not specification in the human embryo. Further, conditioned medium from YSLCs differentiated with LDN treatment during this later window showed reduced capacity to protect epiblast-like human ES cell spheroids, indicating functional impairment in the generation of secreted signalling antagonists (Fig. 5o,p). Taken together, our results indicate that the role of BMP signalling in anterior specification diverges between the mouse and human.

### NOTCH/γ-secretase signalling is important for human epiblast

Next, we investigated the putative role of NOTCH signalling as predicted in our CellPhoneDB analysis (Fig. 1h). We found that NOTCH ligands *DLL3* and *JAG1* and receptor *NOTCH1* were expressed in the human epiblast, and the NOTCH effector *RBPJ* was increased upon embryo development through implantation in vitro (Extended Data Fig. 3b). To address the role of NOTCH signalling in the human embryo, we inhibited γ-secretase, which cleaves NOTCH receptors to generate the NOTCH intra-cellular domain. We found that γ-secretase inhibition (with DAPT, Compound-E or MK-0752) from day 5–7 embryos had minimal effect on total SOX2-positive epiblast or GATA6-positive hypoblast numbers. Similarly, epiblast and hypoblast were unaffected in E3.5 + 48 h mouse embryos treated with DAPT (Fig. 6a–d and Extended Data Fig. 9a–d). In contrast, DAPT addition between days 7 and 9 significantly decreased both epiblast and hypoblast total cell numbers. Compound-E and MK-0752 treatment also decreased the number of epiblast cells, albeit to a lesser extent (Fig. 6e–h and Extended Data Fig. 9b). Only γ-secretase inhibition using DAPT in human embryos showed a decrease in hypoblast cell numbers at day 9, indicating this loss may be related to the more drastic effect on the epiblast in this condition, rather than a direct effect on the hypoblast (Fig. 6e,g and Extended Data Fig. 9b). Mouse embryos treated with DAPT from E5.0 for an additional 36 h exhibited reduced epiblast length and aberrantly displayed epiblast cells within the pro-amniotic cavity, but no difference in total epiblast area (Extended Data Fig. 9f–j). Interestingly, 48-h γ-secretase inhibition (DAPT) of naive or primed human ES cells in adherent monolayer conditions (Matrigel-coated dishes) did not change the proportion of cleaved caspase-3-positive apoptotic cells relative to controls (Fig. 6i,j). In contrast, DAPT treatment of human ES cell-derived spheroids, which

more closely resemble the epiblast architecture[7,56], increased apoptosis, replicating the loss of the epiblast in human embryos post-implantation (Fig. 6i–k).

DAPT-treated mouse embryos also exhibited a loss of the AVE (Extended Data Fig. 9f–h). Similarly, all three γ-secretase inhibitors used to treat human embryos caused a reduction in CER1-positive anterior hypoblast cells, though their effects varied in severity and stage (Fig. 6e,h and Extended Data Fig. 9a). To investigate if NOTCH/γ-secretase inhibition within the epiblast has an indirect impact on the hypoblast, we utilized our directed YSLC differentiation approach[48] in combination with γ-secretase inhibition. DAPT treatment did not prevent GATA6 expression. However, both initiation and maintenance of CER1 expression were reduced (Fig. 6l,m), suggesting a cell-autonomous role for NOTCH/γ-secretase in anterior hypoblast. Additionally, conditioned media from YSLCs differentiated in the presence of DAPT showed reduced functionality in the protection of epiblast-like spheroids (Fig. 6n,o). Together, these data support a species-conserved role of NOTCH/γ-secretase-mediated signalling in anterior specification, as well as in post-implantation epiblast survival in the human embryo.

## Discussion

Our analysis of signalling dynamics during peri-implantation embryo development and anterior hypoblast specification revealed conserved and divergent roles of NODAL, BMP and NOTCH during early development across species. Taken together, our results demonstrate a crucial role of NODAL, BMP and NOTCH in human anterior hypoblast formation and suggest implantation as a switch point in BMP, NODAL and NOTCH signalling activity, particularly in the transition of the epiblast and maturation of the hypoblast (Extended Data Fig. 10).

The formation of the extra-embryonic anterior signalling centre is conserved in mouse and human, while the morphology and timing differ[10]. We demonstrate that, as in mouse, NODAL signalling is required in human for both the initial specification of the CER1-positive hypoblast and maintenance of CER1-positive cells. In contrast to the mouse, however, we find that BMP signalling is necessary for the maintenance of the anterior hypoblast after implantation. Importantly, by day 9, both BMP and NODAL show enriched activity in hypoblast cells directly underlying the epiblast. As both NODAL and BMP appear necessary for the maintenance of the anterior hypoblast, which underlies the epiblast, this pattern of enrichment supports their role. Further, the divergence in signalling activity between cells underlying the epiblast (visceral endoderm-like) and those lining the putative yolk sac cavity in post-implantation stages suggests the existence of spatially organized hypoblast-derived subpopulations with distinct signalling environments. NOTCH signalling also appears important in anteriorization as we observed a decrease in CER1-positive cells after γ-secretase inhibition in human and a loss of AVE formation in mouse embryos. This may

**Fig. 5 | BMP signalling is enriched in pre-implantation epiblast and important for maintenance of the anterior hypoblast. a**, Immunofluorescence images of embryos cultured from day 5 to day 7 in control (*N* = 44 embryos), 200 nM LDN (*N* = 8 embryos) or 50 ng ml⁻¹ BMP6 (*N* = 8 embryos) conditions. **b–d**, Quantification of the number of epiblast (**b**), hypoblast (**c**) and CER1-positive hypoblast cells (**d**) at day 7. **e**, Immunofluorescence images of embryos cultured from day 7 to day 9 in control (*N* = 39 embryos), 200 nM LDN (*N* = 6 embryos) or 50 ng ml⁻¹ BMP6 (*N* = 6 embryos) conditions. **f–h**, Quantification of the number of epiblast (**f**), hypoblast (**g**) and CER1-positive hypoblast cells (**h**) at day 9. **i**, Pearson regression correlation coefficient *ID2* expression with naive pluripotency markers. **j,k**, Immunofluorescence and quantification of pSMAD1.5.9 in PXGL (naive: 3,057 cells) and mTeSR (primed: 5,761 cells) human (h)ES cells and in mouse (m)ES cells in 2iLif conditions (naive; 297 cells) or basal N2B27 (naive exit; 789 cells). *N* = 2 experiments. **l**, Immunofluorescence images of differentiated YSLCs ± 200 nM LDN for 48 h during specification or maturation of YSLCs. **m**, Quantification of GATA6 fluorescence levels in relative to DAPI and normalized to control. D2–D4: control *n* = 4,085; LDN *n* = 4,777. D4–D6: control *n* = 4,139; LDN

*n* = 4,461. *N* = 2 experiments. **n**, Percentage of GATA6-positive cells that are CER1 positive. D2–D4: control *n* = 2,502; LDN *n* = 2,796. D4–D6: control *n* = 2,500; LDN *n* = 2,425. *N* = 2 experiments. **o,p**, Immunofluorescence images and quantification of pSMAD1.5.9 expression in hES cell-derived spheroids cultured in mTeSR+ medium (mTeSR+ *n* = 1,060 cells) or mTeSR+ medium conditioned on either control YSLC (YSLC CM *n* = 1,836 cells) or on YSLC differentiated with 48 h of LDN treatment (YSLC + LDN CM *n* = 1427 cells). *N* = 3 experiments. For box plots, box encompasses 25th and 75th quartiles with median marked by central line. Minimum and maximum and denoted by whiskers. Plus symbol marks the mean. Error bars denote standard error (**k**, **m–n** and **p**). Statistical tests: two-sided Mann–Whitney test (**b–d** and **f–h**); coefficients were tested (cor.test; two-tailed) and corrected for multiple hypothesis testing with the Benjamini–Hochberg method (**i**); two-sided unpaired *t*-test (**k**, **m** and **n**); one way analysis of variance with Tukey–Kramer post-hoc (**p**). ****P* < 0.0001, ****P* < 0.001, **P* < 0.05. Unmarked pairwise comparisons are not significant (NS). Exact *P* values presented in Supplementary Table 8. Scale bars: 100 μm (**a** and **e**) and 50 μm (**j**, **l** and **o**).

be partly due to downstream loss of NODAL as has been reported in mouse[57]. The distinct roles of NODAL, BMP and NOTCH signalling we report here in human embryos could be recapitulated in YSLCs in vitro[48].

Concomitant with anterior hypoblast formation, the mammalian epiblast transitions from a naive state of pluripotency to a primed state[10,49,58], and these distinct stages can be captured in ES cell-derived human and mouse embryo models[59,60]. However, the conditions used to capture these distinct pluripotency states differ between mouse and human, indicative of evolutionary divergence in the pluripotency network. Here we find that the pre-implantation human epiblast

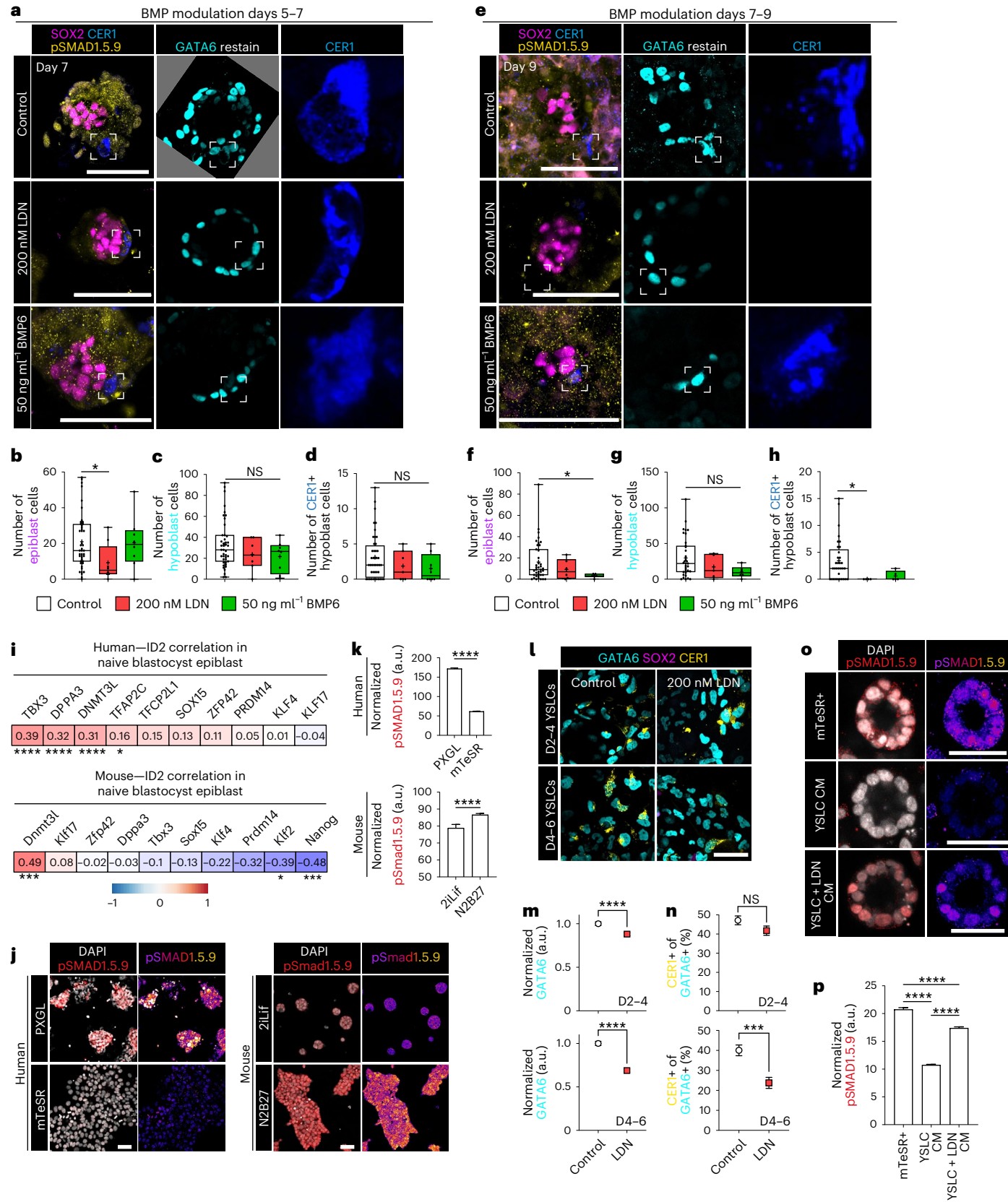

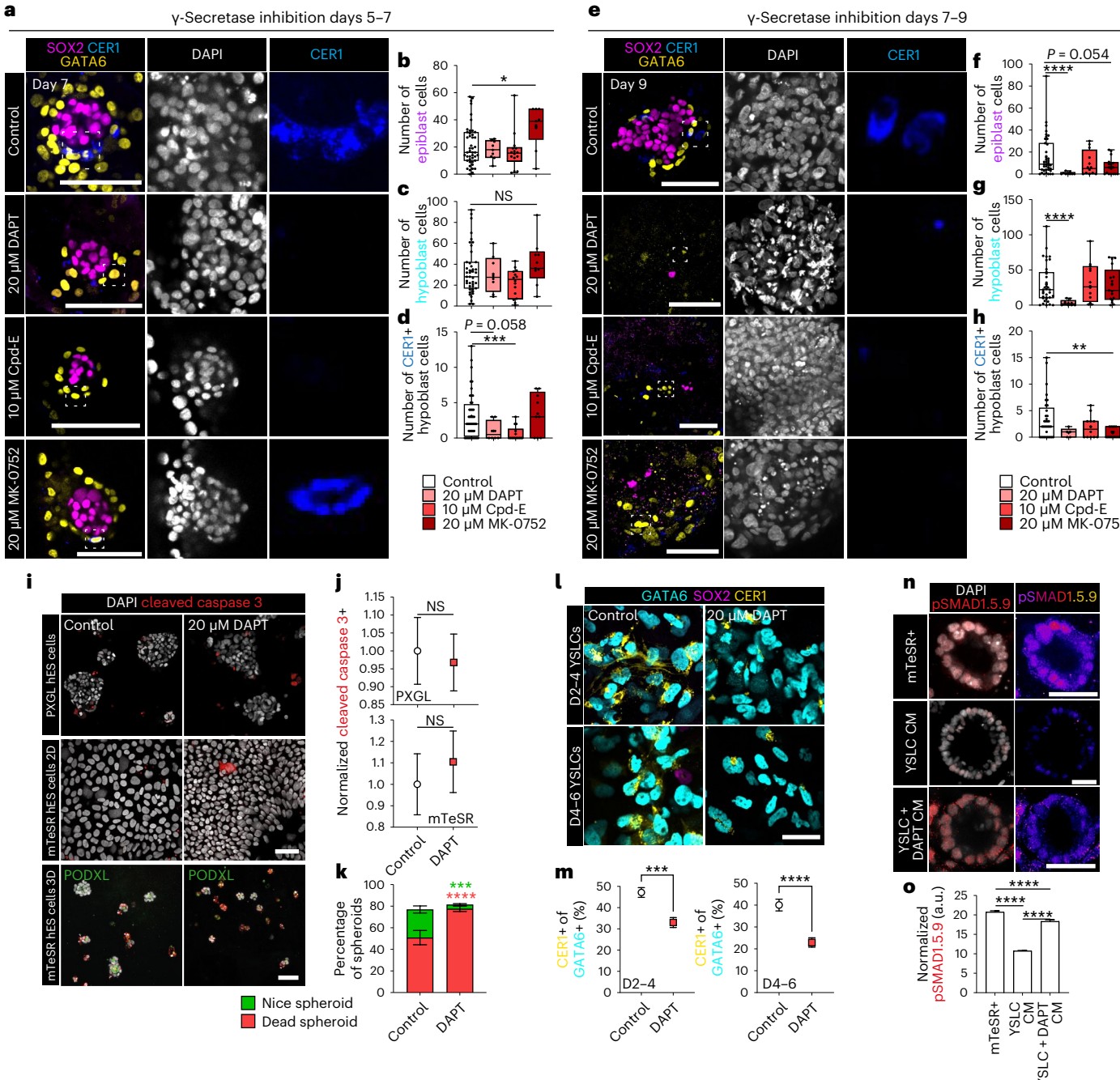

**Fig. 6 | Functional interrogation of NOTCH signalling in human peri-implantation. a**, Immunofluorescence images of embryos cultured from day 5 to day 7 in control conditions ($N = 44$ embryos), 20 μM DAPT ($N = 8$ embryos), 10 μM Compound-E ($N = 14$ embryos) or 20 μM MK-0752 ($N = 9$ embryos). **b–d**, Quantification of the number of epiblast (**b**), hypoblast (**d**) and CER1-positive hypoblast cells (**d**) at day 7. **e**, Immunofluorescence images of embryos cultured from day 7 to day 9 in control conditions ($N = 39$ embryos), DAPT ($N = 9$ embryos), Compound-E ($N = 12$ embryos) or MK-0752 ($N = 17$ embryos). **f–h**, Quantification of the number of epiblast (**f**), total hypoblast (**g**) and CER1-positive hypoblast cells (**h**) at day 9. **i–k**, Immunofluorescence images and quantification of 2D naive PXGL and primed human (h)ES cells treated for 48 h in control conditions (PXGL $n = 2,336$, primed $n = 7,796$) or 20 μM DAPT (PXGL $n = 2,386$, primed $n = 6,136$; $N = 2$ experiments) and 3D spheroids derived from primed hES cells cultured in control conditions ($n = 364$) or DAPT ($n = 643$; $N = 2$ experiments). Cells were stained for SOX2 and cleaved caspase 3 (Ccasp3) with spheroids additionally stained for PODLX. **l**, Immunofluorescence images of the differentiation of YSLCs that were treated with DAPT for 48 h during YSLC

specification or maturation. **m**, Percentage of GATA6-positive cells that are CER1 positive. D2–D4: control $n = 2,502$; DAPT $n = 2,690$. D4–D6: control $n = 2,500$; DAPT $n = 2,701$; $N = 2$ experiments. **n**, Immunofluorescence images of pSMAD1.5.9 expression in hES cell-derived spheroids cultured in mTeSR+ medium (mTeSR+ $n = 1,060$ cells) or mTeSR+ medium conditioned on either control YSLC (YSLC CM $n = 1,836$ cells) or on YSLC differentiated with 48 h of DAPT treatment (YSLC + DAPT CM $n = 501$ cells). $N = 3$ experiments. **o**, Quantification of normalized pSMAD1.5.9 fluorescence from **n**. For box plots, box encompasses 25th and 75th quartiles with median marked by central line. Minimum and maximum and denoted by whiskers. Plus symbol marks the mean. Error bars denote standard error (**j**, **k** and **m–o**). Statistics: two-sided Mann–Whitney test (**b–d** and **f–h**); two-sided unpaired $t$-test (**j** and **m**); two-way analysis of variance (ANOVA) with Šidák's multiple comparison's test (**k**); one way ANOVA with Tukey–Kramer post-hoc (**o**). $P < 0.06$ is noted. $****P < 0.0001$, $***P < 0.001$, $**P < 0.01$, $*P < 0.05$. Unmarked comparisons to control are not significant (NS). Exact $P$ values presented in Supplementary Table 8. Scale bars: 100 μm (**a** and **e**) and 50 μm (**i**, **l** and **n**).

exhibits high levels of both NODAL and BMP activity compared with the post-implantation epiblast. In line with this observation, BMP inhibition before, but not after, implantation decreases the number of human epiblast cells. These results contrast with analogous mouse treatments, where BMP or NODAL inhibition decreased epiblast size post-implantation. In vitro models of human and mouse naive and primed pluripotency also recapitulated these stage-based divergences in BMP activity between species. In addition, γ-secretase/NOTCH inhibition demonstrated distinct effects on the pre- or post-implantation human embryo, having minimal effects on epiblast cell number before implantation and decreasing this population when treated immediately following implantation. In mouse, γ-secretase/NOTCH inhibition altered epiblast tissue integrity with cells present in the pro-amniotic cavity, but did not appear to affect survival. NOTCH is active in the mouse epiblast from implantation to gastrulation[61]; however, its role is not well understood. Here we show its role in the survival of the human epiblast during its transition from a more disorganized pre-implantation state to the epithelialized rosette. The γ-secretase complex is known to mediate cleavage of several substrates in addition to NOTCH, including E-Cadherin, ERBB4, LRP6 and IGF1R[62]. Further work specifically targeting the NOTCH pathway will be essential for confirming its role in post-implantation development.

While we focus largely on the inner cell mass-derived lineages, our scRNA-seq analysis may also be used to identify key modulators of trophoblast development including terminal differentiation of the primate-specific syncytiotrophoblast and extravillous trophoblast. We show that NODAL signalling from the epiblast may drive maturation of the human trophectoderm to implantation-competent trophoblast. Previous work has shown that isolated inner cell masses in MEK and NODAL inhibitors generated trophectoderm cell outgrowths robustly and that these conditions can be used to differentiate trophectoderm cells from naïve human ES cells[49]. However, the addition of A83-01 from day 5 to day 7 does not appear to drive differentiation of the pre-implantation epiblast to trophectoderm. This suggests that either the combinatorial inhibition of both FGF-MAPK and NODAL could be crucial for trophectoderm differentiation, or that the differentiation of inner cell mass outgrowths and naive ES cells towards trophectoderm follows distinct pathways compared with trophectoderm specification in the morula. BMP signalling has also been associated with human trophoblast, and yet its role, tissue expression pattern and potential evolutionary divergence remain controversial[46,49,63-66]. BMP6 treatment alone may not be indicative of other BMP ligands' effects. Of note, the extra-embryonic mesenchyme and amnion in primates are likely to be sources of BMP ligands in addition to the hypoblast[20,21,34,35,67]; however, these populations do not differentiate robustly during in vitro culture of human embryos in current systems[10,24-27,46]. Further experiments will be required to fully investigate the pleiotropic role of BMP during human implantation.

In summary, we have sought to uncover roles of multiple signalling pathways operating during human implantation, which should be useful in efforts to improve stem cell-derived human embryo-like models while further contributing to our foundational knowledge of this crucial developmental period.

## Online content

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

## Methods

### Ethics statement

Human embryo work was regulated by the Human Fertility and Embryology Authority under licence R0193. Approval was obtained from the Human Biology Research Ethics Committee at the University of Cambridge (reference HBREC.2021.26). All work is compliant with the 2021 International Society for Stem Cell Research (ISSCR) guidelines. Patients undergoing IVF at CARE Fertility, Bourn Hall Fertility Clinic, Herts & Essex Fertility Clinic, and King's Fertility were given the option of continued storage, disposal or donation of embryos to research (including research project specific information) or training at the end of their treatment. Patients were offered counselling, received no financial benefit and could withdraw their participation at any time until the embryo had been used for research. Research consent for donated embryos was obtained from both gamete providers. Embryos were not cultured beyond day 14 post-fertilization or the appearance of the primitive streak. Human stem cell work was approved by the UK Stem Cell Bank Steering Committee (under approval SCSC21-38) and adheres to the regulations of the UK Code of Practice for the Use of Human Stem Cell Lines. Mice were kept in an animal house in individually ventilated housing on 12:12 h light–dark cycle with ad libitum access to food and water. Ambient temperature was maintained at 21–22 °C and humidity at 50%. Experiments with mice are regulated by the Animals (Scientific Procedures) Act 1986 Amendment Regulations 2012 and carried out following ethical review by the University of Cambridge Animal Welfare and Ethical Review Body. Experiments were approved by the Home Office under licences 70/8864 and PP3370287. CD1 wild-type males aged 6–45 weeks and CD1 wild-type females aged 6–18 weeks were used for this study. Animals were inspected daily, and those showing health concerns were culled by cervical dislocation.

### Sequencing analysis and code availability

Raw fastq files from human datasets[26,27,36,45], cynomolgus monkey datasets[28,35] and mouse datasets[68–71] were obtained from public repositories with wget. All human datasets were aligned to the GRCh38 reference using kb-python's kb ref function to generate a reference. For cynomolgus monkey, National Center for Biotechnology Information (NCBI) genome build 5.0 transcriptome fasta files were adjusted to Ensembl style and used in kb ref to generate a custom index. For the mouse, GRCm39 reference was used with kb ref to generate a custom index. All datasets were re-aligned using either kb-python or kallisto[72,73], after data handling as below. Human datasets: 10x v2 data from Molè et al. were processed as previously described[10]. For Zhou et al.[27], read1 files were trimmed using cutadapt[74] for the reported adapter sequence. Trimmed reads were then aligned using the kb-python kb count function with custom specifications (-x 1,0,8:1,8,16:0,0,0) and the custom barcode whitelist available. Each pair of fastqs was processed individually into barcode–gene matrices and concatenated. For Xiang et al.[26], a batch file was generated with cell ID, read1 and read2 for each fastq pair listed. Kallisto's pseudo –quant command was then used to generate a cell ID–gene matrix. For Blakely et al.[36], reads were aligned using kallisto pseudo –quant. For Petropoulos et al.[45], single-end reads were processed with kallisto pseudo –quant with a pre-made batch file as above with 43 base pair read length specified. Cynomolgus datasets: for Ma et al.[28], read1 fastqs were trimmed using cutadapt for TSO and polyA tail as described in the original publication. Next, kb python's kb count function was used with custom specifications (−x 1,0,8:1,8,16:0,0,0). For Yang et al.[35], reads were aligned using kb python's kb count command with 10xv3 technology specified. For Nakamura et al.[21], available count tables were used given the use of SOLiD sequencer limiting re-alignment program options. Mouse datasets: for Mohammed et al.[70], kallisto pseudo –quant with a generated batch file was used to generate a cell ID–gene matrix. For Deng et al.[69] and Cheng et al.[68], single-end reads were aligned with kallisto pseudo –quant. Finally, for Pijuan-Sala et al.[71], each sample set of 33 fastq files was aligned with kb count, with

10xv1 technology specified. The resulting set of barcode–gene matrices was then concatenated for downstream analysis.

Following re-alignment, any datasets not generated using unique molecular identifier counts were normalized using quminorm[75]. First, matrices were converted to transcripts per kilobase million (TPM), and then the TPM matrix ran through quminorm with a shape parameter up to a maximum of 2 that did not create not available/applicable (NA) values in the matrix. Then, each individual dataset was made into a Seurat object[76]. Each individual dataset was then merged into a species-specific Seurat object, with SCT batch correction applied across datasets. Clusters were identified on the basis of canonical marker expression. To perform module scoring, gene lists were obtained from rWikiPathways[77]. For the monkey and mouse, gene symbols were converted to human homologues using bioMart[78]. Seurat's AddModuleScore function was used with WikiPathway gene lists of interests as input. For CellPhoneDB analysis[41], human data were split on the basis of stage, and subset matrix and metadata for cell type were output as txt files. CellPhoneDB was then run with respective files and –counts-data set to gene_name. Data visualization was performed using Seurat's DimPlot, FeaturePlot and VlnPlot functions, Scillus' (https://scillus.netlify.app) Plot_Measure function, pheatmap and CellPhoneDB's dotplot function.

The scripts used for analyses are available at ref. 79.

### Thawing and in vitro culture of human embryos

Human embryos were thawed and cultured as described previously[10,24]. Briefly, cryopreserved human blastocysts (day 5 or 6) were thawed using the Kitazato thaw kit (VT8202-2, Hunter Scientific) according to the manufacturer's instructions. The day before thawing, TS solution was placed at 37 °C overnight. The next day, IVF straws were submerged in 1 ml pre-warmed TS for 1 min. Embryos were then transferred to DS for 3 min, WS1 for 5 min and WS2 for 1 min. These steps were performed in reproplates (REPROPLATE, Hunter Scientific) using a STRIPPER micropipette (Origio). Embryos were incubated at 37 °C and 5% $CO_2$ in normoxia and in pre-equilibrated human IVC1 supplemented with 50 ng ml$^{-1}$ insulin growth factor-1 (IGF1) (78078, STEMCELL Technologies) under mineral oil for 1–4 h to allow for recovery. Following thaw, blastocysts were briefly treated with acidic Tyrode's solution (T1788, Sigma) to remove the zona pellucida and placed in pre-equilibrated human IVC1 in eight-well μ-slide tissue culture plates (80826, Ibidi) in approximately 400 μl volume per embryo per well. Half medium changes were done every 24 h. For small-molecule experiments, human IVC1 was supplemented with either 2 μM A83-01 (72022, STEMCELL Technologies)[80,81], 25 ng ml$^{-1}$ Activin-A (Qk001, QKINE)[82–84], 200 nM LDN (S2618, SelleckChem)[85,86], 50 ng ml$^{-1}$ BMP6 (SRP3017, Sigma Aldrich)[85,86], 20 μM DAPT (72082, STEMCELL Technologies)[87–90], 10 μM Compound-E (ab142164, Abcam)[91–93], 20 μM MK-0752 (S2660, Selleck Chemicals)[94–96] or dimethyl sulfoxide (DMSO) for 48 h. In all cases, these concentrations fall within a range of those used for either vertebrate embryos or complex human ES cell-derived models of the embryo. Within these ranges, a low-to-intermediate concentration was selected to avoid non-specific cytotoxic effects while still considering the higher concentration needed for embryo permeation compared with minimal 2D cell culture to achieve inhibitor action. Further, all small molecules and proteins were tested on human ES cells to validate the efficacy and test for cytotoxicity. For analysis, embryos were fixed in 4% paraformaldehyde for 20 min at room temperature for downstream analysis.

### Recovery of mouse embryos and in vitro culture

Pregnant, time-staged mice were culled by cervical dislocation, and uteri were dissected and placed in M2 medium (pre-warmed if embryos were for in vitro culture, ice cold if for fixing). E3.5 blastocysts were flushed out of uteri of pregnant females and either fixed for immunofluorescence analysis or transferred to acidic Tyrode's solution for zona pellucida removal. Embryos were cultured for

48 h in CMRL (11530037, Thermo Fisher Scientific) supplemented with 1× B27 (17504001, Thermo Fisher Scientific), 1× N2 (made in-house), 1× penicillin–streptomycin (15140122, Thermo Fisher Scientific), 1× GlutaMAX (35050-038, Thermo Fisher Scientific), 1× sodium pyruvate (11360039, Thermo Fisher Scientific), 1× essential amino acids (11130-036, Thermo Fisher Scientific), 1× non-essential amino acids (11140-035, Thermo Fisher Scientific) and 1.8 mM glucose (G8644, Sigma) supplemented with 20% foetal bovine serum[5,28]. Embryos were incubated with 25 ng ml⁻¹ Activin-A, 200 nM LDN, 50 ng ml⁻¹ BMP6, 20 μM DAPT or DMSO for 48 h. For E4.5, E5.5 and E5.75 collections, embryos were dissected directly from the uteri and fixed for analysis. For E5.0 collection, embryos were dissected from the uteri, and Reichert's membrane was removed before culturing or 36 h with relevant small molecules as described above.

## Human ES cell culture
Shef6 human ES cells (R-05-031, UK Stem Cell Bank) were routinely cultured on 1.6% v/v Matrigel (354230, Corning) in mTeSR1 medium (85850, STEMCELL Technologies) at 37 °C and 5% $CO_2$. Cells were passaged every 3–5 days with TrypLE Express Enzyme (12604-021, Thermo Fisher Scientific). The ROCK inhibitor Y-27632 (72304, STEMCELL Technologies) was added for 24 h after passaging. Cells were routinely tested for mycoplasma contamination by polymerase chain reaction. To convert primed human ES cells to RSeT or PXGL naive conditions, cells were passaged onto mitomycin-C inactivated CF-1 MEFs ($3 × 10^3$ cells cm⁻²; GSC-6101G, Amsbio) in human ES cell medium containing Dulbecco's modified Eagle medium (DMEM)/F12 supplemented with 20% Knockout Serum Replacement (10828010, Thermo Fisher Scientific), 100 μM β-mercaptoethanol (31350-010, Thermo Fisher Scientific), 1× GlutaMAX (35050061, Thermo Fisher Scientific), 1× non-essential amino acids, 1× penicillin–streptomycin and 10 ng ml⁻¹ FGF2 (University of Cambridge, Department of Biochemistry) and 10 μM ROCK inhibitor Y-27632 (72304, STEMCELL Technologies). For RSeT conversion, cells were switched to RSeT medium (05978, STEMCELL Technologies). Cells were maintained in RSeT and passaged as above every 4–6 days. For PXGL conversion, previously described protocols were used[97]. Briefly, cells were cultured in hypoxia and medium was switched to chemically Resetting Media 1 (cRM-1), which consists of N2B27 supplemented with 1 μM PD0325901 (University of Cambridge, Stem Cell Institute), 10 ng ml⁻¹ human recombinant LIF (300-05, PeproTech) and 1 mM valproic acid. N2B27 contains 1:1 DMEM/F12 and Neurobasal A (10888-0222, Thermo Fisher Scientific) supplemented with 0.5× B27 (10889-038, Thermo Fisher Scientific) and 0.5× N2 (made in-house), 100 μM β-mercaptoethanol, 1× GlutaMAX and 1× penicillin–streptomycin. cRM-1 was changed every 48 h for 4 days. Subsequently, medium was changed to PXGL–N2B27 supplemented with 1 μM PD0325901, 10 ng ml⁻¹ human recombinant LIF, 2 μM Gö6983 (2285, Tocris) and 2 μM XAV939 (X3004, Merck). PXGL cells were passaged every 4–6 days using TrypLE (12604013, Thermo Fisher Scientific) for 3 min, and 10 μM ROCK inhibitor Y-27632 and 1 μl cm⁻² Geltrex (A1413201, Thermo Fisher Scientific) were added at passage for 24 h.

For small-molecule experiments, primed or PXGL human ES cells were plated into ibiTreat dishes at normal passage densities. Forty-eight hours after passage, medium was changed to N2B27 supplemented with 25 ng ml⁻¹ Activin-A, 2 μM A83-01, 50 ng ml⁻¹ BMP6, 200 nM LDN or 20 μM DAPT. Plates were then fixed for 20 min in 4% paraformaldehyde for downstream analysis. For 3D culture of primed human ES cells, 30,000 cells were resuspended in 200 μl of ice-cold Geltrex and the resulting mix was plated into a single well of an 8 μ-well ibiTreat dish. Geltrex was polymerized by placement at 37 °C for 10 min. Two-hundred microlitres of mTeSR1 with ROCK inhibitor Y-27632 was added after polymerization. Twenty-four hours later, the medium was changed to N2B27 (±10 μM DAPT). Medium was refreshed 24 h later, and the plate was fixed in 4% paraformaldehyde for 30 min after a total of 48 h in experimental conditions. Conditioned medium experiments were performed as described previously[48]. Briefly, 80 μl of ice-cold Geltrex was added to an 8 μ-well ibiTreat dish to create a 100% Geltrex bed. This was polymerized at 37 °C for 4 min. A total of $1 × 10^3$ cells cm⁻² primed human ES cells were then added onto this bed in DMEM/F12 and allowed to settle for 15 min. After this, medium was carefully switched to conditioned medium (described below) with 5% Geltrex (v/v) and 10 μM ROCK inhibitor Y-27632. Conditioned medium with 5% Geltrex was refreshed daily for the next 2 days, and the resulting spheroids were fixed after a total of 72 h.

## Human YSLC culture
YSLC differentiation was carried out as published[48]. Briefly, Shef6 human ES cells cultured in RSeT medium for at least 2 weeks were plated onto ibiTreat dishes at $1 × 10^3$ cells cm⁻² in RSeT medium with 10 μM Y-27632. Medium was changed the next day to ACL differentiation medium consisting of N2B27 supplemented with 5% v/v Knockout Serum Replacement, 100 ng ml⁻¹ Activin-A, 3 μM CHIR99021 (University of Cambridge Stem Cell Institute) and 10 ng ml⁻¹ human recombinant LIF. Medium was refreshed every 48 h, and 2 μM A83-01, 200 nM LDN or 20 μM DAPT was added to ACL medium for 48 h from either day 2 to day 4, followed by fixation, or day 4 to day 6 followed by fixation. For conditioned medium experiments, at day 6 cells were washed three times with phosphate-buffered saline and then mTeSR Plus medium (100-0276; STEMCELL Technologies) was added for 24 h. Medium was collected from YSLCs and passed through a 0.45-μm filter (16555, Sartorious), and stored for up to 1 week at 4 °C.

## Mouse ES cell culture
CD1 mouse ES cells (generous gift from Prof. Jennifer Nichols (Stem Cell Institute, University of Cambridge, UK)) were routinely cultured on gelatin-coated (G7765, Sigma Aldrich) dishes in N2B27 supplemented with 1 μm PD0325901, 3 μm CHIR99021 and 10 ng ml⁻¹ mouse Lif (University of Cambridge, Stem Cell Institute). Medium was changed every 48 h, and cells were passaged every 3–5 days using trypsin–ethylenediaminetetraacetic acid (25300062; Life Technologies). For experiments, cells were passaged as normal into ibiTreat dishes. The following day, medium was switched to either N2B27 + 2iLif, N2B27, or N2B27 + 200 nM LDN. Medium was refreshed after 24 h, and cells were fixed after 48 h.

## Immunostaining
Embryos were fixed in 4% paraformaldehyde, permeabilized in 0.1 M glycine with 0.3% Triton X-100 and placed in blocking buffer containing 1% bovine serum albumin and 10% foetal bovine serum. Primary antibodies were diluted in blocking buffer and added overnight at 4 °C. Fluorescently tagged secondary antibodies were added for 2 h at room temperature. Primary antibodies used in this study are as follows: mouse monoclonal anti OCT3/4 (sc5279, Santa Cruz; 1:200 dilution), rat monoclonal anti SOX2 (14-19811-82, Thermo Fisher Scientific; 1:500 dilution), goat polyclonal anti NANOG (AF1997 R&D Systems; 1:500 dilution), rabbit monoclonal anti GATA6 (5851, Cell Signaling Technology; 1:2,000 dilution), goat polyclonal anti GATA6 (AF1700, R&D Systems; 1:200 dilution), mouse anti monoclonal Cdx2 (MU392-UC, Biogenex; 1:200 dilution), goat polyclonal anti CER1 (AF1075, R&D Systems; 1:250 dilution), rat monoclonal anti Cerebus1 (MAB1986, R&D Systems; 1:200 dilution), rabbit monoclonal anti Phospho-Smad1(Ser463/465)/Smad5(Ser463/465)/Smad9(Ser465/467) (13820T, Cell Signaling Technology; 1:200 dilution), rabbit monoclonal anti Smad2.3 (8685T, Cell Signaling Technology; 1:200 dilution), rabbit monoclonal anti-cleaved caspase 3 (9664, Cell Signaling Technology; 1:200 dilution), mouse monoclonal anti Podocalyxin (MAB1658, R&D Systems; 1:500 dilution), goat polyclonal anti Brachyury (AF2085, R&D Systems; 1:500 dilution), rat monoclonal anti GATA4 (14-9980-82, Thermo Fisher Scientific; 1:500 dilution), goat polyclonal anti AP2-gamma (AF5059, R&D Systems; 1:500 dilution), goat polyclonal anti Otx2 (AF1979, R&D

Systems; 1:1,000 dilution) and Alexa Flour 594 Phalloidin (A12381, Thermo Fisher Scientific; 1:500 dilution).

## Quantifications

Immunofluorescence images were captured on a Leica SP8 confocal and processed and analysed using Fiji (http://fiji.sc). Epiblast, hypoblast and CER1-positive cell numbers were manually counted using the multi-point cell counter plugin. Quantification of trophectoderm was performed using Imaris software (version 9.1.2) using the spots tool with manual curation. To quantify n/c SMAD2.3 in human and mouse embryos, the central three planes of individual cells were used to generate a three-plane z-stack. Individual 4′,6-diamidino-2-phenylindole (DAPI)-positive nuclei were used to generate a nuclear mask using the 'Analyze Particles' function on either the DAPI or lineage-associated transcription factor channel. The adjacent cytoplasmic area was drawn individually for each nucleus and the mean fluorescence of each region was measured, and the ratio computed. When embryos were stained with E-Cadherin, the membrane was delineated to allow for cytoplasmic region of interest determination. When embryos were stained with podocalyxin, the cytoplasmic region of interest was drawn to ensure delineation of a region captures suitable intra-cellular variation allowing for valid normalization. Measurements were computed on raw SMAD2.3 signal. To quantify pSMAD1.5.9 nuclear intensity, a nuclear mask generated on a central three-plane z-stack for each nucleus, and mean fluorescence values were measured. Within each three-plane z-stack, a background fluorescence taken adjacent to or within a cavity of the embryo was used for background normalization ($\frac{\text{mean nuclear intensity}}{1+\text{mean background intensity}}$). Background was normalized to (that is, to provide a comparable signal-to-noise ratio) rather than subtracted to account for the variability in laser penetration between experiments and z-planes. In stem cell experiments, nuclear masks were generated, mean fluorescence was measured, and all values were normalized to a control (DMSO) value of 1. To calculate the percentage of cleaved caspase-3-positive or CER1-positive cells, individual cells were manually counted using the cell counter plugin and presented as a percentage of all DAPI-negative or GATA6-positive cells. For 3D spheroid classification, the total number of structures was counted manually using the Cell Counter plugin, and each was assigned to a class of spheroid. For conditioned medium 3D spheroid quantifications, the central three planes of individual spheroids were used to generate a nuclear mask on the DAPI channel, and the mean nuclear pSMAD1.5.9 signal was quantified along with the signal of an acellular region for background normalization. To generate figures, images were processed by generating z-stacks of approximately five to ten planes to allow for visualization of embryo topology with cells on disparate planes followed by consistent adjustment of brightness and contrast.

## Statistics and reproducibility

No statistical method was used to pre-determine sample sizes. Sample sizes are similar to previous publications[7,10,24]. For characterization of normal development, embryos lacking any of the three lineages were excluded. Multiple SMAD fluorescence intensities were taken per lineage per embryo. All embryos were included in functional experiments and each cell type count is taken from individual embryos. All stem cell experiments were performed independently at least twice. Investigators were not blinded to group allocation during the experiment or analysis, as blinding would not have been possible due to medium preparation and changing requirements. Group allocation was not performed randomly; rather, based on visual assessment of embryos, investigators attempted to ensure balanced distributions of blastocysts/implanting embryos assessed as expanded with nice inner cell masses versus embryos that appeared delayed or with visible cell death across experimental groups. Statistical tests, except the Bayesian distribution model, were performed in Prism 9 (GraphPad), and where relevant, two-sided tests were used. Normality was tested with a Shapiro–Wilk test. Bayesian distribution modelling, which is suited to the small sample sizes used in human embryo studies, was used as a supplemental tool to assess how each small-molecule treatment affected the distribution of cell number. To do this, the brms R package was used[98,99], with the assumption of a Poisson distribution and the Control counts set to inform the priors and be used as reference. Brms' default Markov chain Monte Carlo settings were used. Coefficient ± credible intervals were either below −0.33, denoting a decrease in the distribution compared to control, or above 0.33, indicating an increase in the distribution compared to control. Credible intervals that bridged this range indicate no significant difference. All coefficients and credible intervals in addition to Mann–Whitney test P values are presented in Supplementary Table 8. For data presentation, box plots encompass the 25th to 75th percentile in the box, with the median marked by the central line, and the mean marked by a cross. The minimum and maximum are marked by the whiskers. For violin plots, the dashed line marks the median value and dotted lines mark the 25th and 75th percentiles. For summary plots (for example, Fig. 2h,i), the mean ± standard error of the mean is plotted.

## Reporting summary

Further information on research design is available in the Nature Portfolio Reporting Summary linked to this article.

## Data availability

All raw data used here are previously published and publicly available. For aligning sequencing data, GRCh38 (https://www.ncbi.nlm.nih.gov/assembly/GCF_000001405.26/), Genome assembly Macaca_fascicularis_5.0 (https://www.ncbi.nlm.nih.gov/datasets/genome/GCF_000364345.1/) and GRCm39 (https://www.ncbi.nlm.nih.gov/datasets/genome/GCF_000001635.27/) were used. For human data: Molè et al.[10], ArrayExpress E-MTAB-8060; Xiang et al.[26], Gene Expression Omnibus GSE136447; Zhou et al.[27], Gene Expression Omnibus GSE109555; Petropoulos et al.[45], ArrayExpress E-MTAB-3929; Blakely et al.[36], Gene Expression Omnibus GSE66507. For cynomolgus monkey data: Yang et al.[35], Gene Expression Omnibus GSE148683; Ma et al.[28], Gene Expression Omnibus GSE130114; Nakamura et al.[21], Gene Expression Omnibus GSE74767. For mouse data: Pijuan-Sala et al.[71], ArrayExpress E-MTAB-6967; Mohammed et al.[70], Gene Expression Omnibus GSE100597; Cheng et al.[68], Gene Expression Omnibus GSE109071; Deng et al.[69], Gene Expression Omnibus GSE45719. Scripts used for analysis are available at ref. 79. The integrated Seurat objects for each species are available on Zenodo[100] (https://doi.org/10.5281/zenodo.7689580). All other data supporting the findings of this study are available from the corresponding author on reasonable request. Source data are provided with this paper.

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

## Acknowledgements

The authors thank patients at CARE, Herts & Essex, Bourn Hall Fertility and King's Fertility Clinics for their generous donations, as well as the embryologists and members of each clinic for facilitating donations. We thank G. Serapio-García for advice on Bayesian statistical analysis and the bioinformaticians and data scientists who made their code, packages and vignettes available. This work is supported by Wellcome Trust (207415/Z/17/Z) and Open Atlas and NOMIS awards to M.Z.-G. B.A.T.W. was supported by the Gates Cambridge Trust. C.W.G. was supported by a Leverhulme Trust Early Career Fellowship. L.K.I.-S. was supported by the Rosetrees Trust.

## Author contributions

B.A.T.W., A.W., C.W.G. and L.K.I.-S. thawed and cultured human embryos for research. B.A.T.W. performed single-cell sequencing analyses. Z.B., A.B., A.C., P.C., C.D., P.E., S.F., S.G.V., M.L., R.O., C.P., N.C., L.R., A.M., L.W., L.C., K.E. and P.S. interfaced with patients, prepared informed consent documentation and prepared embryos for transfer from clinical to research setting. A.W. dissected and cultured mouse embryos. B.A.T.W. and C.W.G. cultured and differentiated human and mouse stem cells. B.A.T.W. and C.W.G. performed quantitative image analyses. B.A.T.W., C.W.G., A.W. and M.Z.-G. wrote the manuscript and conceived the study. B.A.T.W., A.W. and C.W.G. contributed equally.

## Competing interests

The authors declare no competing interests.

## Additional information

**Extended data** is available for this paper at https://doi.org/10.1038/s41556-024-01367-1.

**Correspondence and requests for materials** should be addressed to Magdalena Zernicka-Goetz.

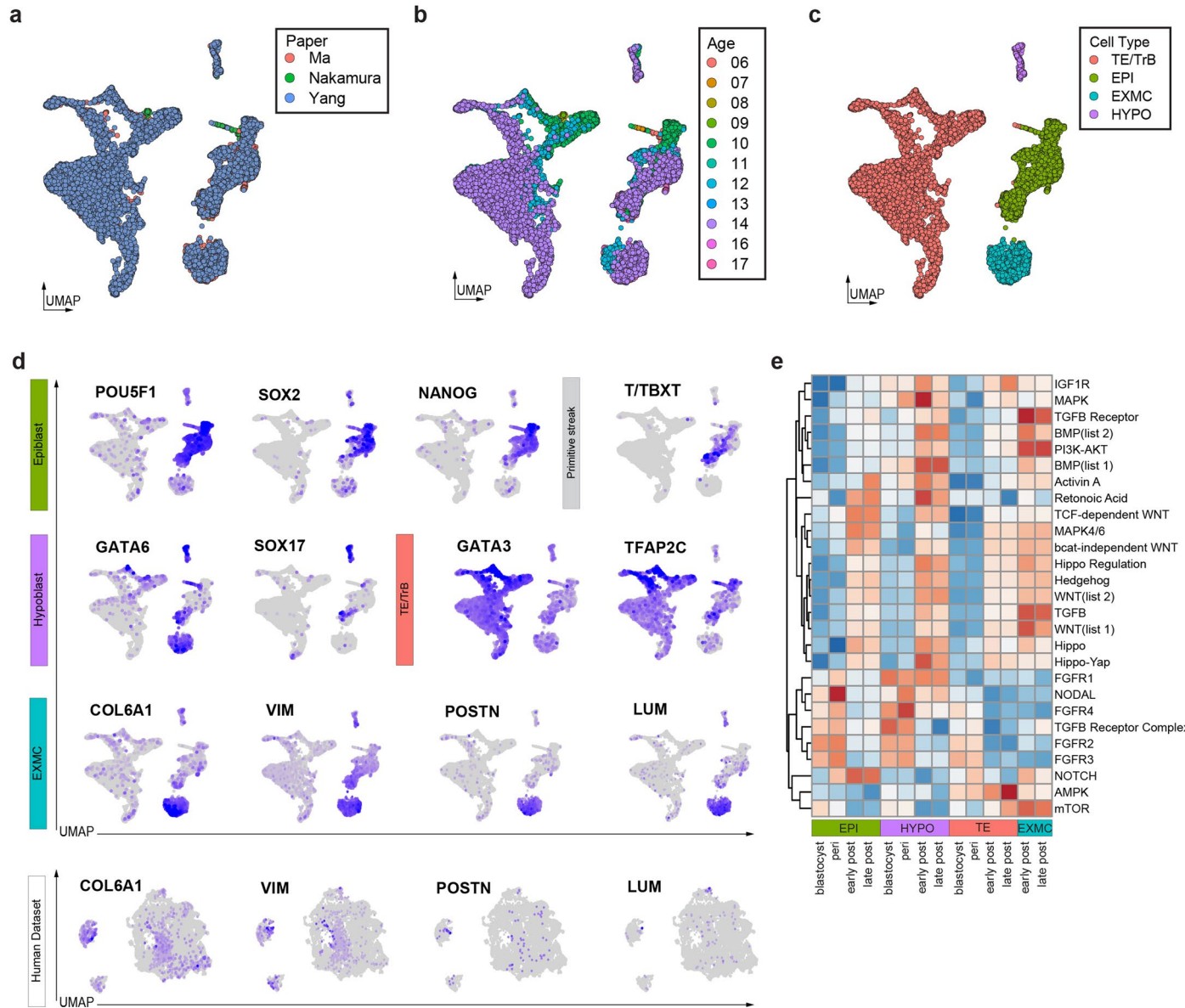

**Extended Data Fig. 1 | Generation of combined cynomolgus monkey dataset.**
**a**, Uniform Manifold Approximation Projection (UMAP) of cynomolgus monkey
scRNA-seq datasets colored by original publication. N = 10221 single cells.
**b**, UMAP of monkey datasets colored by age of embryo. **c**, UMAP of monkey
datasets colored by assigned cell type. **d**, UMAPs of monkey datasets colored by
a gradient indicating gene expression for canonical markers of epiblast (EPI),

hypoblast (HYPO), extraembryonic mesenchyme (EXMC), primitive streak, and
trophectoderm/trophoblast (TE/TrB). Extraembryonic mesenchyme markers are
also plotted on human dataset. **e**, Heatmap of average WikiPathways signaling
pathway module score split by cell type and stage. Visualized value is calculated
average scaled score from individual single-cell module scores. Gene names were
converted from mfac to hgnc gene symbols for module scoring.

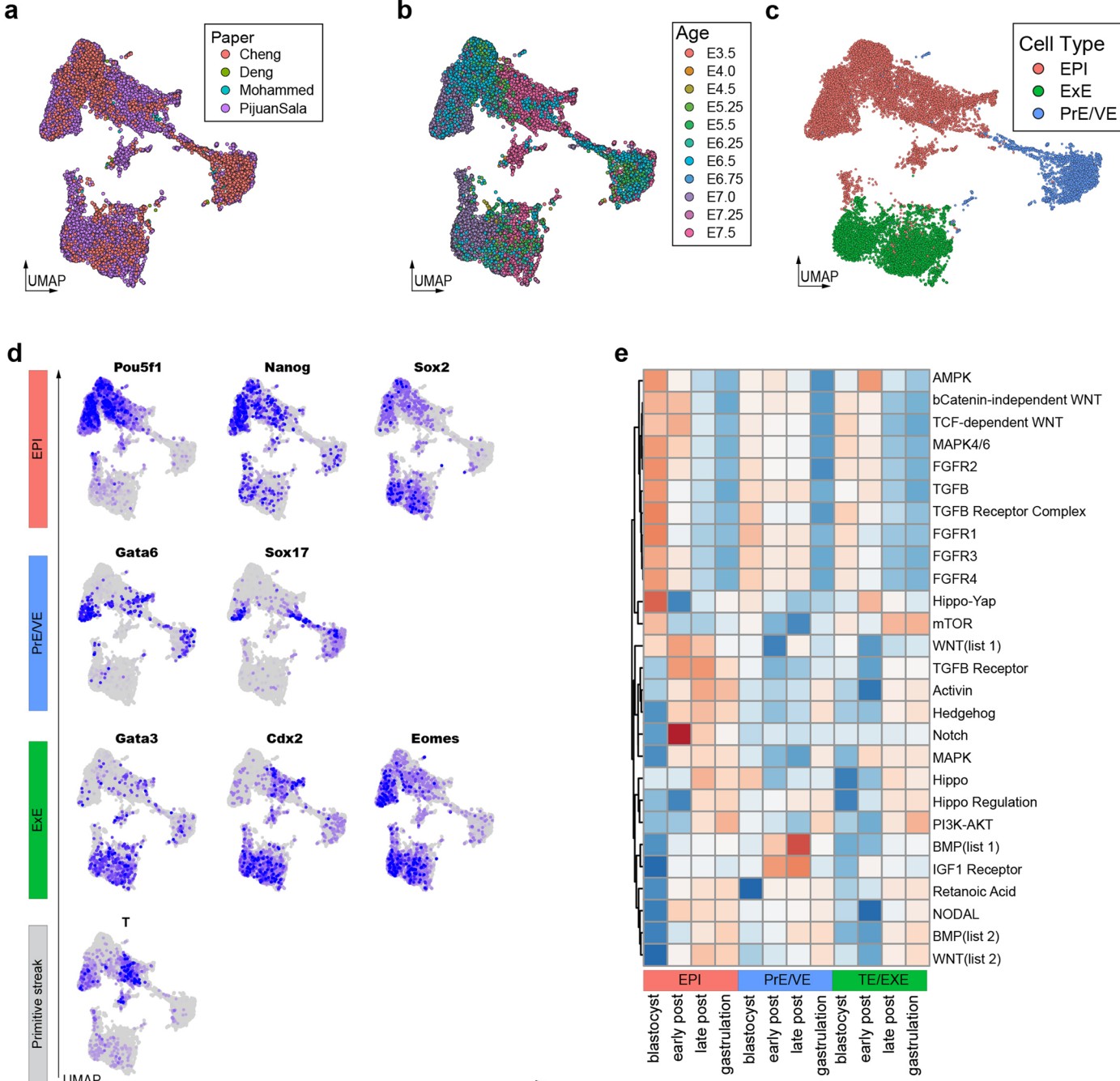

**Extended Data Fig. 2 | Generation of combined mouse dataset. a**, Uniform Manifold Approximation Projection (UMAP) of mouse scRNA-seq datasets colored by original publication. N = 16395 single cells. **b**, UMAP of mouse datasets colored by age. **c**, UMAP of mouse datasets colored by assigned cell type. **d**, UMAPs of mouse datasets colored by a gradient indicating gene expression for canonical markers of epiblast (EPI), primitive endoderm/visceral endoderm (PrE/VE), primitive streak, and trophectoderm/extraembryonic ectoderm (TE/ExE). **e**, Heatmap of average WikiPathways signaling pathway module score split by cell type and stage. Visualized value is calculated average scaled score from individual single-cell module scores. Gene names were converted from mmus to hgnc gene symbols for module scoring.

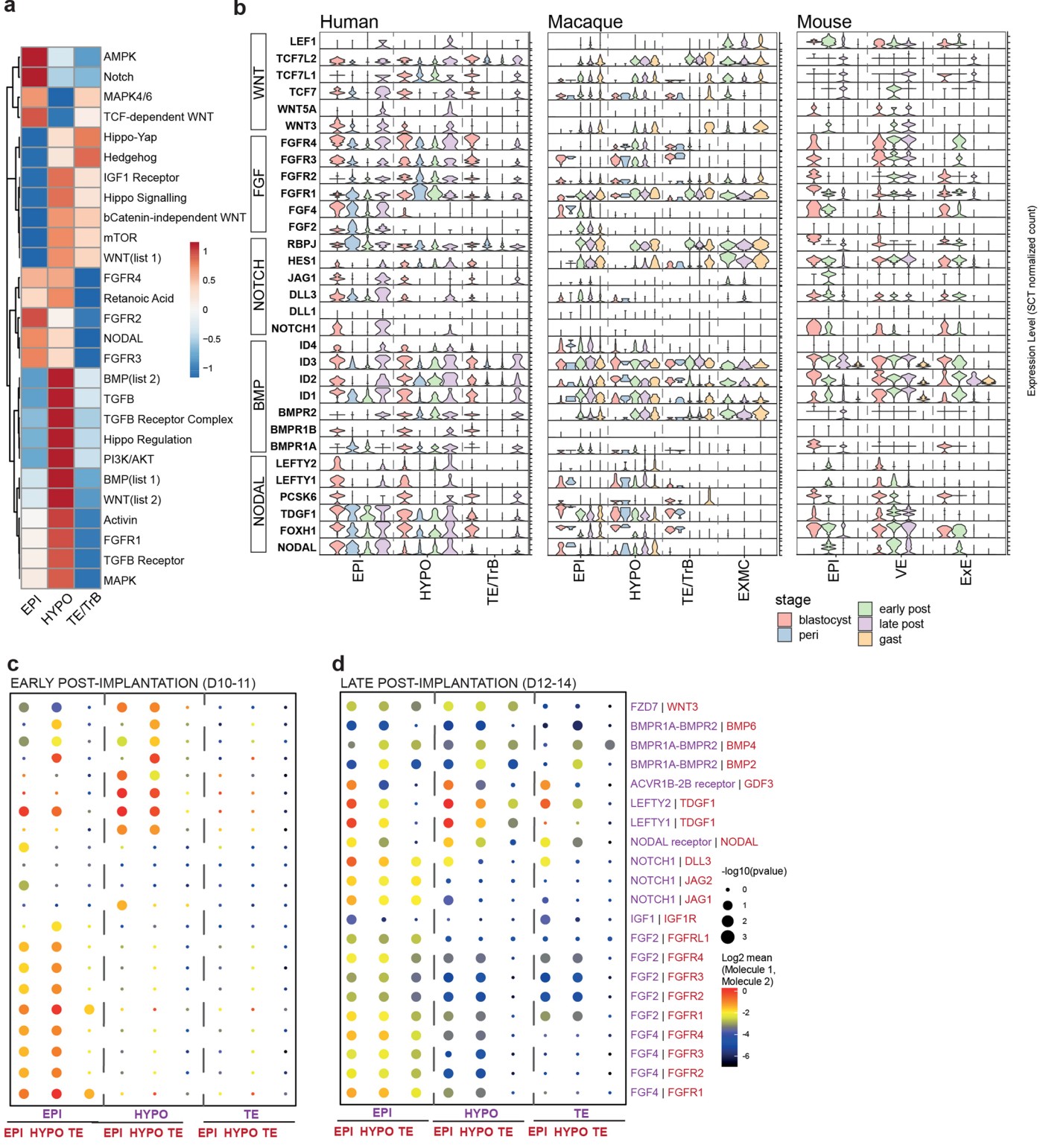

**Extended Data Fig. 3 | Signaling dynamics during human implantation.**
**a**, Average scaled WikiPathways module scores for combined human dataset between lineages. **b**, Expression of WNT, FGF, BMP and NODAL associated genes plotted by stage during human, macaque, and mouse pre-gastrulation development. **c-d**, CellphoneDB dotplots for early post-implantation (**c**) and late post-implantation (**d**) stages depicting the predicted activity of individual receptor-ligand pairs. Each ligand-receptor pair is color matched to the expressing lineage on the x-axis.

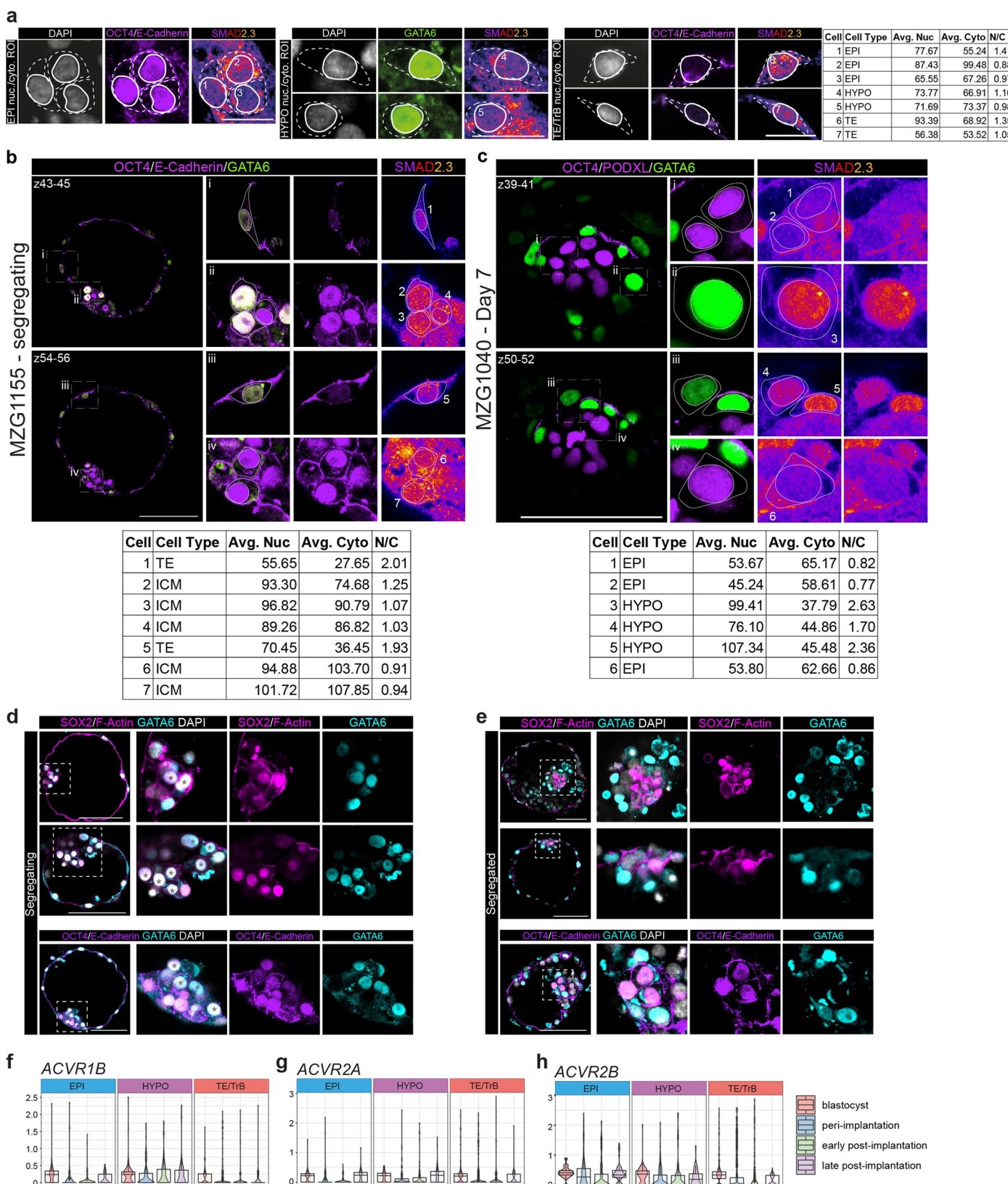

**Extended Data Fig. 4 | Total SMAD2.3 quantification methodology.**
**a**, Examples from a segregated blastocyst presented in Fig. 2a of Epiblast (EPI), Hypoblast (HYPO) and Trophectoderm (TE) cells with nuclear (solid line) and cytoplasmic (dotted line) regions of interest (ROI) outlined. The measured nuclear (Avg. Nuc) and cytoplasmic (Avg. Cyto) and calculated nuclear/cytoplasmic ratio (N/C) of total SMAD2.3 intensity for individual cells is shown. **b-c**, Examples from a segregating blastocyst (**b**) and day 7 embryo (**c**) of cells with nuclear and cytoplasmic ROIs outlined. Measured nuclear, cytoplasmic, and calculated nuclear-to-cytoplasmic ratios of total SMAD2.3 intensities are shown for each cell. Note that here the raw, unprocessed SMAD2.3 signal is shown for the central 3-plane z-stacks used directly for quantification. N = 14 experiments. **d-e**, Examples of segregating (**d**) and segregated (**e**) human blastocysts. Inner cell mass cells co-expressing epiblast and hypoblast markers are marked with grey asterisks. N = 3 experiments. **f-h**, Violin plots depicting expression of NODAL receptors *ACVR1B, ACVR2A*, and *ACVR2B* across stages and lineages in human single cell RNA sequencing data. N = 9862 single cells. Scale bars: 100 μm.

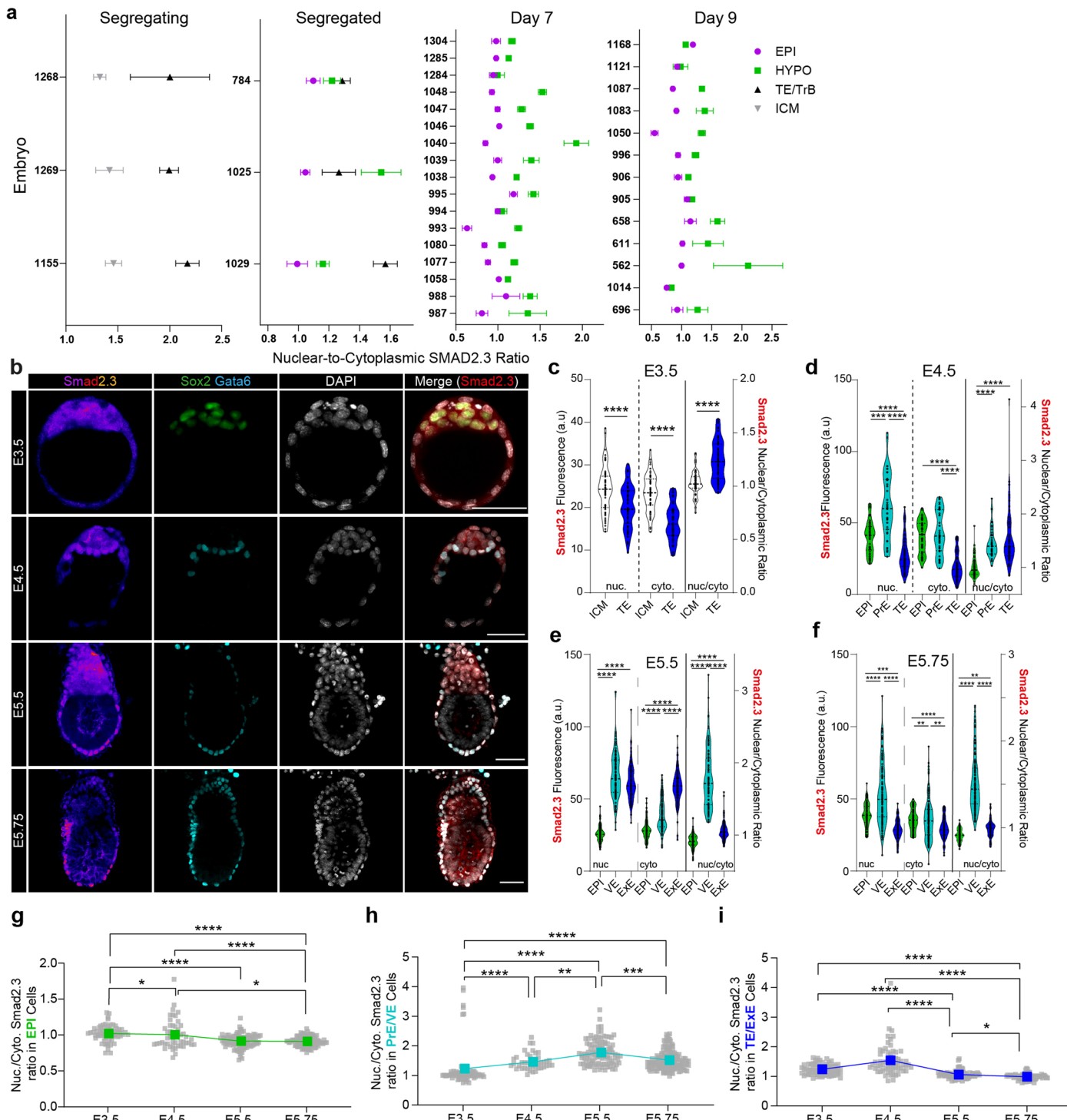

**Extended Data Fig. 5 | Characterization of NODAL signaling across mouse implantation. a**, Mean ± S.E.M. nuclear-to-cytoplasmic ratio of total SMAD2.3 within each lineage of segregating and segregated blastocysts, day 7, and day 9 embryos, separated by individual embryo. Note consistency of trends, particularly of TE versus ICM enrichment in segregating blastocysts, and of hypoblast versus epiblast. **b**, Immunofluorescence images of mouse embryos at E3.5 (N = 10 embryos), E4.5 (N = 6 embryos), E5.5 (N = 6 embryos) and E5.75 (N = 6 embryos) stained for total Smad2.3, DAPI, and either Sox2 or Gata6. **c-f**, Quantification of nuclear and cytoplasmic Smad2.3. fluorescence and their ratio. **g-i**, Quantification of the nuclear/cytoplasmic ratio of Smad2.3 within the epiblast (**g**; n = 63 E3.5, 50 E4.5, 95 E5.5, 95 E5.75 cells), primitive endoderm/visceral endoderm (PrE/VE; n = 69 E3.5, 42 E4.5, 105 E5.5, 129 E5.75 cells) (**h**) or trophectoderm/extraembryonic ectoderm (TE/ExE) (**i**; n = 62 E3.5, 60 E4.5, 89 E5.5, 69 E5.75 cells) over time. For Violin plots **c-f**, central dotted line denotes median and dotted lines mark the 25th and 75th quartiles. Statistical tests: (**c**) two-sided unpaired T-test; (**d-f**) Kruskal Wallis with Dunn's post-hoc; ****p < 0.0001, ***p < 0.001, *p < 0.05. Unmarked pairwise comparisons are not significant (ns). Exact p-values presented in Supplemental Table 8. Scale bars: 100 μm.

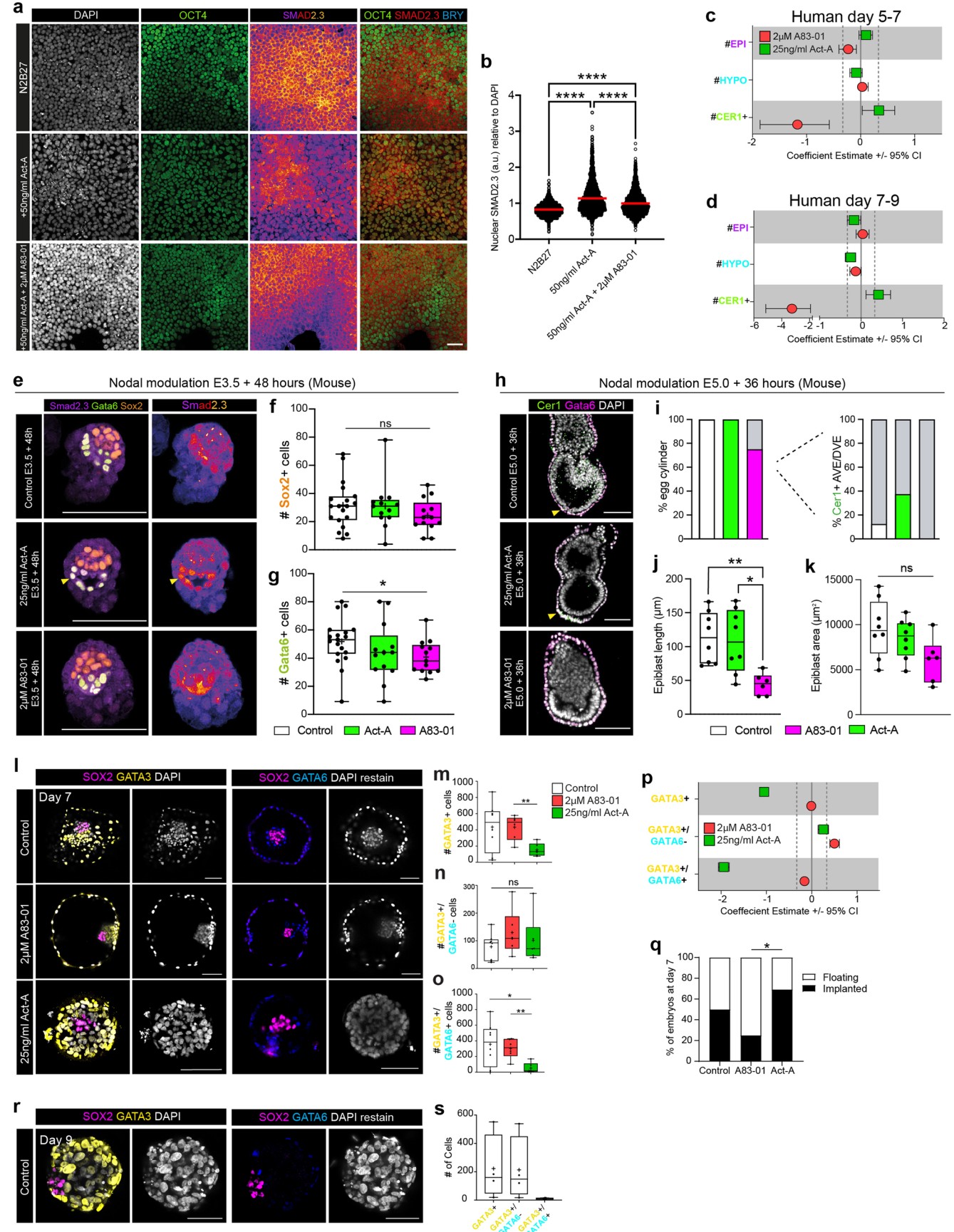

**Extended Data Fig. 6 | See next page for caption.**

**Extended Data Fig. 6 | NODAL modulation in human ESC, mouse embryos, and effect of NODAL modulation on human trophectoderm.**
**a-b**, Immunofluorescence images and quantification of primed hESCs cultured with either DMSO (n = 2588), 50 ng/ml Activin-A (n = 2465), or Activin-A with 2 µM A83-01 (n = 2463) for 48 hours. **c-d**, Bayesian coefficients ± credible interval for counts presented in Fig. 3c–e (**c**) and Fig. 3h–j. (**d**). **e**, Immunofluorescence images of mouse embryos recovered at E3.5 and cultured *in vitro* for 48 h in control (N = 19), 25 ng/ml Activin-A (N = 13 embryos) or 2 µM A83-01 (N = 13 embryos) conditions. **f-g**, Quantification of the number of epiblast (**f**) and primitive endoderm/visceral endoderm (**g**) from **e**. **h**, Immunofluorescence images of mouse embryos recovered at E5.0 and cultured *in vitro* for 36 h in control (N = 8), Activin-A (N = 8), or A83-01 (N = 6) conditions. **i**, Quantification of the proportion of embryos that successfully made an egg cylinder and the proportion of egg cylinders with a Cer1-positive DVE/AVE. **j**, Quantification of epiblast length. **k**, Quantification of epiblast area. **l**, Immunofluorescence images of embryos cultured from day 5 to 7 in control (N = 8 embryos), 2 µM A83-01 (N = 6 embryos), or 25 ng/ml Activin-A (N = 6 embryos) conditions.

**m-o**, Quantification of the number of total GATA3-positive (GATA3 + ) (**m**), GATA3 + /GATA6-negative (-) (**n**), and GATA3 + /GATA6+ trophectoderm/trophoblast cells (**o**) at day 7. **p**, Bayesian coefficients ± credible interval for counts presented in **m-o**. (N = 8 control, 8 Activin-A, and 6 A83-01 treated embryos). **q**, Quantification of the percentage of embryos cultured from day 5 to 7 in control conditions (N = 16), A83-01 (N = 12), or Activin-A (N = 13) that underwent *in vitro* implantation. **r**, Immunofluorescence images of a day 9 embryo cultured in control conditions (N = 4 embryos). **s**, Quantification of the number of total GATA3 + , GATA3 + /GATA6-, and GATA3 + /GATA6+ cells at day 9. For box plots, the box encompasses the 25th-75th quartiles with the median marked by the central line. The minimum and maximum are denoted by the whiskers. For violin plots, the mean is marked by the red line. Statistical tests: (**b, j-k**) One-way ANOVA with Tukey-Kramer post-hoc; (**f-g, m-o**) two-sided Mann-Whitney test; (**q**) Fisher's Exact Test. ****p < 0.0001. **p < 0.01, *p < 0.05. Unmarked pairwise comparisons are not significant (ns). Exact p-values presented in Supplemental Table 8. Scale bars: (**a**) 50 µm, (**e, h, l, r**) 100 µm.

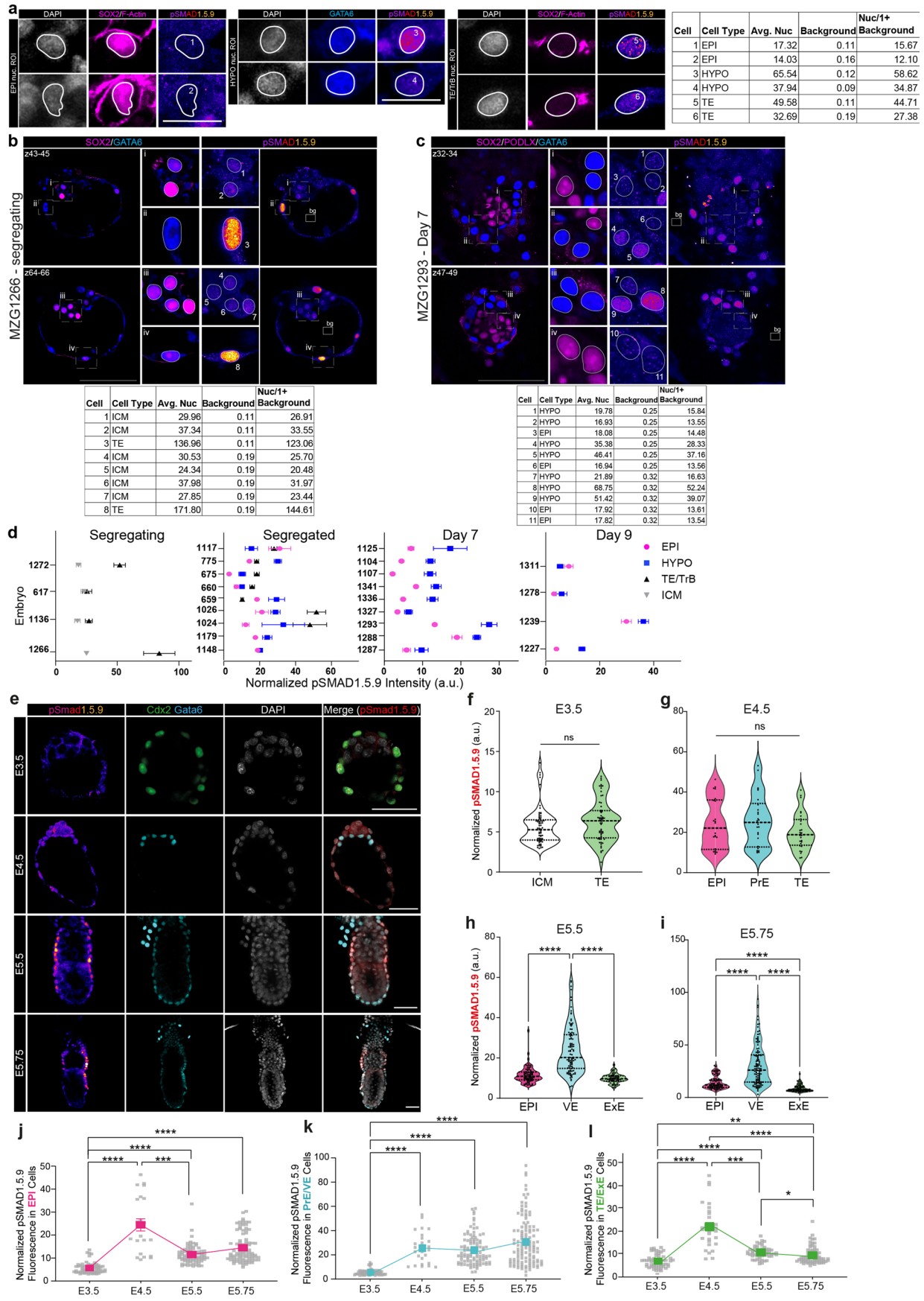

**Extended Data Fig. 7 | See next page for caption.**

**Extended Data Fig. 7 | Comparative characterization of BMP signaling across mouse implantation. a**, Examples from a segregated blastocyst presented in Fig. 4a of Epiblast (EPI), Hypoblast (HYPO) and Trophectoderm (TE) cells with nuclear region of interest (ROI) outlined. The measured nuclear (Avg. Nuc) and background and calculated normalized (Nuc/1+Background) pSMAD1.5.9 intensity for individual cells is shown. **b-c**, Examples from a segregating blastocyst (**b**) and day 7 embryo (**c**) of cells with nuclear and background ROIs outlined. Measured nuclear and background (bg) and calculated normalized pSMAD1.5.9 intensities are shown for each cell. Note that here the raw, unprocessed pSMAD1.5.9 signal is shown for the central 3-plane z-stacks used directly for quantification. Cells that are central to the same 3-plane z-stack use the same background value for normalization. **d**, Mean ± S.E.M. normalized pSMAD1.5.9 intensity within each lineage of segregating (N = 9) and segregated blastocysts (N = 9), day 7 (N = 6), and day 9 embryos, separated by individual embryo. Note consistency of trends, particularly of TE versus ICM enrichment

in segregating blastocysts, and of hypoblast versus epiblast in segregated blastocysts and at day 7. **e**, Immunofluorescence images of mouse embryos at E3.5 (N = 9 embryos), E4.5 (N = 4 embryos), E5.5 (N = 6 embryos) and E5.75 (N = 5 embryos) stained for phosphorylated (p)Smad1.5.9, DAPI and either Cdx2 or Gata6. **f-i**, Quantification of normalized pSmad1.5.9 fluorescence between lineages at the different stages of mouse development depicted in **e**. **j-l**, Quantification of normalized pSmad1.5.9 in epiblast (EPI) (**j**; n = 69 E3.5, 22 E4.5, 82 E5.5, 93 E5.75 cells), primitive endoderm/visceral endoderm (PrE/VE) (**k**; n = 69 E3.5, 26 E4.5, 102 E5.5, and 121 E5.75 cells) (**j**) or trophectoderm/extraembryonic ectoderm (TE/ExE) (**l**; n = 60 E3.5, 28 E4.5, 62 E5.5, and 69 E5.75 cells) across stages. For Violin plots **e-h**, central dotted line denotes median and dotted lines mark the 25th and 75th quartiles. Statistical tests: (**f**) two-sided Mann Whitney, (**g-i**) Kruskal Wallis with Dunn's post-hoc; ****p < 0.0001, ***p < 0.001, *p < 0.05. Unmarked pairwise comparisons are not significant (ns). Exact p-values presented in Supplemental Table 8. Scale bars: 100 μm.

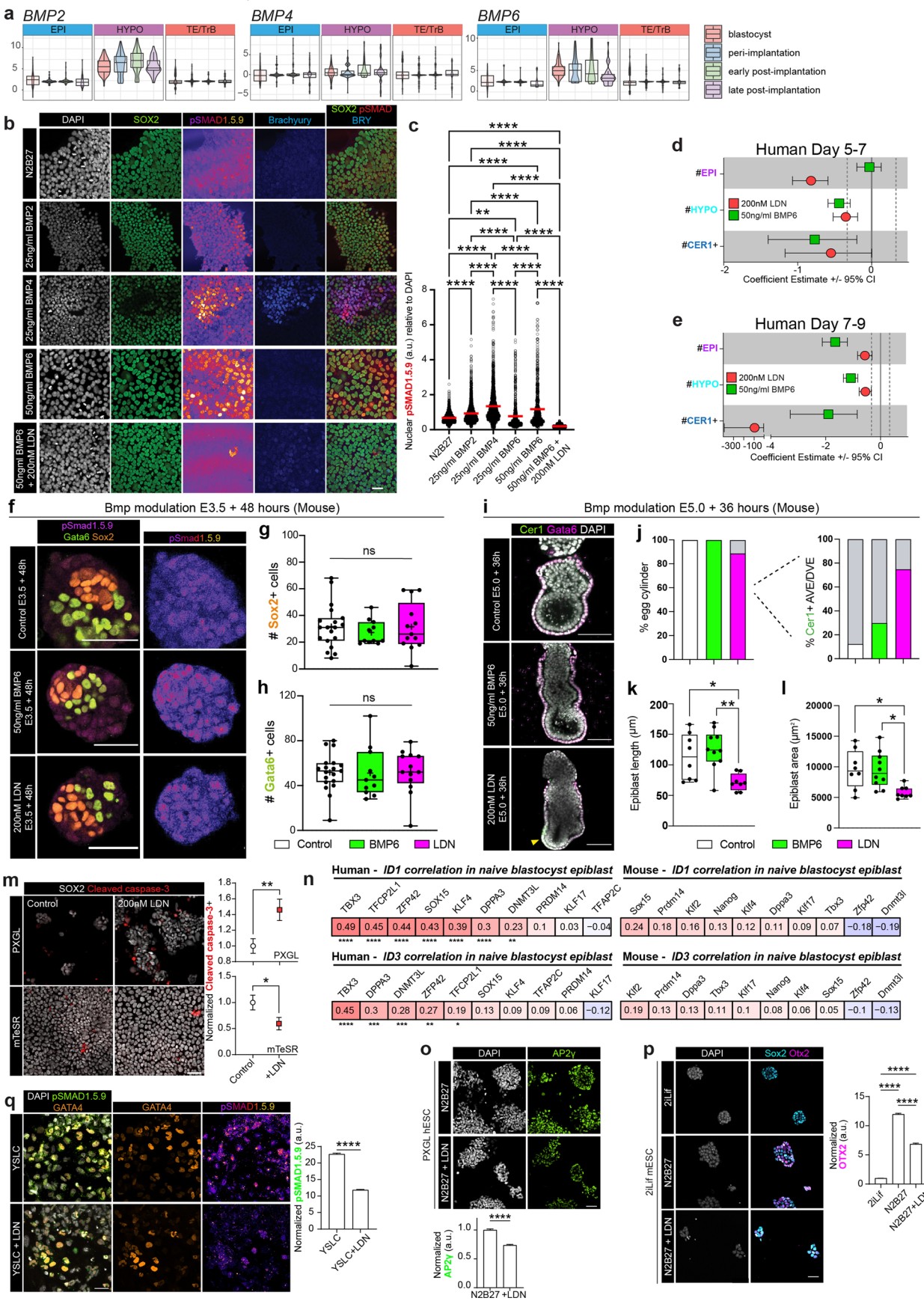

**Extended Data Fig. 8 | See next page for caption.**

**Extended Data Fig. 8 | Validation of BMP modulation, comparative functional modulation in mouse, and divergent roles for BMP in naïve pluripotency. a**, Violin plots of *BMP2/4/6* expression. N = 9862 single cells. **b-c**, Immunofluorescence images and quantification of primed hESCs cultured in control conditions (n = 2825), 25 ng/ml BMP2 (n = 2184), 25 ng/ml BMP4 (n = 1681), 50 ng/ml BMP6 (n = 889), or 50 ng/ml BMP6 + 200 nM LDN193189 (n = 1406). **d-e**, Bayesian coefficients ± credible interval for counts presented in Fig. 5b-d (**d**) and Fig. 5f-h (**e**). **f**, Immunofluorescence images of mouse embryos recovered at E3.5 and cultured *in vitro* for 48 h in control (N = 19), 50 ng/ml BMP6 (N = 11), or 200 nM LDN193189 (N = 13) conditions. **g-h**, Quantification of the number epiblast (**g**) and primitive endoderm/visceral endoderm (**h**) from **f**. **i**, Immunofluorescence images of mouse embryos recovered at E5.0 and cultured *in vitro* for 36 h in control (N = 8), BMP6 (N = 10) or LDN193189 (N = 8) conditions. **j**, Quantification of the proportion of embryos that successfully made an egg cylinder and have a Cer1-positive DVE/AVE. **k-l**, Quantification of the epiblast length (**k**) and area (**l**). **m**, Immunofluorescence images and quantification of hESCs treated for 48 hours with control conditions (PXGL n = 2336, primed n = 7796) or 200 nM LDN193189 (PXGL n = 2398, primed

n = 6500, N = 2 experiments). **n**, Pearson regression correlation coefficients for *ID1* and *ID3* expression with naïve pluripotency markers. **o**, Immunofluorescence and quantification of AP2γ in hESCs (LDN: n = 1695 untreated cells; n = 1426 LDN treated cells; N = 2 experiments). **p**, Immunofluorescence and quantification of Otx2 in mESCs cultured in either 2iLif naïve conditions (n = 1570 cells), N2B27 (n = 3602 cells), or N2B27 + LDN (n = 1064 cells). N = 3 experiments. **q**, Immunofluorescence images and quantification of pSMAD1.5.9 in YSLCs (control: n = 1150; LDN: n = 827; N = 2 experiments). For box plots, the box encompasses the 25th-75th quartiles with the median marked by the central line. The minimum and maximum are denoted by the whiskers. For violin plots, the mean is marked by the red line. Statistical tests: (**c, k-l, p**) One-way ANOVA with Tukey-Kramer post-hoc; (**e-f**) two-sided Mann-Whitney test, (**m, o, q**) two-sided unpaired T-test; (**o**) Coefficients were tested (cor.test; two-tailed) and corrected for multiple hypothesis testing with the Benjamini-Hochberg method; ****p < 0.0001. ***p < 0.001. **p < 0.01. *p < 0.05. Unmarked pairwise comparisons are not significant (ns). Exact p-values presented in Supplemental Table 8. Scale bars: (**b, m, o-q**) 50 μm, (**f, i**) 100 μm.

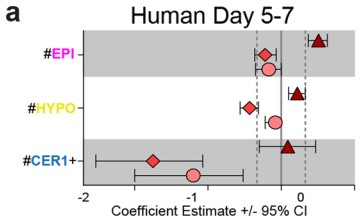

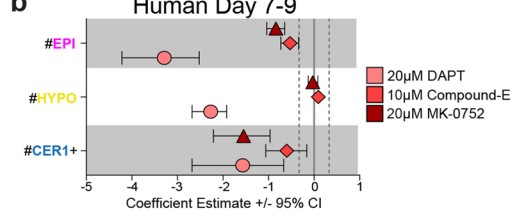

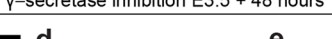

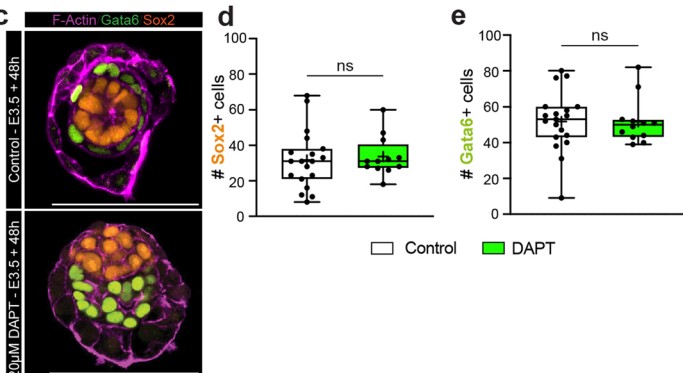

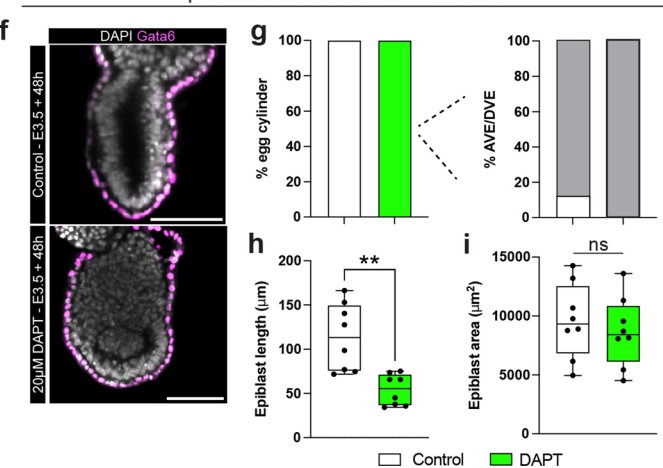

**Extended Data Fig. 9 | Comparative functional Notch interrogation of mouse. a-b**, Bayesian coefficients ± credible interval for counts presented in Fig. 6b–d (**a**) and Fig. 6f–h (**b**). **c**, Immunofluorescence images of representative mouse embryo recovered at E3.5 and cultured *in vitro* for 48 hours in either control conditions (N = 19 embryos) or with 20 µM DAPT added to the media (N = 12 embryos). **d-e**, Quantification of number of epiblast (**d**) and primitive endoderm/ visceral endoderm (**e**) from **c**. **f**, Immunofluorescence images of a representative mouse embryo recovered at E5.0 and cultured *in vitro* for 36 h in either control conditions (N = 8 embryos) with 20 µM DAPT added to the media (N = 8

embryos). **g**, Quantification of the proportion of embryos that successfully made an egg cylinder and of the proportion of egg cylinders that had a Cer1-positive DVE or AVE. **h**, Quantification of epiblast length in µm at the central plane. **i**, Quantification of epiblast area in µm² at the central plane. For box plots, the box encompasses the 25th-75th quartiles with the median marked by the central line. The minimum and maximum are denoted by the whiskers. Statistical tests: (**d-e**) Two-sided Mann-Whitney test; (**h-i**) two-sided unpaired T-test; **p < 0.01. Exact p-values presented in Supplemental Table 8. Scale bars = 100 µm.

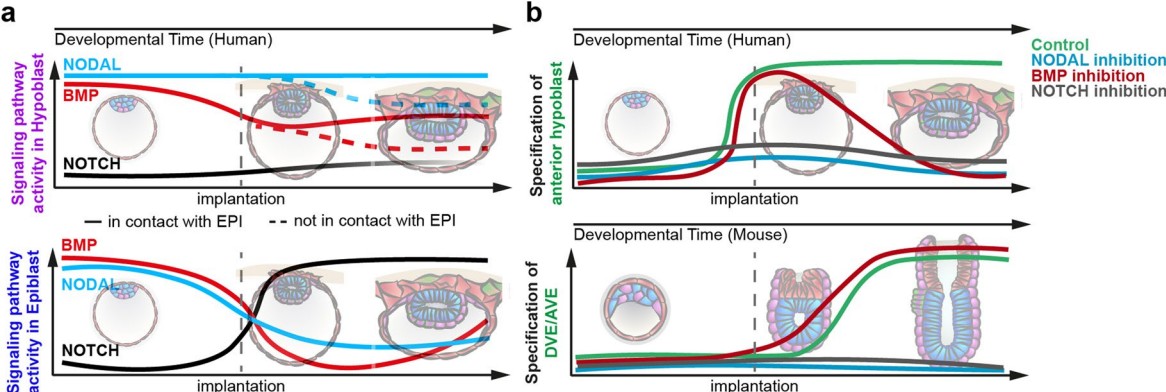

**Extended Data Fig. 10 | Proposed model of peri-implantation signaling activity and anterior specification. a**, Proposed model of BMP, NODAL and NOTCH signaling activity in the Hypoblast (top) and Epiblast (bottom) during human peri-implantation. Note implantation as a switch point in the activity of pathways for the epiblast, and a point of divergence of subpopulations within the hypoblast. The role of NOTCH in the hypoblast remains less clear, indicated by a gradient. **b**, Summary of signaling perturbation effects on human anterior hypoblast (top) and mouse distal/anterior visceral endoderm (DVE/AVE; bottom) specification. Schematics of human embryos correspond to day 5, day 7, and day 9 and mouse embryo schematics correspond to E3.5, E5.0, E5.75.

# Reporting Summary

## Statistics

For all statistical analyses, confirm that the following items are present in the figure legend, table legend, main text, or Methods section.

| n/a | Confirmed | |
|---|---|---|
| ☐ | ☒ | The exact sample size (*n*) for each experimental group/condition, given as a discrete number and unit of measurement |
| ☐ | ☒ | A statement on whether measurements were taken from distinct samples or whether the same sample was measured repeatedly |
| ☐ | ☒ | The statistical test(s) used AND whether they are one- or two-sided *Only common tests should be described solely by name; describe more complex techniques in the Methods section.* |
| ☒ | ☐ | A description of all covariates tested |
| ☐ | ☒ | A description of any assumptions or corrections, such as tests of normality and adjustment for multiple comparisons |
| ☐ | ☒ | A full description of the statistical parameters including central tendency (e.g. means) or other basic estimates (e.g. regression coefficient) AND variation (e.g. standard deviation) or associated estimates of uncertainty (e.g. confidence intervals) |
| ☐ | ☒ | For null hypothesis testing, the test statistic (e.g. *F*, *t*, *r*) with confidence intervals, effect sizes, degrees of freedom and *P* value noted *Give P values as exact values whenever suitable.* |
| ☐ | ☒ | For Bayesian analysis, information on the choice of priors and Markov chain Monte Carlo settings |
| ☒ | ☐ | For hierarchical and complex designs, identification of the appropriate level for tests and full reporting of outcomes |
| ☐ | ☒ | Estimates of effect sizes (e.g. Cohen's *d*, Pearson's *r*), indicating how they were calculated |

*Our web collection on statistics for biologists contains articles on many of the points above.*

## Software and code

Policy information about availability of computer code

| Data collection | wget (1.21.2) was used to download publicly available data |
|---|---|
| Data analysis | Code to used to analyze these data is available at https://github.com/bweatherbee/PeriImplantation and combined sequencing atlas objects are available on Zenodo at https://doi.org/10.5281/zenodo.7689580.<br>List of software used -<br>command line tools - kallisto, kb-python, fastQC, cutadapt<br>R - R (4.1.2); Seurat (4.2.0), ggplot2 (3.3.6), Scillus (0.5.0), RColorBrewer(1.1-3), data.table (1.14.2), scmap (1.16.0), biomaRt (2.50.3), SingleCellExperiment (1.16.0), tidyverse (1.3.2), org.Hs.eg.db (3.14.0), rWikiPathways (1.14.0), brms (2.18.0)<br>Python - CellPhoneDB (2.1.7), python (3.8), matplotlib (3.3.0), anndata (0.7.5), scanpy (1.5.1), numpy (1.17.5)<br>GraphPad Prism (9.3)<br>Fiji (2.14.0) |

For manuscripts utilizing custom algorithms or software that are central to the research but not yet described in published literature, software must be made available to editors and reviewers. We strongly encourage code deposition in a community repository (e.g. GitHub). See the Nature Portfolio guidelines for submitting code & software for further information.

# Data

Policy information about availability of data

All manuscripts must include a data availability statement. This statement should provide the following information, where applicable:

- Accession codes, unique identifiers, or web links for publicly available datasets
- A description of any restrictions on data availability
- For clinical datasets or third party data, please ensure that the statement adheres to our policy

All raw data used here is previously published and publicly available.

For aligning sequencing data, GRCh38 (https://www.ncbi.nlm.nih.gov/assembly/GCF_000001405.26/),  Genome assembly Macaca_fascicularis_5.0 (https://www.ncbi.nlm.nih.gov/datasets/genome/GCF_000364345.1/), and GRCm39 (https://www.ncbi.nlm.nih.gov/datasets/genome/GCF_000001635.27/) were used.

Human data
Molè et al., 2021: ArrayExpress E-MTAB-8060
Xiang et al., 2020: Gene Expression Omnibus GSE136447
Zhou et al., 2019: Gene Expression Omnibus GSE109555
Petropoulos et al., 2016: ArrayExpress E-MTAB-3929
Blakely et al., 2015: Gene Expression Omnibus GSE66507

Cynomolgus Monkey
Yang et al., 2021: Gene Expression Omnibus GSE148683
Ma et al., 2019: Gene Expression Omnibus GSE130114
Nakamura et al., 2016: Gene Expression Omnibus GSE74767

Mouse
Pijuan-Sala et al., 2019: ArrayExpress E-MTAB-6967
Mohammed et al., 2017: Gene Expression Omnibus GSE100597
Cheng et al., 2019: Gene Expression Omnibus GSE109071
Deng et al., 2014: Gene Expression Omnibus GSE45719

# Human research participants

Policy information about studies involving human research participants and Sex and Gender in Research.

| | |
|---|---|
| Reporting on sex and gender | We do not have access to prenatal genetic testing for the vast majority of embryos. Therefore, the composition of sex chromosomes of embryos cultured in the lab is largely unknown. |
| Population characteristics | According to the United Kingdom's Human Fertilisation and Embryology Act, which governs human embryo research, identifiable information of parents donating embryos to research is redacted. Therefore population characteristics of donating patients and their embryos is unknown. |
| Recruitment | Human embryos are donated by patients in the UK from collaborating IVF clinics under HFEA licence R0193. Patients undergoing IVF at CARE Fertility, Bourn Hall Fertility Clinic, Herts & Essex Fertility Clinic, and King's Fertility was given the option of continued storage, disposal, or donation of embryos to research (including project specific information) or training at the end of their treatment. Patients were offered counseling, received no financial benefit, and could withdraw their participation at any time until the embryo had been used for research. <br><br> All information of patients is required to be redacted prior to donation to research. Therefore, potential biases based on recruitment is unknown. Please note this manuscript does not perform any experimentation on human embryos, rather we seek to provide a single example of an embryo cultured in vitro as a reference image for the natural post-implantation embryo. |
| Ethics oversight | Ethical oversight is provided both by the HFEA and the Human Biological Research Ethics Committee at the University of Cambridge. The recruitment of patients to donate human embryos to research follows the Human Fertilisation and Embryology Authority's guidelines. This includes the provision of project-specific information, the offering of counseling, and the ability to withdraw consent at any time until the embryos have been used. Stem cell work is approved by the UK Stem Cell Bank. |

Note that full information on the approval of the study protocol must also be provided in the manuscript.

# Field-specific reporting

Please select the one below that is the best fit for your research. If you are not sure, read the appropriate sections before making your selection.

☒ Life sciences          ☐ Behavioural & social sciences          ☐ Ecological, evolutionary & environmental sciences

# Life sciences study design

All studies must disclose on these points even when the disclosure is negative.

| | |
|---|---|
| Sample size | No tests were used to predetermine sample size. Sample sizes for experimentation was determined based on our previous experience with human embryos (Shahbazi et al., 2016, Mole et al., 2021) and human stem cells and 3D stem cell models (Shahbazi et al., 2017, Mackinlay et al., 2021, Weatherbee et al., 2023). |
| Data exclusions | For embryos assessed for normal development (Figures 2 and 3), only embryos that contained all three major lineages were included. For small molecule perturbation experiments, all embryos were included in all groups.<br>For single cell RNA-seq data, only those cells included for downstream analysis in the original publications were included in this study. |
| Replication | All experiments were performed at least twice, with a minimum of 5 embryos from 3 patients/crosses. Human embryo experiments were performed by 4 authors over time and mouse experiments by 2. |
| Randomization | Allocation was not performed randomly into groups for any experiments. For human and mouse embryo experiments, based on visual assessment of embryos, investigators attempted to ensure balanced distributions of blastocysts/implanting embryos assessed as expanded with nice inner cell masses versus embryos that appeared delayed or with visible cell death across experimental groups. For cell culture experiments, microwells with many cells or spheroids were all treated and quantified, and randomization/allocation of individuals would have been impossible. |
| Blinding | Investigators were not blinded to experimental groups in any of the human embryo, mouse embryo or stem cell experiments. It would not be feasible to blind the media changes with the addition of small molecules as we made media in-house. |

# Reporting for specific materials, systems and methods

We require information from authors about some types of materials, experimental systems and methods used in many studies. Here, indicate whether each material, system or method listed is relevant to your study. If you are not sure if a list item applies to your research, read the appropriate section before selecting a response.

## Materials & experimental systems

| n/a | Involved in the study |
|---|---|
| ☐ | ☒ Antibodies |
| ☐ | ☒ Eukaryotic cell lines |
| ☒ | ☐ Palaeontology and archaeology |
| ☐ | ☒ Animals and other organisms |
| ☒ | ☐ Clinical data |
| ☒ | ☐ Dual use research of concern |

## Methods

| n/a | Involved in the study |
|---|---|
| ☒ | ☐ ChIP-seq |
| ☒ | ☐ Flow cytometry |
| ☒ | ☐ MRI-based neuroimaging |

## Antibodies

| | |
|---|---|
| Antibodies used | mouse monoclonal anti OCT3/4 (sc5279, Santa Cruz; clone C-10; 1:200 dilution), rat monoclonal anti SOX2 (14-19811-82, Thermo Fisher Scientific; clone Btjce; 1:500 dilution), goat polyclonal anti NANOG (AF1997 R&D Systems; 1:500 dilution), rabbit monoclonal anti GATA6 (5851, clone D61E4; Cell Signaling Technology; 1:2000 dilution), goat polyclonal anti GATA6 (AF1700, R&D Systems; 1:200 dilution), mouse anti monoclonal Cdx2 (MU392-UC, Biogenex; clone CDX2-88; 1:200 dilution), goat polyclonal anti CER1 (AF1075, R&D Systems; 1:250 dilution), rat monoclonal anti Cerebus1 (MAB1986, R&D Systems; clone 225807; 1:200 dilution), rabbit monoclonal anti Phospho-Smad1(Ser463/465)/Smad5(Ser463/465)/Smad9(Ser465/467) (13820T, Cell Signaling Technology; clone D5B10; 1:200 dilution), rabbit monoclonal anti Smad2.3 (8685T, Cell Signaling Technology; clone D7G7; 1:200 dilution), Rabbit monoclonal anti Cleaved Caspase 3 (9664, Cell Signaling Technology; clone 5A1E; 1:200 dilution), mouse monoclonal anti Podocalyxin (MAB1658, R&D Systems; clone 222328; 1:500 dilution), goat polyclonal anti Brachyury (AF2085, R&D Systems; 1:500 dilution), rat monoclonal anti GATA4 (14-9980-82, Thermo Fisher Scientific; clone eBioEvan; 1:500 dilution), goat polyclonal anti AP2-gamma (AF5059, R&D Systems; 1:500 dilution), goat polyclonal anti Otx2 (AF1979, R&D Systems; 1:1000 dilution), Alexa Flour 594 Phalloidin (A12381, Thermo Fisher Scientific; 1:500 dilution). |
| Validation | All antibodies are validated according to supplier's websites. Details of the validation statement, antibody profiles and relevant citations can be found on the manufacturer's website. In addition to that, all antibodies in this study showed expected staining patterns based on protein type (e.g. transcription factors in the nucleus, membrane-bound proteins at the membrane) in human embryonic stem cells. |

# Eukaryotic cell lines

Policy information about cell lines and Sex and Gender in Research

| | |
|---|---|
| Cell line source(s) | UK Stem Cell Bank (Shef6), Prof Jennifer Nichols (Stem Cell Institute, University of Cambridge, UK) (mouse CD1 ESCs) |
| Authentication | The UK Stem Cell Bank validates deposited cell lines by STR analysis. mESCs were validated in-house by STR analysis. |
| Mycoplasma contamination | Cells were tested regularly for mycoplasma, and were negative. |
| Commonly misidentified lines (See ICLAC register) | No commonly misidentified lines were used in this study. |

# Animals and other research organisms

Policy information about studies involving animals; ARRIVE guidelines recommended for reporting animal research, and Sex and Gender in Research

| | |
|---|---|
| Laboratory animals | CD1 wildtype males aged 6 to 45 weeks and CD1 wildtype females aged 6 to 18 weeks were used for this study. Mice were kept in an animal house in individually ventilated housing on 12:12 hour light-dark cycle with ad libitum access to food and water. Ambient temperature was maintained at 21-22ºC and humidity at 50%. |
| Wild animals | The study did not involve wild animals |
| Reporting on sex | Sex was not considered in this study as embryos were recovered from the mother and used for experimentation at the 8-cell stage. Genotyping was not performed. |
| Field-collected samples | Field-collected samples were not used in this study. |
| Ethics oversight | Mice were kept in an animal house on 12:12 hour light dark cycle with ad libitum access to food and water. Experiments with mice are regulated by the Animals (Scientific Procedures) Act 1986 Amendment Regulations 2012 and carried out following ethical review by the University of Cambridge Animal Welfare and Ethical Review Body (AWERB). Experiments were approved by the Home Office under Licenses 70/8864 and PP3370287. Animals were inspected daily and those showing health concerns were culled by cervical dislocation. |

Note that full information on the approval of the study protocol must also be provided in the manuscript.

