## [Peer Review File · Nature Cell Biology]

Peer Review Information

Journal: Nature Cell Biology

Manuscript Title: Distinct pathways drive anterior hypoblast specification in the implanting human embryo

Corresponding author name(s): Professor Magdalena Zernicka-Goetz

Editorial Notes:

EA delete any non-applicable rows, and then delete this instruction.

Reviewer Comments & Decisions:

Decision Letter, initial version:

Dear Magda,

Your manuscript, "Distinct mechanisms drive anterior-posterior axis specification in the implanting human embryo", has now been seen by 4 referees, who are experts in development, human/cynomolgus monkey/mouse, scRNAseq, BMP/NOTCH (referee 1); human/non-human primate/mouse early development, scRNAseq, integrative studies (referee 2); bioethics (referee 3); and mammalian development, scRNAseq, bmp/nodal (referee 4). As you will see from their comments (attached below) they find this work of potential interest, but have raised substantial concerns, which in our view would need to be addressed with considerable revisions before we can consider publication in Nature Cell Biology.

Nature Cell Biology editors discuss the referee reports in detail within the editorial team, including the chief editor, to identify key referee points that should be addressed with priority, and requests that are overruled as being beyond the scope of the current study. To guide the scope of the revisions, I have listed these points below. I should stress that the referees' concerns point to a premature dataset and these points would need to be addressed with experiments and data, and reconsideration of the study for this journal and re-engagement of referees would depend on strength of these revisions.

In particular, it would be essential to:

(A) Address the issues raised by referee #1 concerning the analysis and statistics approaches:

"I simply wonder whether the integrated scRNA-seq data is reliable enough and whether there is a need to integrate them. There are only 3 major clusters in the UMAP derived from the integrated datasets (Fig1E), while more clusters were observed in the original papers, for example, Xiang et al. (Xiang et al., 2019), suggesting that data integration by a strong normalization method (maybe too much artificial) reduces the resolution and masks key features in each dataset. Rather, the authors have already identified AVEs in their own scRNA-seq data reported in the previous report (Molè et al., 2021). I therefore wonder if it would be possible to obtain more meaningful results from the analysis of each single dataset, or with the re-analysis of the authors' previous dataset".

"While the analyses using wikipathway and CellPhonDB are useful, I think that simple expression data is also very informative for the reader's understanding. Could the authors provide information of expression levels of at least key genes using standard methods, such as heatmap or violin plot? In addition, the real values, such as log₂(UMI) values, rather than the scaled ones as shown in Fig. 5A and Fig. 6A, should be provided, because although it is easy to see the differences between samples in

the scaled values, it is not clear whether the expression levels themselves are biologically significant or not".

"In line #112, based on the expression of GATA6 and COL6A1, the authors claimed that extra-embryonic mesenchyme was not observed in in vitro cultured human embryos. These genes were reported as markers for the extra-embryonic mesenchyme, but they are broadly expressed and are indeed also found in the other cell types in monkey embryos (Zhai et al., 2022) (Nakamura et al., 2016) (Yang et al., 2021). So they may not be critical markers. On the other hand, Pham et al. recently reported the induction of mesenchymal cells from human naïve pluripotent stem cells with the identification of new markers, such as LUM, POSTM, VIM and DCM (Pham et al., 2022). I suggest the authors to examine the expression of these genes and confirm whether the extraembryonic mesenchyme is really absent in in vitro cultured human embryos".

"In line #245, the authors used a Bayesian statistical model for the cell count analysis. Is this method also reliable enough? I think it is important for the authors to provide some results of "control" analysis that shows the validity of the statistical method used".

(B) Ensure that all claims in the manuscript can be clearly depicted in the figures by improving your approaches and performing additional experiments, as requested in each case by:

Referee #1:

"...On the other hand, I found that some of the data shown in the figures are not really convincing".

"In the IF analysis, the authors differentiated the "Segregating" and "Segregated" blastocysts. However, according to Fig2B, I could not distinguish between these two and I wonder whether the authors classified the embryo stages in a proper manner. In the pSMAD-level analysis, I also wondered whether the authors defined the nucleus/cytoplasm in each cell in an appropriate manner. These issues are essential for the following quantification, statistical analysis and the author's statements, but unfortunately, I do not think they are convincing enough".

"In IF analysis in Fig4-6, it is not possible for me to distinguish CER1 expression".

Referee #4:

"The effects of specific signaling pathway inhibition on human embryos were mostly examined using IF. Some of them were however somewhat difficult to evaluate without proper quantification (see below for examples). Single-cell RNA sequencing could be one way to validate these results and conclusions, and further shed light on the underlying mechanism".

"To quantify the activity of NODAL, the authors used the n/c ratio of total SMAD2.3 in individual cells. This could be misleading, especially when both signals in the nucleus and cytoplasm are high, but the ratio is low. For example, the authors stated "In segregating blastocysts, the n/c SMAD2.3 ratio was significantly higher in trophectoderm cells than inner cell mass cells, similar to the E3.5 early mouse blastocyst (Figure 2B-C, Extended Figure 4D-E) ". However, I cannot see a higher n/c SMAD2.3 ratio

in trophectoderm cells from IF results. Moreover, the signal of SMAD2.3 seems higher in ICM/EPI/HYPO than that in TE. The authors should show the original signals in the nucleus and cytoplasm side by side".

"The authors stated that "pSMAD1.5.9 expression decreased over time in the human epiblast, with a distinct bimodal intensity pattern at D9". However, I did not see the bimodality of pSMAD1.5.9 in Figure 3E".

(C) Strengthen the mechanistic part of the study, as indicated by:

Referee #1:

"Although the authors identified the signaling pathways that could be important for the maintenance/specification of EPI/HYPO/AVES, they did not provide evidence on the mechanism for the AP axis formation and symmetry breaking in the human embryos with a disc-shaped early post-implantation structure".

"In all analyses using signaling agonists/antagonists, the authors only discussed the appearance of cells expressing some key genes. These are informative experiments, but in my opinion not sufficient to claim that the mechanisms for the symmetry breaking of the disc-shape human embryos is clarified".

Referee #4:

"The differences in the requirement of BMP4 in pre-implantation epiblast survival in human and mouse are intriguing. Can the authors find any possible mechanisms underlying such species-specific divergence based on the scRNA-seq data?"

(D) Address the issue of the potentially uneven quality of IVF embryos raised by referee #2:

"In this study, the authors mainly used left-over IVF embryo cultures to examine the perturbation of NODAL, BMP, and NOTCH signaling on epiblast and hypoblast development. However, their results may be complicated by the uneven quality of IVF embryos during in vitro cultures. The generation of human embryoids and gastruloids from well-characterized human stem cells has provided a well-controlled system to study the patterning of anterior-posterior axis (e.g. doi.org/10.1038/s41586-020-2383-9). The authors have previously generated embryoids and gastruloids. The authors shall test the effect of compounds affecting BMP, NODAL, and Notch signaling during the time course of embryoid and gastruloid formation. These new data would allow authors to make more convincing conclusions".

(E) Provide the additional information on ethics requested by referee #3:

"What is missing from the description is anything about how the particular donors were recruited and chosen; the information they were given about other options for embryo management (e.g. freezing

or disposal without donation to research); and whether the donors were the progenitors (ie, were the sperm and egg providers the same people as those making the decision to donate, or were anonymous or known gamete donors being used by a couple in which one or both partners could not provide a gamete)".

(F) All other referee concerns pertaining to strengthening existing data, providing controls, clarifications and textual changes, should also be addressed.

(G) Finally please pay close attention to our guidelines on statistical and methodological reporting (listed below) as failure to do so may delay the reconsideration of the revised manuscript. In particular please provide:

- a Supplementary Figure including unprocessed images of all gels/blots in the form of a multi-page pdf file. Please ensure that blots/gels are labeled and the sections presented in the figures are clearly indicated.
- a Supplementary Table including all numerical source data in Excel format, with data for different figures provided as different sheets within a single Excel file. The file should include source data giving rise to graphical representations and statistical descriptions in the paper and for all instances where the figures present representative experiments of multiple independent repeats, the source data of all repeats should be provided.

We would be happy to consider a revised manuscript that would satisfactorily address these points, unless a similar paper is published elsewhere, or is accepted for publication in Nature Cell Biology in the meantime.

- ensure that it conforms to our format instructions and publication policies (see below and <https://www.nature.com/nature/for-authors>).
- provide a point-by-point rebuttal to the full referee reports verbatim, as provided at the end of this letter.
- provide the completed Reporting Summary (found here <https://www.nature.com/documents/nr-reporting-summary.pdf>). This is essential for reconsideration of the manuscript will be available to editors and referees in the event of peer review. For more information see <http://www.nature.com/authors/policies/availability.html> or contact me.

When submitting the revised version of your manuscript, please pay close attention to our [href="https://www.nature.com/nature-portfolio/editorial-policies/image-integrity">Digital Image Integrity Guidelines](https://www.nature.com/nature-portfolio/editorial-policies/image-integrity). and to the following points below:

- that unprocessed scans are clearly labelled and match the gels and western blots presented in

figures.

-- that control panels for gels and western blots are appropriately described as loading on sample processing controls

-- all images in the paper are checked for duplication of panels and for splicing of gel lanes.

Nature Cell Biology is committed to improving transparency in authorship. As part of our efforts in this direction, we are now requesting that all authors identified as 'corresponding author' on published papers create and link their Open Researcher and Contributor Identifier (ORCID) with their account on the Manuscript Tracking System (MTS), prior to acceptance. ORCID helps the scientific community achieve unambiguous attribution of all scholarly contributions. You can create and link your ORCID from the home page of the MTS by clicking on 'Modify my Springer Nature account'. For more information please visit www.springernature.com/orcid.

This journal strongly supports public availability of data. Please place the data used in your paper into a public data repository, or alternatively, present the data as Supplementary Information. If data can only be shared on request, please explain why in your Data Availability Statement, and also in the correspondence with your editor. Please note that for some data types, deposition in a public repository is mandatory - more information on our data deposition policies and available repositories appears below.

[Redacted]

We would like to receive a revised submission within six months.

We hope that you will find our referees' comments, and editorial guidance helpful. Please do not hesitate to contact me if there is anything you would like to discuss.

Best wishes,

Stelios

Stylianos Lefkopoulos, PhD
He/him/his
Associate Editor
Nature Cell Biology
Springer Nature

Heidelberger Platz 3, 14197 Berlin, Germany

E-mail: stylianos.lefkopoulos@springernature.com

Twitter: @s_lefkopoulos

Reviewers' Comments:

Reviewer #1:

Remarks to the Author:

The manuscript by Weberling et al. explored the molecular mechanism for the specification of the anterior visceral endoderm (AVE) in human embryos. The authors generated integrated massive scRNA-seq datasets of peri-implantation embryos in humans, monkeys and mice, and found that NODAL, BMP and NOTCH pathways are potentially active in the epiblast (EPI) and hypoblast (HYPO) lineages. By using agonists/antagonists of these signaling pathways for culturing human embryos and stem cell-based differentiation models, they found that both NODAL and BMP signaling are required for the AVE maturation/maintenance, which is somewhat in contrast to the case in mice that require high NODAL/low BMP activity for the AVE specification. They also found that NOTCH activity is crucial for the survival of both EPI and HYPO in an early post-implantation period.

The authors reported previously that putative AVEs were identified in human embryos cultured in vitro (Molè et al., 2021), and this work is a follow-up study, aiming to understand the mechanism that determines the anterior-posterior (AP) axis in human embryos. On the other hand, I found that some of the data shown in the figures are not really convincing, and some of the statements proposed are inconsistent. Although the authors identified the signaling pathways that could be important for the maintenance/specification of EPI/HYPO/AVEs, they did not provide evidence on the mechanism for the AP axis formation and symmetry breaking in the human embryos with a disc-shaped early post-implantation structure. Below are my major concerns:

Major points:

1. I simply wonder whether the integrated scRNA-seq data is reliable enough and whether there is a need to integrate them. There are only 3 major clusters in the UMAP derived from the integrated datasets (Fig1E), while more clusters were observed in the original papers, for example, Xiang et al. (Xiang et al., 2019), suggesting that data integration by a strong normalization method (maybe too much artificial) reduces the resolution and masks key features in each dataset. Rather, the authors have already identified AVEs in their own scRNA-seq data reported in the previous report (Molè et al., 2021). I therefore wonder if it would be possible to obtain more meaningful results from the analysis of each single dataset, or with the re-analysis of the authors' previous dataset.

2. While the analyses using wikipathway and CellPhonDB are useful, I think that simple expression data is also very informative for the reader's understanding. Could the authors provide information of expression levels of at least key genes using standard methods, such as heatmap or violin plot? In addition, the real values, such as $\log_2(\text{UMI})$ values, rather than the scaled ones as shown in Fig. 5A and Fig. 6A, should be provided, because although it is easy to see the differences between samples in the scaled values, it is not clear whether the expression levels themselves are biologically significant or not.

3. In line #112, based on the expression of GATA6 and COL6A1, the authors claimed that extra-embryonic mesenchyme was not observed in in vitro cultured human embryos. These genes were reported as markers for the extra-embryonic mesenchyme, but they are broadly expressed and are indeed also found in the other cell types in monkey embryos (Zhai et al., 2022) (Nakamura et al., 2016) (Yang et al., 2021). So they may not be critical markers. On the other hand, Pham et al. recently reported the induction of mesenchymal cells from human naïve pluripotent stem cells with the identification of new markers, such as LUM, POSTM, VIM and DCM (Pham et al., 2022). I suggest the authors to examine the expression of these genes and confirm whether the extraembryonic mesenchyme is really absent in in vitro cultured human embryos.
4. The authors suggested that the main source of BMP ligands is HYPO, but amniotic ectoderm also expresses BMPs in monkey embryo (Sasaki et al., 2016) and human in vitro amnion models (Zheng et al., 2019; Zheng et al., 2022). The authors should discuss the implications of the amniotic ectoderm.
5. In line #245, the authors used a Bayesian statistical model for the cell count analysis. Is this method also reliable enough? I think it is important for the authors to provide some results of “control” analysis that shows the validity of the statistical method used.
6. As commented in the point 2 above, the expression levels of key genes should be shown in real values, rather than scaled values (Fig. 5A, Fig. 6A, ExFig. 3B, ExFig. 4A-C, ExFig. 5A, B, ExFig. 7C).
7. In the IF analysis, the authors differentiated the “Segregating” and “Segregated” blastocysts. However, according to Fig2B, I could not distinguish between these two and I wonder whether the authors classified the embryo stages in a proper manner. In the pSMAD-level analysis, I also wondered whether the authors defined the nucleus/cytoplasm in each cell in an appropriate manner. These issues are essential for the following quantification, statistical analysis and the author’s statements, but unfortunately, I do not think they are convincing enough.
8. In line #188, the authors stated that the SMAD2/3 activity seems high in trophoctoderm cells. However, several reports about human trophoctoderm induction from naïve pluripotent stem cells consistently show that activin/nodal inhibition (by A83-01) is required for trophoctoderm differentiation. So, it sounds controversial. Could the authors elaborate this point further?
9. In IF analysis in Fig4-6, it is not possible for me to distinguish CER1 expression.
10. Similar to the point 8 above, did the A83-01 treated embryos increase the number of trophoctoderm cells? The authors simply examined the number of POU5F1/GATA6 positive cells (EPI/HYPO) as the effect of the A83-01 treatment of human embryos. However, A83-01 enhances the naïve pluripotent stem cells to differentiate to trophoctoderm, so the authors should also include and discuss the effect on trophoctoderm.
11. This might be an issue for the previous paper about yolk sac-like cell (YSLC) induction (Mackinlay et al., 2021); although the YSLC is not the definitive endoderm, the cells in the RseT medium that was established according to Gafni et al. (Gafni et al., 2013) must express some gastrulating genes such as T (Irie et al., 2015), and their transcriptome is corresponding to monkey post-implantation epiblast (Nakamura et al., 2016), suggesting they are neither so-called “naïve” nor “formative”. If YSLCs are “yolk sac” cells, what differentiation pathway do they follow? I wonder whether it would be appropriate to use YSLCs for the AVE analysis.

12. Fig5M-P shows that LDN treatment decreased the number of GATA6- and CER1-positive cells during the YSLC induction, and the authors suggested that the activation of BMP pathway is required for the VE/AVE differentiation. On the other hand, Mackinlay et al. reported that YSLCs themselves express BMP antagonists (Mackinlay et al., 2021). How much of the downstream SMADs phosphorylated in YSLCs under the control and LDN-treated conditions?

13. In the naive pluripotent stem cells in the PXGL media, the LDN treatment increased cleaved-Cas3-positive, i.e., apoptotic, cells. However, according to a previous report (Io et al., 2021), the SMAD1/5/8 of hPSCs in PXGL media are not phosphorylated. This may suggest that the effect of LDN may be a side effect. Can the authors clarify this point?

14. In all analyses using signaling agonists/antagonists, the authors only discussed the appearance of cells expressing some key genes. These are informative experiments, but in my opinion not sufficient to claim that the mechanisms for the symmetry breaking of the disc-shape human embryos is clarified.

Minor points:

1. In line #110, the authors annotated the syncytiotrophoblast and extravillous trophoblast by the expression of SDC1 and CGB1, however, no references were cited.
2. I could not access the analysis code. The authors should provide appropriate information for their computational analysis so that the readers can reproduce the authors' analysis.

References

- Gafni, O., Weinberger, L., Mansour, A.A., Manor, Y.S., Chomsky, E., Ben-Yosef, D., Kalma, Y., Viukov, S., Maza, I., Zviran, A., et al. (2013). Derivation of novel human ground state naive pluripotent stem cells. *Nature* 504, 282-286. [10.1038/nature12745](https://doi.org/10.1038/nature12745).
- Io, S., Kabata, M., Iemura, Y., Semi, K., Morone, N., Minagawa, A., Wang, B., Okamoto, I., Nakamura, T., Kojima, Y., et al. (2021). Capturing human trophoblast development with naive pluripotent stem cells in vitro. *Cell Stem Cell* 28, 1023-1039 e1013. [10.1016/j.stem.2021.03.013](https://doi.org/10.1016/j.stem.2021.03.013).
- Irie, N., Weinberger, L., Tang, W.W., Kobayashi, T., Viukov, S., Manor, Y.S., Dietmann, S., Hanna, J.H., and Surani, M.A. (2015). SOX17 is a critical specifier of human primordial germ cell fate. *Cell* 160, 253-268. [10.1016/j.cell.2014.12.013](https://doi.org/10.1016/j.cell.2014.12.013).
- Mackinlay, K.M., Weatherbee, B.A., Souza Rosa, V., Handford, C.E., Hudson, G., Coorens, T., Pereira, L.V., Behjati, S., Vallier, L., Shahbazi, M.N., and Zernicka-Goetz, M. (2021). An in vitro stem cell model of human epiblast and yolk sac interaction. *Elife* 10. [10.7554/eLife.63930](https://doi.org/10.7554/eLife.63930).
- Molè, M.A., Coorens, T.H.H., Shahbazi, M.N., Weberling, A., Weatherbee, B.A.T., Gantner, C.W., Sancho-Serra, C., Richardson, L., Drinkwater, A., Syed, N., et al. (2021). A single cell characterisation of human embryogenesis identifies pluripotency transitions and putative anterior hypoblast centre. *Nature Communications* 12. [10.1038/s41467-021-23758-w](https://doi.org/10.1038/s41467-021-23758-w).
- Nakamura, T., Okamoto, I., Sasaki, K., Yabuta, Y., Iwatani, C., Tsuchiya, H., Seita, Y., Nakamura, S., Yamamoto, T., and Saitou, M. (2016). A developmental coordinate of pluripotency among mice, monkeys and humans. *Nature* 537, 57-62. [10.1038/nature19096](https://doi.org/10.1038/nature19096).
- Pham, T.X.A., Panda, A., Kagawa, H., To, S.K., Ertekin, C., Georgolopoulos, G., van Knippenberg, S.,

Allsop, R.N., Bruneau, A., Chui, J.S., et al. (2022). Modeling human extraembryonic mesoderm cells using naive pluripotent stem cells. *Cell Stem Cell* 29, 1346-1365 e1310. 10.1016/j.stem.2022.08.001.

Sasaki, K., Nakamura, T., Okamoto, I., Yabuta, Y., Iwatani, C., Tsuchiya, H., Seita, Y., Nakamura, S., Shiraki, N., Takakuwa, T., et al. (2016). The Germ Cell Fate of Cynomolgus Monkeys Is Specified in the Nascent Amnion. *Dev Cell* 39, 169-185. 10.1016/j.devcel.2016.09.007.

Xiang, L., Yin, Y., Zheng, Y., Ma, Y., Li, Y., Zhao, Z., Guo, J., Ai, Z., Niu, Y., Duan, K., et al. (2019). A developmental landscape of 3D-cultured human pre-gastrulation embryos. *Nature*. 10.1038/s41586-019-1875-y.

Yang, R., Goedel, A., Kang, Y., Si, C., Chu, C., Zheng, Y., Chen, Z., Gruber, P.J., Xiao, Y., Zhou, C., et al. (2021). Amnion signals are essential for mesoderm formation in primates. *Nature Communications* 12. 10.1038/s41467-021-25186-2.

Zhai, J., Guo, J., Wan, H., Qi, L., Liu, L., Xiao, Z., Yan, L., Schmitz, D.A., Xu, Y., Yu, D., et al. (2022). Primate gastrulation and early organogenesis at single-cell resolution. *Nature*. 10.1038/s41586-022-05526-y.

Zheng, Y., Xue, X., Shao, Y., Wang, S., Esfahani, S.N., Li, Z., Muncie, J.M., Lakins, J.N., Weaver, V.M., Gumucio, D.L., and Fu, J. (2019). Controlled modelling of human epiblast and amnion development using stem cells. *Nature* 573, 421-425. 10.1038/s41586-019-1535-2.

Zheng, Y., Yan, R.Z., Sun, S., Kobayashi, M., Xiang, L., Yang, R., Goedel, A., Kang, Y., Xue, X., Esfahani, S.N., et al. (2022). Single-cell analysis of embryoids reveals lineage diversification roadmaps of early human development. *Cell Stem Cell* 29, 1402-1419 e1408. 10.1016/j.stem.2022.08.009.

Reviewer #2:

Remarks to the Author:

In this study, Weberling et al. first performed bioinformatics analysis of signaling pathways in mouse, cynomolgus macaque, and human embryos by using single cell RNAseq datasets publically available. They showed that NODAL and BMP signaling are enriched in the human epiblast at pre-implantation compared to post-implantation stages. By using human embryo cultures and embryonic stem cell models, the authors validated the role of NODAL and BMP in hypoblast specification and peri-implantation epiblast survival in vitro. While BMP signaling specifies anterior hypoblast identity in human, it inhibits anterior hypoblast in the mouse. In the mouse embryo, NODAL signaling is required to specify the anterior hypoblast. Lastly, by using in vitro human embryo culture with or without NOTCH inhibitor DAPT treatment, the authors showed that the NOTCH signaling is more important for survival of epiblasts at postimplantation stages but not preimplantation.

Major Critiques:

1) A recent research paper in *Dev Cell* also showed similar bioinformatics analysis and further demo the importance of NODAL signaling in mouse and human anterior-posterior formation. (Zhu et al. 2023 Jan 9; doi: 10.1016/j.devcel.2022.12.004. Decoding anterior-posterior axis emergence among mouse, monkey, and human embryos). The novelty of current paper would be on the BMP signaling in hypoblast specification and survival as well as the different role for BMP pathways in human and mouse early embryo development. The authors shall consider to substantially revise their manuscript by focusing on the novel role of BMP in anterior-posterior axis.

2) In this study, the authors mainly used left-over IVF embryo cultures to examine the perturbation of NODAL, BMP, and NOTCH signaling on epiblast and hypoblast development. However, their results

may be complicated by the uneven quality of IVF embryos during in vitro cultures. The generation of human embryoids and gastruloids from well-characterized human stem cells has provided a well-controlled system to study the patterning of anterior-posterior axis (e.g. doi.org/10.1038/s41586-020-2383-9). The authors have previously generated embryoids and gastruloids. The authors shall test the effect of compounds affecting BMP, NODAL, and Notch signaling during the time course of embryoid and gastruloid formation. These new data would allow authors to make more convincing conclusions.

3) The effect of NOTCH signaling inhibitor DAPT on the survival of D7-9 post-implantation epiblast is a bit striking to this reviewer. I would suggest that the authors may try out another well-established gamma-secretase inhibitor such as MK0752 and see if the result is consistent or not. Alternatively, a separate method such as NOTCH receptor knock-down in D7-9 embryo cultures would be more helpful.

4) The paper may be revised more concisely for take-home messages.

Reviewer #3:

Remarks to the Author:

The ethics statement informs me that this was reviewed and approved by the HFEA. The statement also notes that donors were given information about the project and about the ability to withdraw the embryos. What is missing from the description is anything about how the particular donors were recruited and chosen; the information they were given about other options for embryo management (e.g. freezing or disposal without donation to research); and whether the donors were the progenitors (ie, were the sperm and egg providers the same people as those making the decision to donate, or were anonymous or known gamete donors being used by a couple in which one or both partners could not provide a gamete).

Reviewer #4:

Remarks to the Author:

How the anterior-posterior axis is specified in post-implantation human embryos remain unclear due to the inaccessibility of human embryonic development at this stage. Weberling et al. examined the signaling pathways active in this key developmental event in human, monkey and mouse embryos using public scRNA-seq datasets, and interrogated the function of key signaling pathways using human and mouse in vitro models, including in vitro cultured human embryos beyond implantation. They uncovered both conserved and species-specific requirements for NODAL and BMP signaling in specifying and maintaining anterior hypoblast, and found that NOTCH signaling is required to maintain the epiblast and hypoblast in human embryos after implantation.

Overall, this study aimed to address a key question in human embryo development and put forth substantial efforts to understand the functions of key signaling pathways in specifying human anterior-posterior axis. Such investigations were now possible thanks to recent advance in in vitro culture of human embryos through implantation. Both conserved and species-specific regulation between human and mouse were identified. However, there are several issues that should be addressed as illustrated below.

Major comments:

1. The effects of specific signaling pathway inhibition on human embryos were mostly examined using IF. Some of them were however somewhat difficult to evaluate without proper quantification (see below for examples). Single-cell RNA sequencing could be one way to validate these results and conclusions, and further shed light on the underlying mechanism.
2. To quantify the activity of NODAL, the authors used the n/c ratio of total SMAD2.3 in individual cells. This could be misleading, especially when both signals in the nucleus and cytoplasm are high, but the ratio is low. For example, the authors stated "In segregating blastocysts, the n/c SMAD2.3 ratio was significantly higher in trophectoderm cells than inner cell mass cells, similar to the E3.5 early mouse blastocyst (Figure 2B-C, Extended Figure 4D-E)". However, I cannot see a higher n/c SMAD2.3 ratio in trophectoderm cells from IF results. Moreover, the signal of SMAD2.3 seems higher in ICM/EPI/HYPO than that in TE. The authors should show the original signals in the nucleus and cytoplasm side by side.
3. In fact, the activity of NODAL signaling is often evaluated by pSMAD2/3. In addition, the authors quantified BMP signaling by pSMAD1/5/9 (or pSMAD1/5/8). It is unclear why the authors used different strategies (n/c SMAD2.3) in these cases.
4. The differences in the requirement of BMP4 in pre-implantation epiblast survival in human and mouse are intriguing. Can the authors find any possible mechanisms underlying such species-specific divergence based on the scRNA-seq data?
5. Some statements were overstated in the paper. For example, the authors stated that "The NODAL module was noticeably enriched in the epiblast at the blastocyst stage in human and macaque". But NODAL module is much more enriched in peri-implantation epiblast in monkey (Fig. S1E). In addition, "restricted" was frequently used in this manuscript, but was misleading in some occasions. For example, BMP is not restricted to hypoblast, but also enriched in EXMC in monkey (Fig. S1E). NODAL is not restricted to, but only relatively more enriched in the hypoblast in human. The authors should tone down their statements and describe the results/conclusions accurately.
6. In Figure 4L, why activin A at a relative low concentration can induce apoptosis of primed hESCs, given it was shown to facilitate the maintenance of hESCs at a primed state (James et al., *Development*, 2005; Beattie et al., *Stem Cells*, 2005)?
7. The authors stated that "pSMAD1.5.9 expression decreased over time in the human epiblast, with a distinct bimodal intensity pattern at D9". However, I did not see the bimodality of pSMAD1.5.9 in Figure 3E.
8. Figure 2J. Could the authors provide an IF figure?
9. The model could be more informative by including the functional information of these signaling pathways.

Minor comments:

1. Extended Figure 4. The description in the manuscript and the figure index did not match. Fig. S4D is missing.
2. Some references are inappropriately cited. For example, ref. 46 was not mainly about the nuclear effectors of BMP signaling.
3. To easily reproduce the analysis shown in this work, I suggest that the authors publicly provide the processed data (given all the raw scRNA-seq data were published) (i.e., Seurat object). I also cannot see the source code at <https://github.com/bweatherbee/PeriImplantation>.

FINANCIAL AND NON-FINANCIAL COMPETING INTERESTS – the authors must include one of three declarations: (1) that they have no financial and non-financial competing interests; (2) that they have financial and non-financial competing interests; or (3) that they decline to respond, after the Author Contributions section. This statement will be published with the article, and in cases where financial

and non-financial competing interests are declared, these will be itemized in a web supplement to the article. For further details please see <https://www.nature.com/licenceforms/nrg/competing-interests.pdf>.

Methods should be written concisely, but should contain all elements necessary to allow interpretation and replication of the results. As a guideline, Methods sections typically do not exceed 3,000 words. The Methods should be divided into subsections listing reagents and techniques. When citing previous methods, accurate references should be provided and any alterations should be noted. Information must be provided about: antibody dilutions, company names, catalogue numbers and clone numbers for monoclonal antibodies; sequences of RNAi and cDNA probes/primers or company names and catalogue numbers if reagents are commercial; cell line names, sources and information on cell line identity and authentication. Animal studies and experiments involving human subjects must be reported in detail, identifying the committees approving the protocols. For studies involving human subjects/samples, a statement must be included confirming that informed consent was obtained. Statistical analyses and information on the reproducibility of experimental results should be provided in a section titled "Statistics and Reproducibility".

All Nature Cell Biology manuscripts submitted on or after March 21 2016 must include a Data availability statement as a separate section after Methods but before references, under the heading "Data Availability". For Springer Nature policies on data availability see <http://www.nature.com/authors/policies/availability.html>; for more information on this particular policy see <http://www.nature.com/authors/policies/data/data-availability-statements-data-citations.pdf>. The Data availability statement should include:

- Accession codes for primary datasets (generated during the study under consideration and designated as "primary accessions") and secondary datasets (published datasets reanalysed during the study under consideration, designated as "referenced accessions"). For primary accessions data should be made public to coincide with publication of the manuscript. A list of data types for which submission to community-endorsed public repositories is mandated (including sequence, structure, microarray, deep sequencing data) can be found here <http://www.nature.com/authors/policies/availability.html#data>.
- Unique identifiers (accession codes, DOIs or other unique persistent identifier) and hyperlinks for datasets deposited in an approved repository, but for which data deposition is not mandated (see here

for details <http://www.nature.com/sdata/data-policies/repositories>).

- At a minimum, please include a statement confirming that all relevant data are available from the authors, and/or are included with the manuscript (e.g. as source data or supplementary information), listing which data are included (e.g. by figure panels and data types) and mentioning any restrictions on availability.
- If a dataset has a Digital Object Identifier (DOI) as its unique identifier, we strongly encourage including this in the Reference list and citing the dataset in the Methods.

We recommend that you upload the step-by-step protocols used in this manuscript to the Protocol Exchange. More details can found at www.nature.com/protocolexchange/about.

All imaging data should be accompanied by scale bars, which should be defined in the legend. Cropped images of gels/blots are acceptable, but need to be accompanied by size markers, and to retain visible background signal within the linear range (i.e. should not be saturated). The boundaries of panels with low background have to be demarked with black lines. Splicing of panels should only be considered if unavoidable, and must be clearly marked on the figure, and noted in the legend with a statement on whether the samples were obtained and processed simultaneously. Quantitative comparisons between samples on different gels/blots are discouraged; if this is unavoidable, it should only be performed for samples derived from the same experiment with gels/blots were processed in parallel, which needs to be stated in the legend.

- For line art, graphs, charts and schematics we prefer Adobe Illustrator (.AI), Encapsulated PostScript (.EPS) or Portable Document Format (.PDF). Files should be saved or exported as such directly from the application in which they were made, to allow us to restyle them according to our journal house

style.

- We accept PowerPoint (.PPT) files if they are fully editable. However, please refrain from adding PowerPoint graphical effects to objects, as this results in them outputting poor quality raster art. Text used for PowerPoint figures should be Helvetica (preferred) or Arial.
- We do not recommend using Adobe Photoshop for designing figures, but we can accept Photoshop generated (.PSD or .TIFF) files only if each element included in the figure (text, labels, pictures, graphs, arrows and scale bars) are on separate layers. All text should be editable in 'type layers' and line-art such as graphs and other simple schematics should be preserved and embedded within 'vector smart objects' - not flattened raster/bitmap graphics.
- Some programs can generate Postscript by 'printing to file' (found in the Print dialogue). If using an application not listed above, save the file in PostScript format or email our Art Editor, Allen Beattie for advice (a.beattie@nature.com).

Supplementary items should relate to a main text figure, wherever possible, and should be mentioned sequentially in the main manuscript, designated as Supplementary Figure, Table, Video, or Note, and numbered continuously (e.g. Supplementary Figure 1, Supplementary Figure 2, Supplementary Table

1, Supplementary Table 2 etc.).

The total number of Supplementary Figures (not including the “unprocessed scans” Supplementary Figure) should not exceed the number of main display items (figures and/or tables (see our Guide to Authors and March 2012 editorial <http://www.nature.com/ncb/authors/submit/index.html#suppinfo>; <http://www.nature.com/ncb/journal/v14/n3/index.html#ed>). No restrictions apply to Supplementary Tables or Videos, but we advise authors to be selective in including supplemental data.

GUIDELINES FOR EXPERIMENTAL AND STATISTICAL REPORTING

REPORTING REQUIREMENTS – We are trying to improve the quality of methods and statistics reporting in our papers. To that end, we are now asking authors to complete a reporting summary that collects information on experimental design and reagents. The Reporting Summary can be found here <https://www.nature.com/documents/nr-reporting-summary.pdf>) If you would like to reference the guidance text as you complete the template, please access these flattened versions at <http://www.nature.com/authors/policies/availability.html>.

Author Rebuttal to Initial comments

Weatherbee, Weberling, & Gantner, et al.; Response to Reviewers

We are very grateful to the reviewers for their thoughtful feedback and suggestions, which have significantly improved our manuscript. Below, we respond to each specific comments. We would like to note that our responses have their own reference list, and the reference numbers may differ from those in the manuscript. Throughout, we provide figure panels where new data has been added or revised and provide details on where this is included in the updated manuscript. Line references are specific to the updated manuscript file submitted alongside this response to reviewers document. Text added to the manuscript is marked by << >>.

In summary, we have added significant new data including:

1. Assessment of NODAL perturbation on trophectoderm development in the human embryo. This has demonstrated a potential positive role for NODAL in driving the maturation of the trophectoderm towards implantation-competent trophoblast.
2. Functional testing of yolk sac-like cells' ability to protect post-implantation epiblast-like embryo models (human ESC spheroids) from BMP signaling. We demonstrate that the inhibition of NODAL, BMP, or NOTCH signaling decreases CER1 expression in yolk sac-like cells (similar to our results in the human embryo itself), and this perturbs the ability of the anterior hypoblast/yolk sac-like cells to antagonize BMP signaling in epiblast-like human ESC spheroids.
3. Demonstration that the differential stage dependence of BMP in mouse and human embryos is likely related to species-specific roles of BMP in the naïve-to-primed pluripotency spectrum. Specifically, BMP response genes in the pre-implantation epiblast correlate positively with human naïve pluripotency genes but show either no or negative correlation with mouse naïve pluripotency. Using stem cell analogous of naïve and primed states in human and mouse, we demonstrate inverse patterns of pSMAD1.5.9 expression between species. We also show that inhibition of BMP downregulates human-specific naïve pluripotency markers but conversely negatively impacts mouse-specific primed markers.
4. Experimentation with two additional γ -secretase inhibitors strengthens our conclusions that NOTCH/ γ -secretase activity is important for the survival of the post-implantation human epiblast and anterior hypoblast specification.
5. Several new experiments adding embryos across many of the conditions have added confidence for our statistical analyses and quantifications.
6. Provision of high magnification and single channel images, individual fluorescence intensity values, and individual gene expression plots provide greater confidence in our analyses and strengthens our conclusions.

We believe that the reviewers' comments have enabled us to significantly improve our study. We hope the reviewers will agree that the work represents an important resource for understanding the signaling dynamics in the human pre- to post-implantation development transition. In particular, we have identified species-specific divergences in the role of BMP signaling for epiblast and hypoblast maturation between mouse and human. Additionally, while our work does not define molecular mechanisms for anterior-posterior axis formation *per se*, we have shown the role of specific pathways in the formation of the human anterior signaling center for the first time.

Reviewer #1

The manuscript by Weberling et al. explored the molecular mechanism for the specification of the anterior visceral endoderm (AVE) in human embryos. The authors generated integrated massive scRNA-seq datasets of peri-implantation embryos in humans, monkeys and mice, and found that NODAL, BMP and NOTCH pathways are potentially active in the epiblast (EPI) and hypoblast (HYPO) lineages. By using agonists/antagonists of these signaling pathways for culturing human embryos and stem cell-based differentiation models, they found that both NODAL and BMP signaling are required for the AVE maturation/maintenance, which is somewhat in contrast to the case in mice that require high NODAL/low BMP activity for the AVE specification. They also found that NOTCH activity is crucial for the survival of both EPI and HYPO in an early post-implantation period.

The authors reported previously that putative AVEs were identified in human embryos cultured in vitro (Molè et al., 2021), and this work is a follow-up study, aiming to understand the mechanism that determines the anterior-posterior (AP) axis in human embryos. On the other hand, I found that some of the data shown in the figures are not really convincing, and some of the statements proposed are inconsistent. Although the authors identified the signaling pathways that could be important for the maintenance/specification of EPI/HYPO/AVEs, they did not provide evidence on the mechanism for the AP axis formation and symmetry breaking in the human embryos with a disc-shaped early post-implantation structure. Below are my major concerns:

Major points:

1. I simply wonder whether the integrated scRNA-seq data is reliable enough and whether there is a need to integrate them. There are only 3 major clusters in the UMAP derived from the integrated datasets (Fig1E), while more clusters were observed in the original papers, for example, Xiang et al. (Xiang et al., 2019), suggesting that data integration by a strong normalization method (maybe too much artificial) reduces the resolution and masks key features in each dataset. Rather, the authors have already identified AVEs in their own scRNA-seq data reported in the previous report (Molè et al., 2021). I therefore wonder if it would be possible to obtain more meaningful results from the analysis of each single dataset, or with the re-analysis of the authors' previous dataset.

We agree with the reviewer that dataset integration has the potential to overcorrect and mask features within the larger dataset. However, we believe that given the limited number of cells and/or timepoints of current datasets (e.g. only 557 cells total in Xiang et al.³ across 7 time points, or in Molè et al. the inclusion of only 9 and 11 d.p.f. samples⁴) and the mix of technologies used to sequence, that integration is very helpful to generate a comprehensive dataset of pre-to-post-implantation development that allows for probing of signaling dynamics.

Regarding cell type identification, course clusters generated by our combined dataset are similar to published data. For example, Xiang et al.³ describe 7 clusters (ICM, EPI, PSA-EPI, PrE, CTBs, STBs and EVT), and in our previous dataset we described 4 (EPI, HYPO, CTB, STB), while we identify 5 major clusters in this integrated dataset (EPI, HYPO, TE/TrB, STB and EVT). However, it is important to note that doubt has been cast on the clustering in Xiang et al. (see Chhabra & Warmflash, *Biol Open*, 2021⁵), including the ICM cluster which appears to be mislabeled cytotrophoblast and expresses *GATA2* and *TEAD4* (Xiang et al.³, Figure 2C). Many of these clusters are corrected based on realignment to a reference genome without pseudogenes included, as described previously⁵. The reference used here does not include pseudogenes, and it is therefore unsurprising that their mislabeled ICM cluster is not identifiable here, nor their reported amnion population when the epiblast is subclustered.

Here we provide two examples of maintained biological differences in our course clusters. Firstly, the anterior hypoblast in our previous manuscript⁴ was identified by subclustering. When subclustering this combined dataset similarly, we are indeed able to identify this subpopulation which is significantly enriched for several markers of the anterior Hypoblast (*CER1*, *LEFTY1*, *NOG*). This data is included below for the reviewer in Response to Reviewers Figure 1.1.

Response to Reviewers Figure 1.1: Subclustering of the hypoblast in the integrated dataset. (A) UMAP of the hypoblast cluster taken directly from integrated, combined dataset, colored by subcluster generated by the FindClusters function in Seurat. (B) UMAP of the hypoblast cluster colored by paper of origin. (C) FeaturePlots of canonical anterior Hypoblast markers within the hypoblast cluster. Each of these is significantly enriched by ROC analysis in subcluster 0.

Secondly, PCA on the integrated dataset clearly separates the naïve blastocyst epiblast in PC3 and this is further confirmed by plotting expression of the naïve marker *KLF17*. Similarly, PC5 distinguishes the putative primitive streak anlage (PSA-EPI) population (which includes several PSA-EPI cells from Xiang, et al.³) by enriched *TBXT* expression. Importantly, when a PCA is run on the ‘raw’ read data (not SCT normalized and thereby not accounting for the integration from various sources), the variation between papers is dominant over any biological signal. This underscores the importance of using integration methods and normalized values when running these analyses, as reported by other groups (see Figure S2 of Stuart et al., *Cell* 2019⁶ – integration of SmartSeq2 and 10x genomics dataset; see also Hafemeister & Satija, *Genome Biology* 2019⁷ for more information on the sctransform normalization method used on the data here). Overall, we are confident that the integration approaches do not overly mask known subpopulations within these clusters.

Responses to Reviewers Figure 1.2: PCA Delineates the Naïve to Primed Transition within the Integrated Epiblast Cluster. (A) PCA plots of the epiblast cluster with principal component analysis run on variable features with SCT normalized counts. **(B)** PCA plots of the epiblast with principal component analysis run on the same variable features as **A** but on raw $\log_1 p([q]UMI)$ counts.

2. While the analyses using wikipathway and CellPhonDB are useful, I think that simple expression data is also very informative for the reader's understanding. Could the authors provide information of expression levels of at least key genes using standard methods, such as heatmap or violin plot? In addition, the real values, such as $\log_2(UMI)$ values, rather than the scaled ones as shown in Fig. 5A and Fig. 6A, should be provided, because although it is easy to see the differences between samples in the scaled values, it is not clear whether the expression levels themselves are biologically significant or not.

We thank the reviewer for their suggestion and now provide violin plots of key pathway components in Extended Data Fig. 3b, and below for convenience.

Please note that given the use of datasets generated using different technologies with varying read depth, the use of raw expression values can be misleading (for example, UMI and non-UMI based methods). We used appropriate normalization methods to mitigate these risks (quminorm and sctransform). Below, we provide the violin plots of both SCT Normalized and $\log_1 p([q]UMI)$ values, and we do note they are largely similar.

Response to Reviewers Figure 1.3: Violin plots of selected pathway component gene expression. (A-B) Expression of WNT, FGF, BMP and NODAL associated-genes plotted across stages of human, macaque, and mouse pre-gastrulation development with expression values normalized using sctransform (A) or log1p([q]UMI) (B).

3. In line #112, based on the expression of GATA6 and COL6A1, the authors claimed that extra-embryonic mesenchyme was not observed in *in vitro* cultured human embryos. These genes were reported as markers for the extra-embryonic mesenchyme, but they are broadly expressed and are indeed also found in the other cell types in monkey embryos (Zhai et al., 2022) (Nakamura et al., 2016) (Yang et al., 2021). So they may not be critical markers. On the other hand, Pham et al. recently reported the induction of mesenchymal cells from human naïve pluripotent stem cells with the identification of new markers, such as LUM, POSTN, VIM and DCM (Pham et al., 2022). I suggest the authors to examine the expression of these genes and confirm whether the extraembryonic mesenchyme is really absent in *in vitro* cultured human embryos.

We thank for this suggestion and have now examined the markers of *in vitro* human extraembryonic mesenchyme reported in Pham et al.⁸ including *VIM*, *POSTN*, and *LUM* and confirm that, while these markers robustly mark the cynomolgus macaque extraembryonic mesenchyme cluster, this cell population is absent in our *in vitro* cultured human embryos, as previously reported (Extended Data Fig. 1d; included below).

We have addressed this in our initial explanation of our experimental regime in L206-209:

<< As amnion and extraembryonic mesenchyme do not form robustly during *in vitro* culture of human embryos^{3-5, 9-11}, we limited our experimentation to time frames preceding their expected differentiation *in vivo* to mitigate signaling defects associated with the lack of these tissues. >>

We have also added the following in the discussion in L453-456.

<< Of note, the extraembryonic mesenchyme and amnion in primates are likely to be sources of BMP ligands in addition to the hypoblast^{8, 12-15}, however, these populations do not differentiate robustly during *in vitro* culture of human embryos in current systems^{3-5, 9-11}. >>

Response to Reviewers Figure 1.4: Expression of extraembryonic mesenchyme markers in macaque and human datasets.

4. The authors suggested that the main source of BMP ligands is HYPO, but amnionic ectoderm also expresses BMPs in monkey embryo (Sasaki et al., 2016) and human *in vitro* amnion models

(Zheng et al., 2019; Zheng et al., 2022). The authors should discuss the implications of the amnionic ectoderm.

We agree with the reviewer that this an important point. Similar to the extraembryonic mesenchyme, we observe that the amnion is not formed robustly in *in vitro* cultured human embryos, as has been reported elsewhere^{4, 9}. Further reanalysis of Xiang et al. previously suggested that 'AME' is likely mislabeled EVT, STB^{3, 16}. In the present manuscript, we have deliberately limited our experimentation to day 9 post-fertilization, prior to formation of the extraembryonic mesoderm and amnion. The lack of these tissues *in vitro* during human embryo culture should have less effect on other tissues at these stages.

As noted above, we have added the following to the discussion section in L453-456:

<< Of note, the extraembryonic mesenchyme and amnion in primates are likely to be sources of BMP ligands in addition to the hypoblast^{8, 12-15}, however, these populations do not differentiate robustly during *in vitro* culture of human embryos in current systems^{3-5, 9-11}. >>

5. In line #245, the authors used a Bayesian statistical model for the cell count analysis. Is this method also reliable enough? I think it is important for the authors to provide some results of "control" analysis that shows the validity of the statistical method used.

We are grateful for this suggestion. We employed Bayesian approaches to our analysis of count data because this method may be more suited for data with wide variation and low sample sizes, such as the human embryos reported here. However, in the updated version of the manuscript, we have added more embryos across many of the conditions, which has improved our statistical power. Therefore, to simplify interpretation of our data we have chosen to remove the Bayesian statistical plots from the main figures. Overall, Bayesian inference-based statistics ask *if it is probable that a specific hypothesis is true based on the data observed*, in contrast to the frequentist question of *if the data observed is probable assuming the null hypothesis is true*. Due to this difference in fundamental pillars, the Bayesian approach does not need to assume sampling to the 'true mean' of a population in order for the test to be valid, while frequentist methods require this assumption (Hackenberger, 2019¹⁷).

Regarding the analysis here, we have now added a set of Bayesian analyses where the Control groups for each count of cells (epiblast, hypoblast and anterior hypoblast at day 7 and 9, and trophoblast/trophectoderm cells at day 7) are subsampled into two groups, and then performed the brms analysis. This serves as a negative control for the analysis whereby the credible interval should cross zero given that they are from the same distribution. This is indeed the case for the most part, and the coefficients and credible intervals are now included in Extended Data Table 8. Given that a few of these control analyses have resulted in credible intervals that come close to, but do not cross zero, we have now imparted stricter criteria for these analyses throughout the manuscript where intervals ± 0.33 are used rather than simply ± 0 .

Nevertheless, we used this method alongside more traditional analysis (e.g. Mann Whitney Test). While the two approaches largely agree, we find both informative when working with this data, and utilize both when inferring biological meaning. For example, in Fig. 4 and Extended Data Fig. 6 we observe that NODAL inhibition (using A83-01) significantly inhibits formation and/or maintenance of the CER1+ Hypoblast when assessed by the Mann-Whitney Test ($p=0.0004$ compared to control), as well as the Bayesian statistical model (coefficient estimate: -3.27; 95% CI: -5.14, -1.94).

6. As commented in the point 2 above, the expression levels of key genes should be shown in real values, rather than scaled values (Fig. 5A, Fig. 6A, ExFig. 3B, ExFig. 4A-C, ExFig. 5A, B, ExFig. 7C).

Given our addition of a large set of violin plots in the new version of the manuscript (included as Response to Reviewers Figure 1.3 above), several of these plots are no longer included in the resubmitted version to avoid redundancy.

In general, we have elected to utilize SCT normalized values to display expression data. This normalization method accounts for differences in read depth across the various datasets used, which is an important consideration given the vast differences between single cell RNA sequencing datasets. We think this might be more appropriate than using raw values that are not controlled for differences in read depth.

7. In the IF analysis, the authors differentiated the “Segregating” and “Segregated” blastocysts. However, according to Fig2B, I could not distinguish between these two and I wonder whether the authors classified the embryo stages in a proper manner. In the pSMAD-level analysis, I also wondered whether the authors defined the nucleus/cytoplasm in each cell in an appropriate manner. These issues are essential for the following quantification, statistical analysis and the author’s statements, but unfortunately, I do not think they are convincing enough.

We thank the reviewer for the chance to clarify our criteria. We have defined segregating and segregated blastocysts based on the co-expression of epiblast and hypoblast markers at day 5 or day 6, as reported elsewhere¹⁸. For example, if cells within the inner cell mass express both GATA6 and SOX2 they were assigned as ‘segregating’ to indicate that these cells did not show fully defined separation of the two inner cell mass-derived lineages. Conversely, embryos at the same stages (day 5-6) which displayed distinct, non-overlapping expression of GATA6 and OCT4 were labelled segregated. We now provide additional examples with higher power of the inner cell mass of ‘segregating’ and ‘segregated’ blastocysts (including those in Figs. 2 and 4 as the second and third examples of each category) in Extended Data Fig. 4a-b (included below). We believe this is the most biologically meaningful method to stage the embryos at these early timepoints as they may have been frozen at different stages in the clinic, and it would not be possible to classify cells co-expressing these markers as either epiblast or hypoblast.

Response to Reviewers Figure 1.5: Examples of segregating and segregated blastocysts. (A) Examples of segregating blastocysts with inner cell mass cells co-expressing epiblast and hypoblast markers marked with grey asterisks. **(B)** Examples of segregated blastocysts. Note exclusive expression of epiblast and hypoblast markers in the inner cell mass. Scale bars: 100µm.

This is noted in the manuscript in L174-177 as follows:

<< To accurately capture early blastocyst development, we further classified blastocysts at day 5 or 6 as ‘segregating’ (inner cell mass co-expressing epiblast and hypoblast markers) or ‘segregated’ (inner cell mass with mutually exclusive epiblast and hypoblast protein expression) (Extended Data Fig. 4a-b)¹⁹⁻²¹ >>

For pSMAD1.5.9 analysis we have directly quantified nuclear signal normalized to background fluorescence taken from an acellular region of the image. We have noticed that this was misstated in some figure captions and have now corrected this. We apologize for this error. For SMAD2.3 analysis we utilized an antibody that detects total SMAD2.3 and therefore, the meaningful proxy of active NODAL signaling is the proportion of SMAD2.3 that translocates to the nucleus. The nuclei region of interest (ROI) was defined based on DAPI, and the cytoplasmic ROI was manually drawn using membrane E-Cadherin or Phalloidin (F-actin) as a reference. Examples of the ROIs are provided in Figs. 2a and 4a. As noted below, we now include individual nuclear and cytoplasmic values in addition to the ratio.

8. In line #188, the authors stated that the SMAD2/3 activity seems high in trophectoderm cells. However, several reports about human trophectoderm induction from naïve pluripotent stem cells consistently show that activin/nodal inhibition (by A83-01) is required for trophectoderm differentiation. So, it sounds controversial. Could the authors elaborate this point further?

We agree with the reviewer that this is an interesting and surprising observation. Our updated manuscript now includes individual nuclear and cytoplasmic SMAD2.3 values plotted alongside the nuclear/cytoplasmic ratio, which are provided below for convenience. These data show that, while the ratio of SMAD2.3 in the trophectoderm is relatively high, the overall SMAD2.3 expression is lower compared to inner cell mass and hypoblast. Our CellphoneDB analysis does highlight the possibility of NODAL signaling in the trophectoderm. Specifically, the trophectoderm appears competent (i.e. expresses NODAL receptor) to receive epiblast-secreted NODAL (see Fig. 1g). It is interesting to speculate that trophectoderm maintenance and maturation in the human blastocyst may not follow the same pathways as deriving trophectoderm from naïve hESC.

Response to Reviewers Figure 1.6: Nuclear, cytoplasmic, and nuclear-to-cytoplasmic ratio quantification of SMAD2.3 in human embryos. (A-D) SMAD2.3 fluorescence in nuclear (nuc; left) or cytoplasmic (cyto; middle) regions of interest plotted on the left axis, and nuclear-to-cytoplasmic ratio (nuc/cyto) plotted in segregating (A) or segregated (B) blastocysts and day 7 (C) and day 9 (D) post-fertilization (d.p.f.). ICM=inner cell mass, EPI=epiblast, HYPO=hypoblast, TE=trophectoderm. Statistics: (A, C-D) Unpaired T-test; (B) One-way ANOVA with Tukey-Kramer post-hoc. ****p<0.0001, ***p<0.001, *p<0.05.

We have now provided a discussion on this point in L441-449, which is included below:

<< Previous work has shown that isolated inner cell masses in MEK and NODAL inhibitors generated trophoblast cell outgrowths robustly and that these conditions can be used to differentiate trophoblast cells from naïve human ESCs²². However, the addition of A83-01 alone from day 5 to 7 does not appear to drive differentiation of the pre-implantation epiblast to the trophoblast in whole embryos in our study. This suggests that either the combinatorial inhibition of both FGF-MAPK and NODAL could be crucial for trophoblast differentiation *in vitro*, or that the differentiation of inner cell mass outgrowths and naïve ESC towards trophoblast follows distinct pathways compared to trophoblast specification in the morula.
>>

9. In IF analysis in Fig4-6, it is not possible for me to distinguish CER1 expression.

We thank the reviewer for this comment and now provide high power images and a single channel of CER1 to make this clearer in these figures. Relevant panels from the NODAL perturbation figure (Fig. 3, previously Fig. 4) are provided below as an example.

Response to Reviewers Figure 1.7: Updated panels with higher power and single channel CER1 expression. (A) Immunofluorescence images of embryos cultured from day 5-7 in control (N=44 embryos), 2µM A83-01 (N=12 embryos), or 25 ng/ml Activin A (N=16 embryos) conditions. Regions for high power images of CER1 are outlined with dashed boxes. (B) Immunofluorescence images of embryos cultured from day 7-9 in control (N=35 embryos), 2µM A83-01 (N=12 embryos), or 25 ng/ml Activin A (N=15 embryos) conditions. Regions for high power images of CER1 are outlined with dashed boxes. (C) Immunofluorescence images of yolk sac-like cells (YSLCs) which were treated with 2µM A83-01 for 48 hours during YSLC specification or maturation showing loss of CER1 expression. Scale bars: (A-B) 100µm, (C) 50µm.

10. Similar to the point 8 above, did the A83-01 treated embryos increase the number of trophoblast cells? The authors simply examined the number of POU5F1/GATA6 positive cells (EPI/HYPO) as the effect of the A83-01 treatment of human embryos. However, A83-01 enhances the naïve pluripotent stem cells to differentiate to trophoblast, so the authors should also include and discuss the effect on trophoblast.

We agree with the reviewer that this is a meaningful question to explore and now include new experiments treating embryos between day 5 and 7 with either A83-01 or Activin-A followed by staining for GATA3 and GATA6 (included in Extended Data Fig. 5I-s and below in Response to Reviewers Fig. 1.8). We do not observe a change in total GATA3-positive cells with A83-01 treatment, but we do observe a reduction in GATA3-positive cells with Activin-A treatment. This reduction can be attributed to a specific reduction in the GATA3/GATA6-double positive

population, which marks pre-implantation trophoctoderm²³. This population is diminished after implantation (e.g. in day 9 embryos; see Response to Reviewers Fig. 1.8G). In line with a role of NODAL signaling in trophoctoderm maturation, we observe an increase in the proportion of embryos implanted at day 7 after Activin-A treatment, compared to A83-01. This is in agreement with recent work suggesting that epiblast-derived signals induce local maturation of the trophoctoderm²⁴. We also do not observe a substantial decrease in epiblast cells after addition of A83-01, which would be expected if A83-01 treatment induced trophoctoderm specification from the epiblast. It is likely that treatment from day 5 to 7 does not capture the differentiation of trophoctoderm from its parental cell population at the morula stage^{25, 26}, and will more closely reflect later effects of NODAL on the trophoctoderm.

Response to Reviewers Figure 1.8: Quantification of trophoctoderm cell number and subtype after NODAL modulation. (A) Immunofluorescence images of embryos cultured from day 5-7 in control (N=8 embryos), 2 μ M A83-01 (N=6 embryos), or 25 ng/ml Activin-A (N=6 embryos) conditions. (B-D) Quantification of the number of total GATA3-positive (GATA3+) (B), GATA3+/GATA6-negative (-) (C), and GATA3+/GATA6+ double-positive (D) trophoctoderm/trophoblast from day 5-7. (E) Visualization of the Bayesian statistical analysis for counts presented in C-E. Coefficient estimated by brms R package for divergence from control where a coefficient with credible interval (CI) above 0.33 indicates a positive shift in the distribution, while below -0.33 indicates a negative shift. (F) Quantification of embryo implantation after culture from day 5-7 in control (N=16 embryos), 2 μ M A83-01 (N=12 embryos), or 25 ng/ml Activin-A (N=13 embryos) conditions (defined as attached with \geq 75% of blastocoelic cavity collapsed). (G) Immunofluorescence images of a day 9 embryo cultured in control conditions (N=4 embryos). (H) Quantification of the number of total GATA3+, GATA3+/GATA6-, and GATA3+/GATA6+ cells at day 9. Note the loss of GATA6 expression in the GATA3+ trophoblast. For box plots, box encompasses 25th and 75th quartiles with median marked by central line. Minimum and maximum and denoted by whiskers. A '+' symbol marks the mean. Statistical tests: (B-D) Mann-Whitney test, (F) Fisher's Exact test. **p<0.01, *p<0.05. Unmarked pairwise comparisons are not significant (ns). Scale bars: 100 μ m.

11. This might be an issue for the previous paper about yolk sac-like cell (YSLC) induction (Mackinlay et al., 2021); although the YSLC is not the definitive endoderm, the cells in the RseT medium that was established according to Gafni et al. (Gafni et al., 2013) must express some gastrulating genes such as T (Irie et al., 2015), and their transcriptome is corresponding to monkey post-implantation epiblast (Nakamura et al., 2016), suggesting they are neither so-called

“naïve” nor “formative”. If YSLCs are “yolk sac” cells, what differentiation pathway do they follow? I wonder whether it would be appropriate to use YSLCs for the AVE analysis.

We agree with the reviewer that the identity of NHSM cells originally reported in Gafni et al. remains controversial. The evidence from our group¹ and the Brickman group²⁷ point to an intermediate peri/early post-implantation identity of RSeT human ESC.

Figure legend on next page

Response to Reviewers Figure 1.9: Validation of post-implantation extraembryonic endoderm differentiation (reproduced from Mackinlay et al., 2021¹). (A) Line graph (\pm SEM) plotting the relative fluorescence intensity (RFI arbitrary units [a.u.]) of SOX2, SOX17, and BRACHYURY in Activin-A, Chiron, and LIF (ACL)-treated populations. (B) Immunofluorescence images of SOX2 (green), SOX17 (red), and BRACHYURY (grey) in hEPSCs, RSeT hESCs, and primed hESCs during 6 days of ACL treatment. Note that Brachyury is not observed during yolk sac-like differentiation from RSeT ESC. (C) Venn diagram depicting the number of shared genes between human extra-embryonic endoderm (D6–14) and human embryonic stem cell (hESC)-derived definitive endoderm (brown), of extra-embryonic endoderm differentially expressed genes (orange), and of definitive endoderm differentially expressed genes (green). (D) MA plot of differentially expressed genes between human definitive endoderm and extra-embryonic endoderm. Top 20 differentially expressed genes are labelled, along with key extra-embryonic endoderm markers *NID2*, *PDGFRA*, *HNF4a*, *HNF1 β* , *PITX2*, *PODXL*, and *HHEX* (>10% of cell type of interest, $\log_2FC > 0.25$, $p < 0.05$). (E) Transcriptomic-signature comparison score of human extra-embryonic endoderm vs. definitive endoderm within ACL-treated hPSCs (gene set variation analysis [GSVA]) score for negative markers subtracted from GSVA score for positive markers and values normalized to 1). A negative value represents definitive endoderm similarity, and a positive value represents an extraembryonic endoderm similarity. (F) Venn diagram depicting the number of shared genes between pre-implantation extra-embryonic endoderm (D6–7) and post-implantation extra-embryonic endoderm (D10–14) (brown), of pre-implantation extraembryonic endoderm differentially expressed genes (green) and of post-implantation extraembryonic endoderm differentially expressed genes (orange). (G) MA plot of differentially expressed genes between human pre-implantation and post-implantation extraembryonic endoderm. Top 20 differentially expressed genes are labelled. (H) Transcriptomic-signature comparison score of pre-implantation extra-embryonic endoderm vs. post-implantation extra-embryonic endoderm for T2iLGö-Pre and RSeT-ACL (gene set variation analysis [GSVA]) score for pre-implantation extra-embryonic endoderm subtracted from GSVA score for post-implantation extraembryonic endoderm and values normalized to 1). A negative value (green) represents pre-implantation extraembryonic endoderm similarity, and a positive value (orange) represents post-implantation extraembryonic endoderm. Statistics: (A) Kruskal Wallis test. **** $p < 0.0001$. Scale bars: 50 μ m.

Importantly, these cells are capable of giving rise to extraembryonic endoderm, unlike traditional mTeSR cultured primed human ESC^{1,28}. In our hands, T/Brachyury is not temporarily upregulated during differentiation to YSLC from RSeT starting state as would be expected if ESCs were transiting through a mesendoderm/gastrulation-like cell state (data from Mackinlay et al.¹, reproduced below). This is in contrast to primed and expanded potential starting states, which also transcriptomically show higher gene set enrichment scores for definitive endoderm versus extraembryonic endoderm-enriched genes. In this previous work, we conclude through a variety of analyses that RSeT human ESC can give rise to extraembryonic endoderm-like cells. Our analyses also demonstrated that extraembryonic endoderm cells derived from naïve PXGL conditions²⁷ give rise to pre-implantation-like rather than post-implantation-like cells. Importantly, PXGL/naïve human ESC-derived endodermal cells do not express CER1 and therefore, are not amenable to understanding formation and maintenance of the post-implantation anterior hypoblast population.

12. Fig5M-P shows that LDN treatment decreased the number of GATA6- and CER1-positive cells during the YSLC induction, and the authors suggested that the activation of BMP pathway is required for the VE/AVE differentiation. On the other hand, Mackinlay et al. reported that YSLCs themselves express BMP antagonists (Mackinlay et al., 2021). How much of the downstream SMADs phosphorylated in YSLCs under the control and LDN-treated conditions?

We thank the reviewer for this interesting question. Our observation in the embryo that BMP signaling appears necessary for maintenance, but not specification, of the CER1-positive domain indicates that hypoblast cells are engaged in BMP signaling despite the presence of CER1 and other BMP antagonists. This is similar to the established role of NODAL in the mouse anterior visceral endoderm^{29, 30}, which also appears conserved in the human based on the present data. NODAL signaling is required for the formation of these cells, and in turn upregulates NODAL antagonists CER1 and LEFTY1^{4, 30-32}.

We have now assessed the presence of pSMAD1.5.9 in YSLCs (Extended Data Fig. 7q) and include this data in Response to Reviewers Figure 1.10. We do see pSMAD1.5.9 expression in GATA4-positive YSLCs, and this is diminished with LDN193189 treatment. We have also now added experiments where we use conditioned media from YSLCs differentiated with or without LDN193189 and show that the addition of LDN193189 during induction reduces the capacity of YSLCs to protect epiblast-like spheroids from BMP signals, demonstrating an important role for BMP signaling in anterior hypoblast specification, at least in yolk sac-like cells.

Response to Reviewers Figure 1.10: BMP signaling is active in YSLCs and important for specifying an anterior hypoblast-like identity. (A) Immunofluorescence images and quantification of normalized phosphorylated (p)SMAD1.5.9 in yolk sac-like cells (YSLCs) differentiated with or without 200nM LDN193189 (LDN) for 48h from day 4-6. N = 1150 (control) and 827 (LDN) cells across 2 experiments. (B) Schematic of conditioned media (CM) experimental design. (C) Immunofluorescence and quantification showing pSMAD1.5.9 expression in hESC-derived spheroids cultured in mTeSR+ media (mTeSR+; 1060 cells), mTeSR+ media conditioned on either control YSLC (YSLC CM; 1836 cells), or YSLC differentiated with 48 hours of LDN treatment (YSLC+LDN CM; 1427 cells). N=3 experiments. Statistics: (A) unpaired T-test; (C) One way ANOVA with Tukey Kramer post-hoc. ****p<0.0001. Scale bars: 50µm.

13. In the naive pluripotent stem cells in the PXGL media, the LDN treatment increased cleaved-Cas3-positive, i.e., apoptotic, cells. However, according to a previous report (Io et al., 2021), the SMAD1/5/8 of hPSCs in PXGL media are not phosphorylated. This may suggest that the effect of LDN may be a side effect. Can the authors clarify this point?

In Figure S1K of Io et al., 2021 (reproduced below), the authors provide a western blot demonstrating transient upregulation of pSMAD1.5, with a peak at nTE_D1 (in line with the addition of BMP4 in that protocol). However, there is clearly low level pSMAD1.5 in both the primed and naïve cells of the blot - higher than that of nCT cells. Therefore, we do not believe that the data presented in Io et al. is conclusive on whether there is *no* pSMAD1.5 in naïve human ESCs; rather, they show that the BMP4 addition during nTE differentiation leads to a higher level

of pSMAD1.5 expression. Indeed, the authors of this study refer to this result as a ‘transient upregulation.’

To answer this question definitively we have analyzed pSMAD1.5.9 expression in PXGL and primed cells, which does demonstrate a relative enrichment of pSMAD1.5.9 in the naïve state of human ESC compared to the primed state (included as Fig. 5j-k and below). Taken together, these data indicate that BMP signaling may indeed be active in PXGL cultured, naïve human ESC.

Response to Reviewers Figure 1.11: pSMAD1.5.9 is expressed in naïve hESC. (A) Reproduction of Figure S1K from Io et al., 2021² showing a western blot of pSMAD1/5 in primed and naïve hESC, as well as during trophectoderm differentiation (nTE) protocol. (B) Immunofluorescence and quantification of pSMAD1.5.9 in naïve (3057 cells) and primed (5761 cells) hESC. N=2 experiments. Statistics: (B) unpaired T-test. ****p<0.0001. Scale bars: 50µm.

14. In all analyses using signaling agonists/antagonists, the authors only discussed the appearance of cells expressing some key genes. These are informative experiments, but in my opinion not sufficient to claim that the mechanisms for the symmetry breaking of the disc-shape human embryos is clarified.

We agree with the reviewer that elucidating mechanism in the human embryo is a daunting challenge. We hope our work sheds light on the role of NODAL and BMP in anterior hypoblast formation. However, we accept that this is insufficient to claim that the mechanism of human anterior-posterior symmetry breaking is resolved through this study. We have therefore toned-down relevant statements in the manuscript.

Minor points:

1. In line #110, the authors annotated the syncytiotrophoblast and extravillous trophoblast by the expression of SDC1 and CGB1, however, no references were cited.

We apologize for not citing the appropriate papers. Please note that the annotation of clusters was not based only on the genes plotted, but also on other differentially expressed genes, which are all provided as Extended Data Tables.

These are now cited in the text:

Okae, H. et al. Derivation of human trophoblast stem cells. *Cell Stem Cell* **22**, 50–63.e6 (2018)³³.

Weatherbee, Weberling, & Gantner, et al.; Response to Reviewers

Lv, B. et al. Single-cell RNA sequencing reveals regulatory mechanism for trophoblast cell-fate divergence in human peri-implantation conceptuses. *PLoS Biol.* **17**, e3000187 (2019)³⁴.

West, R. C. et al. Dynamics of trophoblast differentiation in peri-implantation-stage human embryos. *Proc. Natl Acad. Sci. U.S.A.* **116**, 22635–22644 (2019)³⁵.

Ruane, PT, et al. Trophoblast differentiation to invasive syncytiotrophoblast is promoted by endometrial epithelial cells during human embryo implantation. *Hum Reprod.* **37**(4):777-792 (2022)³⁶.

Molè, M.A. et al. A single cell characterisation of human embryogenesis identifies pluripotency transitions and putative anterior hypoblast centre. *Nat Commun* **12**, 3679 (2021)⁴.

2. I could not access the analysis code. The authors should provide appropriate information for their computational analysis so that the readers can reproduce the authors' analysis.

We apologize for this omission. This repository is now public. The Seurat objects are too large to upload to github, however, we have reserved the DOI: 10.5281/zenodo.7689580 where we will include uploads of the Seurat Objects upon publication.

Reviewer #2

In this study, Weberling et al. first performed bioinformatics analysis of signaling pathways in mouse, cynomolgus macaque, and human embryos by using single cell RNAseq datasets publically available. They showed that NODAL and BMP signaling are enriched in the human epiblast at pre-implantation compared to post-implantation stages. By using human embryo cultures and embryonic stem cell models, the authors validated the role of NODAL and BMP in hypoblast specification and peri-implantation epiblast survival in vitro. While BMP signaling specifies anterior hypoblast identity in human, it inhibits anterior hypoblast in the mouse. In the mouse embryo, NODAL signaling is required to specify the anterior hypoblast. Lastly, by using in vitro human embryo culture with or without NOTCH inhibitor DAPT treatment, the authors showed that the NOTCH signaling is more important for survival of epiblasts at postimplantation stages but not preimplantation.

Major Critiques:

1. A recent research paper in Dev Cell also showed similar bioinformatics analysis and further demo the importance of NODAL signaling in mouse and human anterior-posterior formation. (Zhu et al. 2023 Jan 9; doi: 10.1016/j.devcel.2022.12.004. Decoding anterior-posterior axis emergence among mouse, monkey, and human embryos). The novelty of current paper would be on the BMP signaling in hypoblast specification and survival as well as the different role for BMP pathways in human and mouse early embryo development. The authors shall consider to substantially revise their manuscript by focusing on the novel role of BMP in anterior-posterior axis.

We thank the reviewer for their comment and agree that the paper by Zhu et al.³⁷ is interesting. However, we note several important differences between our work and theirs: (1) Zhu et al. show that NODAL signaling-related genes NODAL, LEFTY1 and LEFTY2 expression delineates the 'AVE' but this is not equivalent to showing the functional importance of NODAL signaling (e.g. these genes may be important for epiblast function not hypoblast specification); (2) They perform no experimentation on human and/or primate embryos; and (3) Zhu et al. do not assess the role of NODAL signaling in tissues other than the hypoblast. Taken together, we do not agree that this paper in any way diminishes the importance of our work.

2. In this study, the authors mainly used left-over IVF embryo cultures to examine the perturbation of NODAL, BMP, and NOTCH signaling on epiblast and hypoblast development. However, their results may be complicated by the uneven quality of IVF embryos during in vitro cultures. The generation of human embryoids and gastruloids from well-characterized human stem cells has provided a well-controlled system to study the patterning of anterior-posterior axis (e.g. doi.org/10.1038/s41586-020-2383-9). The authors have previously generated embryoids and gastruloids. The authors shall test the effect of compounds affecting BMP, NODAL, and Notch signaling during the time course of embryoid and gastruloid formation. These new data would allow authors to make more convincing conclusions.

We agree that IVF embryos exhibit heterogeneity and that this may confound our results. Indeed, this is reflected by the large spread of data in our experiments. For this reason, in any inhibitor/agonist experiments we have included all embryos treated as exclusion criteria cannot be uniformly applied in cases of perturbation experiments. In contrast, in experiments where we seek to understand the 'normal' embryo (for example SMAD2.3 expression) we only assess embryos that contain all three lineages (epiblast, hypoblast and trophectoderm). It is crucial to

note, however, that we are limited by the quality of embryos donated to research, and many may harbor abnormal. To minimize this risk, we only carry forward embryos into experiments that morphologically appear healthy (i.e. a large expanded blastocoel with clearly visible ICM). We note this consideration and others in the ‘*Study Limitations and Considerations*’ subsection.

The reviewer’s suggestion to utilize embryoid/gastruloid models to aid in generating a mechanistic understanding is extremely interesting. However, previously published human embryoid/gastruloid models do not mimic the peri-implantation period (e.g. blastoids only model pre-implantation and lack the anterior hypoblast population; gastruloids, including those in the paper linked by the reviewer, model later, peri-gastrulation-stage development and lack any extraembryonic tissues). Neither model captures the formation of the anterior hypoblast or naïve-to-primed transition of the epiblast, which are the crucial aspects of this study.

We have, during this revision process, reported a model that does include CER1-positive cells in a multi-lineage context. However, in the interest of reproducibility and isolating the effects of tissue-specific pathway inhibition (a crucial advantage of stem cell systems), we have utilized an assay we have previously reported¹.

Response to Reviewers Figure 2.1: NODAL, BMP or NOTCH inhibition functionally decrease secreted YSLC signaling antagonists. (A) Schematic of conditioned media (CM) experimental design. (B-D) Immunofluorescence and quantification showing pSMAD1.5.9 expression in hESC-derived spheroids cultured in mTeSR+ media (mTeSR+; 1060 cells), media conditioned on control YSLC (YSLC CM; 1836 cells), or media conditioned on YSLC differentiated with 48 hours of either A83-01 (B; YSLC+A83 CM; 1654 cells), LDN (C; YSLC+LDN CM; 1427 cells), or DAPT (D; YSLC+DAPT CM; 501 cells) treatment. N=3 experiments. Statistics: (B-D) One way ANOVA with Tukey Kramer post-hoc. ****p<0.0001. Scale bars: 50µm.

In this assay, media is conditioned on our previously published yolk sac-like cells (YSLCs), which normally express antagonists similar to the anterior hypoblast¹. For these experiments, YSLC were differentiated with or without inhibitors of BMP, NODAL and NOTCH, and then used to generate conditioned media. This conditioned media was then used in generation of human ESC spheroids, which model the peri-implantation epiblast^{38, 39} (Response to Reviewers Fig. 2.1 above and Figs. 3n-o, 5o-p, and 6n-o). These experiments demonstrate that the loss of anterior hypoblast identity after BMP/NODAL/NOTCH inhibition affects capacity of *in vitro* YSLCs to protect epiblast-like spheroids from BMP signaling, as reflected by increased levels of

pSMAD1.5.9 in the spheroids. This data is in line with the role of these signaling pathways in anterior hypoblast specification and/or maintenance directly observed in the embryo.

3. The effect of NOTCH signaling inhibitor DAPT on the survival of D7-9 post-implantation epiblast is a bit striking to this reviewer. I would suggest that the authors may try out another well-established gamma-secretase inhibitor such as MK0752 and see if the result is consistent or not. Alternatively, a separate method such as NOTCH receptor knock-down in D7-9 embryo cultures would be more helpful.

We agree with the reviewers that these results were striking. We have now added experimentation with two additional γ -secretase inhibitors: MK-0752 and Compound-E and include this data in Fig. 6a-h and Extended Data Fig. 8a-b and below in Response to Reviewers Figure 2.2.

Response to Reviewers Figure 2.2: γ -secretase inhibition results in post-implantation epiblast and anterior hypoblast loss. (A) Immunofluorescence images of embryos cultured from day 5-7 in control conditions (N=36 embryos), 20 μ M DAPT (N=8 embryos), 10 μ M Compound-E (N=14 embryos), or 20 μ M MK-0752 (N=9 embryos). (B-D), Quantification of the number of epiblast (B), total hypoblast (C), and CER1-positive hypoblast cells (D) from day 5-7. (E) Visualization of the Bayesian statistical analysis for counts presented in B-D. Coefficient estimated by brms R package for divergence from control where a coefficient with credible interval (CI) above 0.33 indicates a positive shift in the distribution, while below -0.33 indicates a negative shift. (F) Immunofluorescence images of embryos cultured from day 7-9 in control conditions (N=31 embryos), 20 μ M DAPT (N=9 embryos), 10 μ M Compound-E (N=12 embryos), or 20 μ M MK-0752 (N=17 embryos). (G-I) Quantification of the number of epiblast (G), total hypoblast (H), and CER1-positive hypoblast cells (I) from day 7-9. (J) Visualization of the Bayesian statistical analysis for the counts presented in G-I. Statistics: (B-D, G-I) Mann-Whitney test with p-value<0.06 noted. ****p<0.0001, ***p<0.001, **p<0.01, *p<0.05. Scale bars: 100 μ m.

These inhibitors showed some variation in terms of effect compared to DAPT treatment; however, all three inhibitors showed a decrease in post-implantation epiblast numbers, albeit to different

Weatherbee, Weberling, & Gantner, et al.; Response to Reviewers

extents, and all three decreased anterior hypoblast numbers, at varying stages. Importantly, neither MK-0752 nor Compound-E treated embryos from day 7-9 exhibited a decrease in hypoblast numbers, in contrast to our previous results with DAPT. This, together with the DAPT causing the starkest decrease in epiblast cells, suggests that the hypoblast loss observed previously may have been a result of the epiblast death and subsequent loss of tissue-tissue cross-talk rather than a direct effect of γ -secretase inhibition on the hypoblast, directly. We have updated our conclusions based on these results accordingly.

4. The paper may be revised more concisely for take-home messages.

We appreciate the suggestion by the reviewer and have revised the text accordingly.

Reviewer #3

The ethics statement informs me that this was reviewed and approved by the HFEA. The statement also notes that donors were given information about the project and about the ability to withdraw the embryos. What is missing from the description is anything about how the particular donors were recruited and chosen; the information they were given about other options for embryo management (e.g. freezing or disposal without donation to research); and whether the donors were the progenitors (ie, were the sperm and egg providers the same people as those making the decision to donate, or were anonymous or known gamete donors being used by a couple in which one or both partners could not provide a gamete).

We appreciate the reviewer's comments and the opportunity to provide further information on the embryo donations. We have revised the ethics statement accordingly and include the relevant section below. Please note that this now includes specific information about how patients were recruited and their options for embryo management. In addition, we also clarify that the choice to donate and consent is obtained from both gamete providers. We note that all specific information regarding embryo donation is addressed by the HFEA Code of Practice.

The updated statement in the Methods section is as follows:

<< Human embryo work was regulated by the Human Fertility and Embryology Authority (HFEA) under license R0193. Approval was obtained from the Human Biology Research Ethics Committee at the University of Cambridge (reference HBREC.2021.26). Patients undergoing IVF at CARE Fertility, Bourn Hall Fertility Clinic, Herts & Essex Fertility Clinic and King's Fertility were given the option of continued storage, disposal, or donation of embryos to research (including project specific information) or training at the end of their treatment. Patients were offered counseling, received no financial benefit and could withdraw their participation at any time until the embryo had been used for research. Research consent for donated embryos was obtained from both gamete providers. Embryos were not cultured beyond day 14 post-fertilization or the appearance of the primitive streak.>>

Reviewer #4

How the anterior-posterior axis is specified in post-implantation human embryos remain unclear due to the inaccessibility of human embryonic development at this stage. Weberling et al. examined the signaling pathways active in this key developmental event in human, monkey and mouse embryos using public scRNA-seq datasets, and interrogated the function of key signaling pathways using human and mouse *in vitro* models, including *in vitro* cultured human embryos beyond implantation. They uncovered both conserved and species-specific requirements for NODAL and BMP signaling in specifying and maintaining anterior hypoblast, and found that NOTCH signaling is required to maintain the epiblast and hypoblast in human embryos after implantation.

Overall, this study aimed to address a key question in human embryo development and put forth substantial efforts to understand the functions of key signaling pathways in specifying human anterior-posterior axis. Such investigations were now possible thanks to recent advance in *in vitro* culture of human embryos through implantation. Both conserved and species-specific regulation between human and mouse were identified. However, there are several issues that should be addressed as illustrated below.

Major comments:

1. The effects of specific signaling pathway inhibition on human embryos were mostly examined using IF. Some of them were however somewhat difficult to evaluate without proper quantification (see below for examples). Single-cell RNA sequencing could be one way to validate these results and conclusions, and further shed light on the underlying mechanism.

We thank the reviewer for their suggestion and agree that a novel sequencing dataset may be helpful to further validate our conclusions. Unfortunately, in our experience it often takes up to 20 embryos per timepoint per condition to generate meaningfully large single cell datasets and we feel this is unfortunately not feasible given limited access to these embryos. In particular, as we are looking at very small populations (e.g. the CER1+ anterior hypoblast) we do not believe we could generate appropriate sequencing datasets given the large number of embryos that would need to be sequenced. For this reason, we have sought to use immunofluorescence as it allows us to specifically assess the effect of perturbation on the anterior hypoblast population and thus provide specific functional insight, albeit only at the level of several proteins.

We have now added more functional assessment of the defects in the differentiation of anterior hypoblast-like YSLCs *in vitro* through the use of conditioned media experiments. In addition to quantifying a decrease in CER1 expression when NODAL, BMP or NOTCH signaling is inhibited during the differentiation of these cells – as in the embryo itself - we now show that their capacity to secrete signaling antagonists that protect epiblast-like spheroids from differentiation cues (BMP) is compromised after NODAL, BMP or NOTCH inhibition (Response to Reviewers Fig. 4.1). These experiments clearly indicate that the effect of inhibition is tissue-specific and not a result of tissue-tissue crosstalk in the embryo.

Response to Reviewers Figure 4.1: NODAL, BMP or NOTCH inhibition functionally decrease secreted YSLC signaling antagonists. (A) Schematic of conditioned media (CM) experimental design. **(B-D)** Immunofluorescence and quantification showing pSMAD1.5.9 expression in hESC-derived spheroids cultured in mTeSR+ media (mTeSR+; 1060 cells), media conditioned on control YSLC (YSLC CM; 1836 cells), or media conditioned on YSLC differentiated with 48 hours of either A83-01 **(B)**; YSLC+A83 CM; 1654 cells), LDN **(C)**; YSLC+LDN CM; 1427 cells), or DAPT **(D)**; YSLC+DAPT CM; 501 cells) treatment. N=3 experiments. Statistics: **(B-D)** One-way ANOVA with Tukey Kramer post-hoc. **** $p < 0.0001$. Scale bars: 50µm.

We also now provided more insight regarding the divergence between mouse and human and stage-dependent requirement of BMP signaling in the epiblast (Figs. 3n-o, 5o-p, and 6n-o; included below as Response to Reviewers Figure 4.4).

2. To quantify the activity of NODAL, the authors used the n/c ratio of total SMAD2.3 in individual cells. This could be misleading, especially when both signals in the nucleus and cytoplasm are high, but the ratio is low. For example, the authors stated “In segregating blastocysts, the n/c SMAD2.3 ratio was significantly higher in trophectoderm cells than inner cell mass cells, similar to the E3.5 early mouse blastocyst (Figure 2B-C, Extended Figure 4D-E) “. However, I cannot see a higher n/c SMAD2.3 ratio in trophectoderm cells from IF results. Moreover, the signal of SMAD2.3 seems higher in ICM/EPI/HYPO than that in TE. The authors should show the original signals in the nucleus and cytoplasm side by side.

We thank the reviewer for their suggestion and now provide both n/c ratios and individual nuclear and cytoplasmic results, and also provide the updated plots below for convenience. These new inclusions more clearly reflect the spread of SMAD2.3 expression in the developing embryo. For example, in the segregating blastocyst (i.e. with ICM cells co-expressing hypoblast and epiblast markers; Response to Reviewers Fig 4.2A), the total SMAD2.3 level is decreased in the trophectoderm for both nuclear and cytoplasmic locations, despite the nuclear/cytoplasmic ratio being higher.

Response to Reviewers Figure 4.2: Nuclear, cytoplasmic, and nuclear-to-cytoplasmic ratio quantification of SMAD2.3 in human embryos. (A-D) SMAD2.3 fluorescence in nuclear (nuc; left) or cytoplasmic (cyto; middle) regions of interest plotted on the left axis, and nuclear-to-cytoplasmic ratio (nuc/cyto) plotted in segregating (A) or segregated (B) blastocysts and day 7 (C) and day 9 (D) post-fertilization (d.p.f.). ICM=inner cell mass, EPI=epiblast, HYPO=hypoblast, TE=trophectoderm. Statistics: (A, C-D) Unpaired T-test; (B) One-way ANOVA with Tukey-Kramer post-hoc. ****p<0.0001, ***p<0.001, *p<0.05.

3. In fact, the activity of NODAL signaling is often evaluated by pSMAD2/3. In addition, the authors quantified BMP signaling by pSMAD1/5/9 (or pSMAD1/5/8). It is unclear why the authors used different strategies (n/c SMAD2.3) in these cases.

We agree that the use of antibodies specifically targeting the phosphorylated form of the SMAD2.3 would be preferable. However, in our hands we have not been able to convincingly demonstrate meaningful staining using a pSMAD2.3 antibody. Given the limited access to human embryos in research, we validated both antibodies and inhibitors in human ESCs (see Extended Data Figs. 5a-b and 7b-c for these results and Response to Reviewers Fig 4.3 below) and only proceeded to stain embryos with antibodies that demonstrated robust response to activation and/or inhibition.

Response to Reviewers Figure 4.3: Validation of total SMAD2.3 antibody. (A-B) Immunofluorescence images and quantification of primed hESCs cultured with either DMSO (n=2588), 50ng/mL Activin A (n=2465), or Activin-A with 2µM A83-01 (n=2463) for 48h. Note the strong nuclear shuttling of SMAD2.3 after Activin-A treatment, which could be blocked by simultaneous addition of the ALK4/5/7 inhibitor A83-01. Statistics: (B) One-way ANOVA with Tukey Kramer post-hoc. ****p<0.0001.

4. The differences in the requirement of BMP4 in pre-implantation epiblast survival in human and mouse are intriguing. Can the authors find any possible mechanisms underlying such species-specific divergence based on the scRNA-seq data?

We agree with the reviewer that this result is extremely interesting and thank them for this suggestion. We and others have demonstrated an important role for BMP signaling in the pre- and peri-implantation mouse embryo but only in the context of extraembryonic tissues. BMP4 is expressed in the pre-implantation human embryo⁴⁰, indicating that it may have a role in pre-implantation epiblast development. Indeed, De Paepe et al., (2019)⁴⁰ demonstrate that exogenous addition of BMP4 does not negatively regulate NANOG expression levels, as might be expected if it was driving differentiation.

To add mechanistic insight, we have now performed a correlation analysis with BMP response/effector ID genes⁴¹. This has demonstrated a differential relationship between ID genes and naïve genes^{4, 42-44} between mouse and human, suggesting an evolutionary divergence in the role of BMP in regulating pluripotency state, with ID gene expression (indicative of active BMP signaling) correlating with expression of several human naïve epiblast markers, including *TBX3*, *DPPA3*, *DNMT3L*, *TFAP2C*, *ZFP42* and *TFCP2L1* (Fig. 5i and Extended Data Fig. 7n; included as Response to Reviewers Figure 4.4A).

Response to Reviewers Figure 4.4: BMP signaling differentially regulates naïve pluripotency in human and mouse. (A) Pearson regression correlation coefficients for single cell RNA sequencing-based expression of BMP response genes *ID1-3* with markers of human or mouse naïve pluripotency at the blastocyst stage. (B) Immunofluorescence and quantification of pSMAD1.5.9 in naïve (3057 cells) and primed (5761 cells) hESC. N=2 experiments. (C) Immunofluorescence and quantification of pSmad1.5.9 in mESCs in 2iLif conditions (naïve/2iLif; 297 cells) or with factors withdrawn (naïve exit/N2B27; 789 cells). N=2 experiments. (D) Immunofluorescence and quantification of human naïve pluripotency marker AP2γ in PXGL naïve ESCs transitioned to N2B27 ± LDN193189 (1695 untreated cells; 1426 LDN treated cells). N=2 experiments. (E) Immunofluorescence and quantification of mouse primed pluripotency marker Otx2 in mESCs cultured in either 2iLif naïve conditions (1570 cells), N2B27 (3602 cells), or N2B27+LDN (1064 cells). N=3 experiments. Statistics: (A) Coefficients were tested (cor.test; two-tailed) and corrected for multiple hypothesis testing with the Benjamini-Hochberg method; (B-D) unpaired T-test; (E) One-way ANOVA with Tukey Kramer post-hoc. ****p<0.0001, ***p<0.001, **p<0.01, *p<0.05. Scale bars: 50µm.

We also provide further evidence of the role of BMP in naïve and primed human and mouse ESCs stained for pSMAD1.5.9. pSMAD1.5.9 expression is enriched in naïve compared to primed human ESC and this pattern is reversed in mouse and this pattern appears functionally important. BMP inhibition decreases expression of the human-specific naïve human ESC marker AP2γ⁴⁵. In contrast, BMP inhibition of mouse ESC exiting the naïve pluripotency state demonstrates a role of BMP signaling in upregulating the primed-specific marker OTX2 (Fig. 5j-k and Extended Data Fig. 7 o-p; included as Response to Reviewers Figure 4.4B-E). Taken together, BMP signaling appears important to human naïve pluripotency in both stem cell analogue of the pre-implantation epiblast, as well as in the embryonic epiblast itself.

5. Some statements were overstated in the paper. For example, the authors stated that “The NODAL module was noticeably enriched in the epiblast at the blastocyst stage in human and macaque”. But NODAL module is much more enriched in peri-implantation epiblast in monkey (Fig. S1E). In addition, “restricted” was frequently used in this manuscript, but was misleading in some occasions. For example, BMP is not restricted to hypoblast, but also enriched in EXMC in monkey (Fig. S1E). NODAL is not restricted to, but only relatively more enriched in the hypoblast in human. The authors should tone down their statements and describe the results/conclusions accurately.

We thank the reviewer for their comment. We have now adjusted our conclusions to reflect the data more accurately and with greater nuance.

6. In Figure 4L, why activin A at a relative low concentrate can induce apoptosis of primed hESCs, given it was shown to facilitate the maintenance of hESCs at a primed state (James et al., Development, 2005; Beattie et al., Stem Cells, 2005)?

While primed human ESCs do rely on Activin-A for maintenance, we speculate that it is likely a dosage dependent response. Indeed, recent work has demonstrated that NODAL signaling is enriched in naïve human ESC and required for consolidation of naïve identity⁴⁶.

Another possibility is that exogenous application of Activin-A drives differentiation toward mesendodermal fates (while A83-01 treated cells follow the ‘default’ neurectoderm specification trajectory) and this is likely associated with increased cell death during the cell fate transition⁴⁷.

However, to streamline the data presented in the manuscript and focus our analyses on phenotypes relevant to our observations in the embryo, we have removed *in vitro* experiments for tissue analogues which were unaffected by perturbation in the embryo. This includes the data referenced here.

7. The authors stated that “pSMAD1.5.9 expression decreased over time in the human epiblast, with a distinct bimodal intensity pattern at D9”. However, I did not see the bimodality of pSMAD1.5.9 in Figure 3E.

In an effort to remove overstatements from the text, we have removed this statement.

8. Figure 2J. Could the authors provide an IF figure?

We now provide immunofluorescence panels showing high-power examples of contacting or non-contacting hypoblast cells and their SMAD2.3 expression, as well as the equivalent images for pSMAD1.5.9 in Fig. 4k. Note the position of these cells in the embryo is denoted by the boxes in the day 9 embryos on Figs. 2 and 4.

Response to Reviewer Figure 4.5: Addition of high-power examples of hypoblast cells contacting or not contacting the epiblast. (A) Immunofluorescence images of day 9 (N=13 embryos) human embryos following *in vitro* implantation stained for OCT4/PODLX, GATA6, SMAD2.3 and DAPI. (B) Quantification of nuclear/cytoplasmic ratio of SMAD2.3 in hypoblast cells at day 7 and day 9 which are in contact (i.e. directly adjacent to the epiblast) or not in contact with epiblast. (C) High power images of regions boxed and labeled in (A) of hypoblast cells that are either contacting (i) or not contacting (ii) the epiblast at day 9. (D) Immunofluorescence images of day 9 (N=13 embryos) human embryos following *in vitro* implantation stained for SOX2/PODLX, GATA6, pSMAD1.5.9 and DAPI. (E) Quantification of normalized pSMAD1.5.9 in hypoblast cells at day 7 and day 9 either contacting or not contacting the epiblast. (F) High power images of regions boxed and labeled in (D) of hypoblast cells that are either contacting (i) or not contacting (ii) the epiblast at day 9. For Violin plots, central dotted line denotes median which is between dotted lines marking the 25th and 75th quartiles. Statistics: (B, E) One-way ANOVA with Tukey-Kramer post-hoc. ***p<0.001, **p<0.01, *p<0.05. Scale bars: (A, D) 100µm, (C, F) 20µm.

9. The model could more informative by including the functional information of these signaling pathways.

We thank the reviewer for their suggestion and now include a new summarized functional data diagram in addition to the characterization of each signaling pathway. This new model focuses on the formation of the human anterior hypoblast and mouse DVE/AVE. The final model is also included below for convenience.

Response to Reviewers Figure 4.6: Proposed model of peri-implantation signaling activity and anterior specification.

(A) Proposed model of BMP, NODAL and NOTCH signaling activity in the Hypoblast (top) and Epiblast (bottom) during human peri-implantation. Note implantation as a switch point in the activity of pathways for the epiblast, and a point of divergence of subpopulations with differing activity in hypoblast. The role of NOTCH in the hypoblast remains less clear, indicated by a gradient. (B) Summary of signaling perturbation effects on human anterior hypoblast (top) and mouse distal/anterior visceral endoderm (DVE/AVE; bottom) specification. Schematics of human embryos correspond to day 5, day 7, and day 9 and mouse embryo schematics correspond to E3.5, E5.0, E5.75.

Minor comments:

1. Extended Figure 4. The description in the manuscript and the figure index did not match. Fig. S4D is missing.

We thank the reviewer for pointing this out and have now fixed this.

2. Some references are inappropriately cited. For example, ref. 46 was not mainly about the nuclear effectors of BMP signaling.

We agree that the paper by Yoney et al.⁴⁸ does not focus predominantly on the nuclear effectors of BMP signaling. However, it contains several pertinent examples of nuclear shuttling of SMAD effectors in response to both Activin-A and BMP, which we feel is relevant to our system, particularly as this work was carried out in human ESCs.

3. To easily reproduce the analysis shown in this work, I suggest that the authors publicly provide the processed data (given all the raw scRNA-seq data were published) (i.e., Seurat object). I also cannot see the source code at <https://github.com/bweatherbee/PeriImplantation>.

We thank the reviewer for their comment and apologize that the repository was not public at the time of review. This repository is now public. The Seurat objects are too large to upload to github, however, we have reserved the DOI: 10.5281/zenodo.7689580 where we will include uploads of the Seurat Objects upon publication.

References

1. Mackinlay, K.M.L. *et al.* An in vitro stem cell model of human epiblast and yolk sac interaction. *eLife* **10** (2021).
2. Io, S. *et al.* Capturing human trophoblast development with naive pluripotent stem cells in vitro. *Cell Stem Cell* **28**, 1023-1039.e1013 (2021).
3. Xiang, L. *et al.* A developmental landscape of 3D-cultured human pre-gastrulation embryos. *Nature* **577**, 537-542 (2020).
4. Molè, M.A. *et al.* A single cell characterisation of human embryogenesis identifies pluripotency transitions and putative anterior hypoblast centre. *Nature Communications* **12** (2021).
5. Chhabra, S. & Warmflash, A. BMP-treated human embryonic stem cells transcriptionally resemble amnion cells in the monkey embryo. *Biology Open* **10** (2021).
6. Stuart, T. *et al.* Comprehensive Integration of Single-Cell Data. *Cell* **177**, 1888-1902.e1821 (2019).
7. Hafemeister, C. & Satija, R. Normalization and variance stabilization of single-cell RNA-seq data using regularized negative binomial regression. *Genome Biol* **20**, 296 (2019).
8. Pham, T.X.A. *et al.* Modeling human extraembryonic mesoderm cells using naive pluripotent stem cells. *Cell Stem Cell* **29**, 1346-1365.e1310 (2022).
9. Zhou, F. *et al.* Reconstituting the transcriptome and DNA methylome landscapes of human implantation. *Nature* **572**, 660-664 (2019).
10. Deglincerti, A. *et al.* Self-organization of the in vitro attached human embryo. *Nature* **533**, 251-254 (2016).
11. Shahbazi, M.N. *et al.* Self-organization of the human embryo in the absence of maternal tissues. *Nature Cell Biology* **18**, 700-708 (2016).
12. Nakamura, T. *et al.* A developmental coordinate of pluripotency among mice, monkeys and humans. *Nature* **537**, 57-62 (2016).
13. Sasaki, K. *et al.* The Germ Cell Fate of Cynomolgus Monkeys Is Specified in the Nascent Amnion. *Developmental Cell* **39**, 169-185 (2016).
14. Bergmann, S. *et al.* Spatial profiling of early primate gastrulation in utero. *Nature* (2022).
15. Yang, R. *et al.* Amnion signals are essential for mesoderm formation in primates. *Nature Communications* **12** (2021).
16. Chhabra, S., Liu, L., Goh, R., Kong, X. & Warmflash, A. Dissecting the dynamics of signaling events in the BMP, WNT, and NODAL cascade during self-organized fate patterning in human gastruloids. *PLoS Biology* **17** (2019).
17. Hackenberger, B.K. Bayes or not Bayes, is this the question? *Croat Med J* **60**, 50-52 (2019).
18. Corujo-Simon, E., Radley, A.H. & Nichols, J. Evidence implicating sequential commitment of the founder lineages in the human blastocyst by order of hypoblast gene activation. *Development* **150** (2023).
19. Meistermann, D. *et al.* Integrated pseudotime analysis of human pre-implantation embryo single-cell transcriptomes reveals the dynamics of lineage specification. *Cell Stem Cell* **28**, 1625-1640.e1626 (2021).
20. Petropoulos, S. *et al.* Single-Cell RNA-Seq Reveals Lineage and X Chromosome Dynamics in Human Preimplantation Embryos. *Cell* **165**, 1012-1026 (2016).
21. Stirparo, G.G. *et al.* Integrated analysis of single-cell embryo data yields a unified transcriptome signature for the human pre-implantation epiblast. *Development (Cambridge)* **145** (2018).
22. Guo, G. *et al.* Human naive epiblast cells possess unrestricted lineage potential. *Cell Stem Cell* **28**, 1040-1056.e1046 (2021).
23. Niakan, K.K. & Eggan, K. Analysis of human embryos from zygote to blastocyst reveals distinct gene expression patterns relative to the mouse. *Developmental Biology* **375**, 54-64 (2013).
24. Kagawa, H. *et al.* Human blastoids model blastocyst development and implantation. *Nature* **601**, 600-605 (2022).
25. Zhu, M. *et al.* Human embryo polarization requires PLC signaling to mediate trophectoderm specification. *eLife* (2021).
26. Gerri, C. *et al.* Initiation of a conserved trophectoderm program in human, cow and mouse embryos. *Nature* **587**, 443-447 (2020).
27. Linneberg-Agerholm, M. *et al.* Naïve human pluripotent stem cells respond to Wnt, Nodal and LIF signalling to produce expandable naïve extra-embryonic endoderm. *Development (Cambridge)* **146** (2019).
28. Weatherbee, B.A.T. *et al.* A model of the post-implantation human embryo derived from pluripotent stem cells. *Nature* (2023).
29. Takaoka, K., Nishimura, H. & Hamada, H. Both Nodal signalling and stochasticity select for prospective distal visceral endoderm in mouse embryos. *Nature Communications* **8** (2017).
30. Rodriguez, T.A., Srinivas, S., Clements, M.P., Smith, J.C. & Beddington, R.S.P. Induction and migration of the anterior visceral endoderm is regulated by the extra-embryonic ectoderm. *Development* **132**, 2513-2520 (2005).
31. Belo, J.A. *et al.* Cerberus-like is a secreted factor with neuralizing activity expressed in the anterior primitive endoderm of the mouse gastrula. *Mechanisms of Development* **68**, 45-57 (1997).
32. Morris, S.A. *et al.* Dynamics of anterior–posterior axis formation in the developing mouse embryo. *Nature Communications* **3**, 673-673 (2012).

Weatherbee, Weberling, & Gantner, et al.; Response to Reviewers

33. Okae, H. *et al.* Derivation of Human Trophoblast Stem Cells. *Cell Stem Cell* **22**, 50-63.e56 (2018).
34. Lv, B. *et al.* Single-cell RNA sequencing reveals regulatory mechanism for trophoblast cell-fate divergence in human peri-implantation conceptuses. *PLoS Biology* **17** (2019).
35. West, R.C. *et al.* Dynamics of trophoblast differentiation in peri-implantation-stage human embryos. *Proceedings of the National Academy of Sciences of the United States of America* **116**, 22635-22644 (2019).
36. Ruane, P.T. *et al.* Trophectoderm differentiation to invasive syncytiotrophoblast is promoted by endometrial epithelial cells during human embryo implantation. *Human Reproduction* **37**, 777-792 (2022).
37. Zhu, Q. *et al.* Decoding anterior-posterior axis emergence among mouse, monkey, and human embryos. *Dev Cell* **58**, 63-79 e64 (2023).
38. Shahbazi, M.N. *et al.* Pluripotent state transitions coordinate morphogenesis in mouse and human embryos. *Nature* **552**, 239-243 (2017).
39. Simunovic, M. *et al.* A 3D model of a human epiblast reveals BMP4-driven symmetry breaking. *Nature Cell Biology* **21**, 900-910 (2019).
40. De Paepe, C. *et al.* BMP4 plays a role in apoptosis during human preimplantation development. *Mol Reprod Dev* **86**, 53-62 (2019).
41. Hollnagel, A., Oehlmann, V., Heymer, J., Rütter, U. & Nordheim, A. Id genes are direct targets of bone morphogenetic protein induction in embryonic stem cells. *Journal of Biological Chemistry* **274**, 19838-19845 (1999).
42. Athanasouli, P. *et al.* The Wnt/TCF7L1 transcriptional repressor axis drives primitive endoderm formation by antagonizing naive and formative pluripotency. *Nat Commun* **14**, 1210 (2023).
43. Maskalenka, K. *et al.* NANOGP1, a tandem duplicate of NANOG, exhibits partial functional conservation in human naive pluripotent stem cells. *Development* **150** (2023).
44. Bi, Y. *et al.* Cell fate roadmap of human primed-to-naive transition reveals preimplantation cell lineage signatures. *Nat Commun* **13**, 3147 (2022).
45. Pastor, W.A. *et al.* TFAP2C regulates transcription in human naive pluripotency by opening enhancers. *Nature Cell Biology* **20**, 553-564 (2018).
46. Osnato, A. *et al.* Tgf β signalling is required to maintain pluripotency of human naive pluripotent stem cells. *eLife* **10** (2021).
47. Chambers, S.M. *et al.* Highly efficient neural conversion of human ES and iPS cells by dual inhibition of SMAD signaling. *Nat Biotechnol* **27**, 275-280 (2009).
48. Yoney, A. *et al.* WNT signaling memory is required for ACTIVIN to function as a morphogen in human gastruloids. *eLife* (2018).

Decision Letter, first revision:

Dear Magda,

As you know, I am writing on behalf of my colleague Dr Stylianos Lefkopoulos, who is out of the office.

First, I wished to apologize again for the long delay in communicating our decision to you.

Thank you for resubmitting your manuscript "Distinct pathways drive anterior hypoblast specification in the implanting human embryo", to the journal. The revision has now been seen by the original Reviewers #1, #2, #4, and their comments are pasted below. In light of their advice, we regret that we cannot offer to publish the study in Nature Cell Biology.

As you will see, although Reviewers #2 and #4 initially offered positive comments, Reviewer #1 continued to have persisting concerns with regard to the imaging analyses and the consistency between the imaging datasets and quantitative analyses. These are serious points that undermine central conclusions of the work, and we asked Reviewer #4 to provide comments on these points. They shared feedback with us and you (see below, "Additional comments from Rev#4 on Rev#1's comments") and agreed with Reviewer #1 that their technical concerns undermine the strength of the claims. Reviewer #4 also offered more positive feedback regarding the degree of advance.

At this stage, we are editorially concerned that the persisting comments from Rev#1, shared by Rev#4, about the quality of the data and how convincing they are, are significant. Regrettably, we consider these reservations sufficiently important to preclude publication of the study in Nature Cell Biology.

We are very sorry that we could not be more positive on this occasion, but we thank you for the opportunity to consider this work. Once again I am very sorry for sharing our decision so late.

With kind regards,
Melina Casadio

Melina Casadio, PhD
Senior Editor, Nature Cell Biology
ORCID ID: <https://orcid.org/0000-0003-2389-2243>

Reviewers' comments:

Reviewer #1 (Remarks to the Author):

In the revised manuscript, the authors have made efforts to address a number of criticisms raised by this reviewer. On the other hand, there remains some concerns that have not been addressed in a proper manner and accordingly, a lack of consistency in the results of immunofluorescence analyses

and the quantitative measurements by the authors. I consider this critical, since key conclusions of this work have been drawn from the image analyses. Some examples are as follows:

1. In Fig. 2a, some examples of the measurements of the pan-SMAD2/3 n/c ratio are shown, but there seems to be no correlation between the staining properties and the measurement values. Additionally, the staining of SMAD itself appears weak or nearly undetectable in some cells. In the ICM where the nuclei are large and the cells are tightly packed, it is very difficult to discern the cytoplasm. The values determined under such conditions may not be meaningful.

2. In Fig. 4a, the staining examples of pSMAD1/5/9 shown on the left appear mostly negative, but the measured values are 15.7 and 3.9, with around a four-fold difference. On the other hand, the middle panel shows a substantial difference in the staining properties, but the measured values are 58.6 and 34.9, with only a 1.68-fold difference; the staining difference in the right panel is also apparent, but the measured values are 44.7 and 27.4, with only a 1.63-fold difference. Thus, there is a significant discrepancy between the visual observations and the quantitative results.

3. In Fig. 5j-k and Response to Reviewers Figure 1.11, the immunofluorescence staining of pSMAD1/5/9 shows a clear difference between "primed" and "naïve", but the western blot shows nearly negative results for both conditions.

As pointed above, the conclusions drawn from ambiguous image-centered data lack consistency. Therefore, unfortunately, the concerns raised in the previous round of the reviewer remain unaddressed yet in the revised manuscript.

Furthermore, as the authors are aware from the toned-down title, this paper did not address the mechanisms of human AP axis determination or symmetry breaking, but rather examines the signaling pathway that can contribute to CER1-positive putative AVE specification. Therefore, the mechanism by which putative AVE appears on one side of the embryo and whether AVE truly contributes to the AP axis determination or symmetry breaking in humans remain unclear.

Given these remaining concerns, I find it difficult to conclude that this work holds significant advance over the author's previous report (Molè et al, 2021).

Reference

Molè MA, Coorens THH, Shahbazi MN, Weberling A, Weatherbee BAT, Gantner CW, Sancho-Serra C, Richardson L, Drinkwater A, Syed N et al (2021) A single cell characterisation of human embryogenesis identifies pluripotency transitions and putative anterior hypoblast centre. Nature Communications 12

Reviewer #2 (Remarks to the Author):

I have read through the rebuttal letter and revised manuscript. The authors have done additional experiments to address most of the concerns raised by reviewers. Therefore, the revised paper shall be acceptable for publication in Nature Cell Biology.

Reviewer #4 (Remarks to the Author):

The authors have satisfactorily addressed all of my concerns. Consequently, I have no additional comments and support this manuscript's publication.

Additional comments from Rev#4 about Rev#1's comments:

All the technical concerns of Reviewer 1 look reasonable to me. Regarding the conceptual advancement, the previous study (Mole et al.) is a single-cell RNA-seq data that characterize implantating human embryos and identify the putative anterior hypoblast center, without much functional study. I do find it important to investigate the roles of NODAL/BMP/NOTCH in epiblast/hypoblast specification in implanting human embryos in this work. However, these technical issues raised by Reviewer 1 do raise sufficient concerns on how reliable some of these conclusions are.

**Although we cannot publish your paper, it may be appropriate for another journal in the Nature Portfolio. If you wish to explore the journals and transfer your manuscript please use our manuscript transfer portal. You will not have to re-supply manuscript metadata and files, unless you wish to make modifications. For more information, please see our manuscript transfer FAQ page.

**For Nature Portfolio general information and news for authors, see <http://npg.nature.com/authors>.

Author Rebuttal, first revision:

We are very grateful to the reviewers for taking the time to consider our revised manuscript. In this response and revised manuscript, we seek to clarify methodology and bolster confidence in our quantification of relative SMAD2.3 and pSMAD1.5.9 immunofluorescence intensities as proxies for NODAL and BMP signaling, respectively. We hope it will be clear from the additional quantification examples on raw imaging intensities, quantification of SMAD levels in embryos treated with small molecule inhibitors, presentation of consistent intra-embryo trends, and methodological clarification that our quantifications are trustworthy and biologically meaningful.

Reviewer 1:

In the revised manuscript, the authors have made efforts to address a number of criticisms raised by this reviewer. On the other hand, there remains some concerns that have not been addressed in a proper manner and accordingly, a lack of consistency in the results of immunofluorescence analyses and the quantitative measurements by the authors. I consider this critical, since key conclusions of this work have been drawn from the image analyses. Some examples are as follows:

We are grateful to the reviewer for their additional time and efforts and appreciate the opportunity for clarification.

1. In Fig. 2a, some examples of the measurements of the pan-SMAD2/3 n/c ratio are shown, but there seems to be no correlation between the staining properties and the measurement values. Additionally, the staining of SMAD itself appears weak or nearly undetectable in some cells. In the ICM where the nuclei are large and the cells are tightly packed, it is very difficult to discern the cytoplasm. The values determined under such conditions may not be meaningful.

The reviewer is correct that Fig. 2a gives nuclear-to-cytoplasmic ratio of pan-SMAD2/3 ratios for a segregated blastocyst, focusing on Epiblast (EPI; left), Hypoblast (HYPO; middle), and Trophectoderm/Trophoblast (TE/TrB; right). We now present Fig. 2a with all quantified values (nuclear, cytoplasmic, and ratio; also included below as Response to Reviewers Figure 1.1a). We believe the images do visually correlate with these measurements, particularly when accounting for the two direct measurements (nuclear and cytoplasmic average intensity). We note that the images in the paper, including Fig. 2a, were processed with adjusted brightness and contrast, as well as projecting the average fluorescence of several z-planes (~5-10 planes; maximum projection function in FIJI) to allow readers to see the overall topology of many cells that otherwise are on disparate planes for the Figures. However, all quantification was performed on raw imaging data in FIJI, similar to many other publications (e.g. refs.¹⁻⁵). Specifically, we identified the middle plane of a nuclei, generated an average maximum projection ± 1 z-plane from that central (total 3 z-planes in each stack), and quantified the fluorescence on that z-stack (in essence, quantifying the average fluorescence of the central 3 planes).

Image processing was performed with the goal of images being representative of the embryo topology, though we acknowledge this complicates the generation of images representative of individual cells' measured values. To address this, we have provided new examples in Extended Data Fig. 4a-b with the raw SMAD2.3 signal, and all measured values (also include below as Response to Reviewers Figure 1.1b-c).

Response to Reviewers Figure 1.1: **a**, Examples from a segregated blastocyst of Epiblast (EPI), Hypoblast (HYPO) and Trophectoderm (TE) cells with nuclear (solid line) and cytoplasmic (dotted line) regions of interest (ROI) outlined. Measured nuclear (Avg. Nuc) and cytoplasmic (Avg. Cyto) average SMAD2.3 intensity values are listed together with calculated nuclear-to-cytoplasmic ratio (N/C). Note images here are processed while measurements are performed on raw data. **b-c**, Examples from a segregating blastocyst (**b**) and day 7 embryo (**c**) of cells with nuclear and cytoplasmic ROIs outlined. Measured nuclear, cytoplasmic, and calculated nuclear-to-cytoplasmic ratios of total SMAD2.3 intensities are shown for each cell. Note that here the raw, unprocessed SMAD2.3 signal is shown for the central 3-plane z-stacks used directly for quantification. Scale bars: 100µm.

The nuclear region of interest (ROI) was defined automatically in FIJI based on DAPI staining. Cytoplasmic ROIs were manually determined. In many cases, embryos were restained for a cell membrane marker (E-Cadherin; see Response to Reviewers Figure 1.1b). In these cases, the cell boundaries were clearly marked in the ICM, allowing us to decisively delineate the cytoplasm. In cases where the apical domain marker podocalyxin was used, several considerations were made when drawing cytoplasmic boundaries (see Response to Reviewers Figure 1.1c): (1) extension of cytoplasmic ROI to the apical area marked by podocalyxin if the cell neighbors the lumen; (2) ensuring no neighboring nuclei are included in the ROI; (3) following the groves of cell membrane sometimes delineated by the total SMAD2.3 staining (such as those seen at the lower boundary of Cell 5 and the lower and right boundaries of Cell 6 in the above Response Figure 1.1c); and (4) generally erring to be closer in to the nucleus when unsure. Together, these factors allow us to confidently delineate a cytoplasmic ROI that captures suitable intra-cellular variation allowing for valid normalization.

We have clarified the above in the methods section:

<< To quantify SMAD2.3 nuclear/cytoplasmic ratio in human and mouse embryos, the central 3 planes of individual cells were used to generate a 3-plane z-stack. Individual DAPI-positive nuclei were used to generate a nuclear mask using the 'Analyze Particles' function on either the DAPI or lineage-associated transcription factor channel. The adjacent cytoplasmic area was drawn individually for each nucleus and the mean fluorescence of each region was measured, and the ratio computed. When embryos were stained with E-Cadherin, the membrane was delineated to allow for cytoplasmic ROI determination. When embryos were stained with podocalyxin, the cytoplasmic ROI was drawn to ensure delineation of a region captures suitable intra-cellular variation allowing for valid normalization. Measurements were computed on raw SMAD2.3 signal. >>

<< To generate Figures, images were processed by generating z-stacks of ~5-10 planes to allow for visualization of embryo topology with cells on disparate planes followed by consistent adjustment of brightness and contrast. >>

The manuscript also includes the quantification of individual nuclear and cytoplasmic SMAD2.3 values plotted alongside the nuclear/cytoplasmic ratio for all embryos (Fig. 2c, d). These data show that nuclear and cytoplasmic measurements range from approximately 5 A.U. to <200 a.u. As the dynamic range is 0-255 a.u., we feel this data supports the fact that the imaging settings (including laser power and gain) are adequate to capture the range of fluorescence levels without clipping and allows for objective quantification. However, there is always some level of background, and this contributes to values close to but not zero in some cells and/or acellular regions. Importantly, these data have been used to quantify **relative** levels.

To bolster confidence in the meaningful nature of these quantifications, we have now added quantification of the nuclear-to-cytoplasmic ratio of control versus A83-01 treated (NODAL inhibition) Day 7 and 9 human embryos (Response to Reviewers Figure 1.2a). These data clearly show that our methods accurately capture the expected decrease in SMAD2.3 signaling upon treatment with the A83-01 inhibitor. We additionally show this signaling perturbation to be functionally relevant for anterior hypoblast specification later in the manuscript.

Last, to confirm that the global differences we observe between lineages is not due to technical variation between imaging batches or high background staining in select embryos, we now plot values from each major lineage from each individual embryo below (Extended Data Fig. 5a; included below as Response to Reviewers Figure 1.2b). This confirms that the global trends we report are consistent within individual embryos.

Together, these analyses and data bolster our confidence that our approach accurately determines meaningful values.

Response to Reviewers Figure 1.2: **a**, Quantification of the nuclear-to-cytoplasmic ratio of total SMAD2.3 in hypoblast (GATA6-positive) cells at day 7 and day 9 in Control (n=322 cells from 17 embryos at Day 7 and 214 cells from 12 embryos at Day 9) versus A83-01 treated embryos (n=67 cells from 2 embryos at Day 7 and 209 cells from 10 embryos at Day 9). **b**, Mean \pm S.E.M. nuclear-to-cytoplasmic ratio of total SMAD2.3 within each lineage of segregating and segregated blastocysts, day 7, and day 9 embryos, separated by individual embryo. Note consistency of trends, particularly of TE versus ICM enrichment in segregating blastocysts, and of hypoblast versus epiblast. Statistical tests: Unpaired T-test (a). ****p<0.0001.

2. In Fig. 4a, the staining examples of pSMAD1/5/9 shown on the left appear mostly negative, but the measured values are 15.7 and 3.9, with around a four-fold difference. On the other hand, the middle panel shows a substantial difference in the staining properties, but the measured values are 58.6 and 34.9, with only a 1.68-fold difference; the staining difference in the right panel is also apparent, but the measured values are 44.7 and 27.4, with only a 1.63-fold difference. Thus, there is a significant discrepancy between the visual observations and the quantitative results.

Fig. 4a shows examples from a segregated blastocyst of EPI (left), HYPO (middle), and TE/TrB (right) cells with nuclear region of interest (ROI) outlined.

Unfortunately, due to an error when copying values over from an Excel sheet to generate the Figures, Cell 2 in the epiblast was previously mislabeled as 3.9 a.u. We have now corrected this, and now provide all measured values (nuclear signal, background, and calculated normalized signal; also provided below as Response to Reviewers Figure 1.3a). We thank the Reviewer for noticing this mistake. We have checked our other Graphpad sheets and Excel files to ensure other errors were not propagated.

As noted in the above response, images presented in the Figures are processed and quantifications are performed on raw signals from the average maximum projection of the central 3 z-planes of a given cell. We now provide additional examples with raw pSMAD1.5.9 signal and all measured values in Extended Data Fig. 7a-b (also provided below as Response to Reviewers Figure 1.3b-c). Please note that normalization of pSMAD1.5.9 was performed to an acellular region within the same 3-plane z-stack. These areas were not visible in Fig. 4a but are noted in the new examples.

Response to Reviewers Figure 1.3: a, Examples from a segregated blastocyst of Epiblast (EPI), Hypoblast (HYPO) and Trophectoderm (TE) cells with nuclear (solid line) regions of interest (ROI) outlined. Measured nuclear (Avg. Nuc) and background average pSMAD1.5.9 intensity values are listed together with calculated normalized pSMAD1.5.9 intensity (Nuc/1+Background). Note images here are processed while measurements are performed on raw data. **b-c**, Examples from a segregating blastocyst (b) and day 7 embryo (c) of cells with nuclear and background ROIs outlined. Measured nuclear and background (bg) and calculated normalized pSMAD1.5.9 intensities are shown for each cell. Note that here the raw, unprocessed pSMAD1.5.9 signal is shown for the central 3-plane z-stacks used directly for quantification. Cells that are central to the same 3-plane z-stack use the same background value for normalization. Scale bars: 100µm.

The reviewer calculated the fold differences between the two nuclei in each lineage. We agree that this difference might not match the visual impression. However, we caution that the fold change is likely not informative. Instead, the absolute values are the values of interest when performing statistical analyses of this data. To provide clarity on visual correlations with quantitative signals, we provide some swatches of the in-built Lookup Tables (LUTs) used in the manuscript from FIJI below:

Response to Reviewers Figure 1.4: Swatches from FIRE (top) and Magenta (bottom) look up tables (LUTs) from ImageJ/FIJI. Higher visual discrimination between values in the FIRE LUT than single color LUTs, such as Magenta, making it useful for visualization of single channel SMAD staining throughout the manuscript. Note that visual differences correlate with absolute value differences rather than fold changes between intensity (a.u.) values.

Last, we have performed additional analyses similar to those presented in Response to Reviewers Figure 1.2 for the pSMAD1.5.9 staining. Specifically, this shows that LDN treatment meaningfully reduces our quantified pSMAD1.5.9 signal (Response to Reviewers Figure 1.5a), lending confidence to our approaches. Further, the trends we report between lineages in our quantifications are conserved within individual embryos (Extended Data Fig. 7c; included below as Response to Reviewers Figure 1.5b). Taken together, we are confident in these approaches.

Response to Reviewers Figure 1.5: **a**, Quantification of normalized pSMAD1.5.9 intensity in hypoblast (GATA6-positive) cells at day 7 and day 9 in Control (n=236 cells from 9 embryos at Day 7 and 87 cells from 4 embryos at Day 9) versus LDN treated embryos (n=121 cells from 7 embryos at Day 7 and 58 cells from 4 embryos at Day 9). **b**, Mean \pm S.E.M. normalized pSMAD1.5.9 intensity within each lineage of segregating and segregated blastocysts, day 7, and day 9 embryos, separated by individual embryo. Note consistency of trends, particularly of TE versus ICM enrichment in segregating blastocysts, and of hypoblast versus epiblast in segregated blastocysts and at day 7. Statistical tests: Unpaired T-test (a). **** $p < 0.0001$.

3. In Fig. 5j-k and Response to Reviewers Figure 1.11, the immunofluorescence staining of pSMAD1/5/9 shows a clear difference between “primed” and “naïve”, but the western blot shows nearly negative results for both conditions.

In the first round of review, the reviewer had mentioned “according to a previous report (Io et al., 2021), the SMAD1/5/8 of hPSCs in PXGL media are not phosphorylated”. To respond to this point, our previous response included the relevant panel from Figure S1K of Io et al., 2021⁶, which is a western blot detecting pSMAD1.5. We agree with the reviewer that the bands in the published blot are faint for both the primed and naïve human PSC H9 samples, detected with Phospho-

SMAD1/5 (Ser463/465) (41D10) Rabbit mAb #9516 Cell Signaling Technology. It is important to note here that this blot included samples with exogenous BMP, (“nTE_D1”), which would have determined the level of exposure during imaging of the blot, and therefore made any potential difference in endogenous levels (i.e. between naïve and primed) difficult to interpret.

To address the reviewer’s point regarding endogenous differences directly in our samples, we determined if SMAD1/5/9 are phosphorylated by performing immunofluorescence of the following: PXGL (naïve: 3057 cells) and mTeSR (primed: 5761 cells) Shef6 hESC, as well as mESCs in 2iLif conditions (naïve; 297 cells) or basal N2B27 (naïve exit; 789 cells), using Phospho-SMAD1 (Ser463/465)/ SMAD5 (Ser463/465)/ SMAD9 (Ser465/467) (D5B10) Rabbit mAb #13820 Cell Signaling Technology.

In Shef6 hESCs, we observed an unambiguous relative enrichment of pSMAD1.5.9 in the naïve state compared to the primed state, and the pattern was reversed in mouse (Fig. 5j-k). Taken together, these data suggest that BMP signaling is active in PXGL cultured, naïve human ESC.

It is difficult to compare the two studies, given the different approaches, cell lines, and antibodies used. However, both studies show that there is phosphorylation of SMAD1.5 in naïve hESCs. If the reviewer or editor feel that it is necessary, then we could also analyze SMAD1/5/9 phosphorylation using the D5B10 antibody in these two states, in Shef6 hESCs, by western blot.

As pointed above, the conclusions drawn from ambiguous image-centered data lack consistency. Therefore, unfortunately, the concerns raised in the previous round of the reviewer remain unaddressed yet in the revised manuscript.

We appreciate the reviewer’s concerns and hope that the additional information and data has attenuated their concerns.

Furthermore, as the authors are aware from the toned-down title, this paper did not address the mechanisms of human AP axis determination or symmetry breaking, but rather examines the signaling pathway that can contribute to CER1-positive putative AVE specification. Therefore, the mechanism by which putative AVE appears on one side of the embryo and whether AVE truly contributes to the AP axis determination or symmetry breaking in humans remain unclear.

We agree with the reviewer and acknowledge that our manuscript does not shed light on how the anterior hypoblast becomes biased to the putative anterior. We believe that the current title, which includes specific mention of ‘hypoblast specification’, accurately captures the manuscript’s major findings.

Given these remaining concerns, I find it difficult to conclude that this work holds significant advance over the author’s previous report (Molè et al, 2021).

Reference

Molè MA, Coorens THH, Shahbazi MN, Weberling A, Weatherbee BAT, Gantner CW, Sancho-Serra C, Richardson L, Drinkwater A, Syed N et al (2021) A single cell characterisation of human embryogenesis identifies pluripotency transitions and putative anterior hypoblast centre. Nature Communications 12

Our previous work described the existence of an anterior signaling center in human embryos. However, this study did not investigate the function of signaling pathways in anterior hypoblast specification. Given the importance of this timepoint of development and the fundamental lack of

data within the context of the human embryo, we believe this manuscript represents a clear advance, which will be of wide interest to the field.

Reviewer #2 (Remarks to the Author):

I have read through the rebuttal letter and revised manuscript. The authors have done additional experiments to address most of the concerns raised by reviewers. Therefore, the revised paper shall be acceptable for publication in Nature Cell Biology.

We thank the reviewer for their encouraging response.

Reviewer #4 (Remarks to the Author):

The authors have satisfactorily addressed all of my concerns. Consequently, I have no additional comments and support this manuscript's publication.

We thank the reviewer for their positive feedback.

Additional comments from Rev#4 about Rev#1's comments:

All the technical concerns of Reviewer 1 look reasonable to me. Regarding the conceptual advancement, the previous study (Mole et al.) is a single-cell RNA-seq data that characterize implanting human embryos and identify the putative anterior hypoblast center, without much functional study. I do find it important to investigate the roles of NODAL/BMP/NOTCH in epiblast/hypoblast specification in implanting human embryos in this work. However, these technical issues raised by Reviewer 1 do raise sufficient concerns on how reliable some of these conclusions are.

We also thank the reviewer for providing feedback on the points raised by R1. We have included new data and clarification to address these points and hope that they alleviate the concerns. We thank the reviewer very much for elaborating on the conceptual advance provided by our current manuscript.

References

1. Chapnick, D.A. & Liu, X. Analysis of ligand-dependent nuclear accumulation of Smads in TGF-beta signaling. *Methods Mol Biol* **647**, 95-111 (2010).
2. Deng, H. et al. Activation of Smad2/3 signaling by low fluid shear stress mediates artery inward remodeling. *Proc Natl Acad Sci U S A* **118** (2021).
3. Frick, C.L., Yarka, C., Nunns, H. & Goentoro, L. Sensing relative signal in the Tgf-beta/Smad pathway. *Proc Natl Acad Sci U S A* **114**, E2975-E2982 (2017).
4. Heemskerk, I. et al. Rapid changes in morphogen concentration control self-organized patterning in human embryonic stem cells. *eLife* (2019).
5. Pedroza, M. et al. Self-patterning of human stem cells into post-implantation lineages. *Nature* (2023).
6. Io, S. et al. Capturing human trophoblast development with naive pluripotent stem cells in vitro. *Cell Stem Cell* **28**, 1023-1039.e1013 (2021).

Decision Letter, second revision:

Dear Magda,

Thank you for submitting your revised manuscript "Distinct pathways drive anterior hypoblast specification in the implanting human embryo" (NCB-A50170B-Z) and we apologize again for the delay in getting back to you.

We sent the manuscript back to the original referee #1, as they were the only referee with remaining concerns. Reviewer #1 found that your manuscript was improved upon revision, but continued to raise some possible issues with the imaging approaches. We therefore approached an additional (new) referee (reviewer #5) who is an expert in microscopy/imaging analysis and early development and who was asked to comment specifically on the imaging analysis issues raised by reviewer #1. The comments by both referees are pasted below. Based on the comments by both reviewers, we have decided that we'll be happy in principle to publish your manuscript in Nature Cell Biology, pending minor revisions to satisfy the referees' final requests and to comply with our editorial and formatting guidelines.

Specifically, we do not expect you to address the remaining points about the imaging analysis raised by reviewer #1 (given reviewer #5's feedback), with the exception of one point: please make sure to explain in your text why you have applied the normalization rather than the subtraction of the noise signal: "Regarding the quantification of pSMAD 1/5/9, the authors applied normalization with [Nuc/1+Background], but I have never seen this kind of correction for treatment of noise on images. I think the formula in this case is [observed signal = truth signal + background noise signal], so the calculation should be [observed value - background noise value]. Could the authors explain why they apply the normalization rather than the subtraction of the noise signal?"

Please do not perform any revision now. We are now performing detailed checks on your paper and will send you a checklist detailing our editorial and formatting requirements in about a week (so you can address this point by reviewer #1 after we send you our checklist). Please do not upload the final materials and make any revisions until you receive this additional information from us.

IMPORTANT: If the current version of your manuscript is in a PDF format, please email us a copy of the file in an editable format (Microsoft Word or LaTeX)-- we can not proceed with PDFs at this stage.

Thank you again for your interest in Nature Cell Biology. Please do not hesitate to contact me if you have any questions.

Best regards,
Stelios

Stylios Lefkopoulos, PhD
He/him/his

Senior Editor, Nature Cell Biology
Springer Nature
Heidelberger Platz 3, 14197 Berlin, Germany

E-mail: stylianos.lefkopoulos@springernature.com
Twitter: @s_lefkopoulos
LinkedIn: [linkedin.com/in/stylianos-lefkopoulos-81b007a0](https://www.linkedin.com/in/stylianos-lefkopoulos-81b007a0)

Reviewer #1 (Remarks to the Author):

In the re-revised manuscript, the authors addressed my concerns regarding image processing and representation. They clarified the differences in the methods used for capturing images for representation versus those for quantification.

Based on the detailed quantification information provided by the authors, I took a second look at the images. While some appear to align visually with the measurements, I found that others do not.

For example, in my view, in Fig 2a:

For Cell #1, visually, I don't perceive a difference in intensity between the nucleus and the cytoplasm; however, the measurements indicate $nuc=77.67$ and $cyto=55.24$, suggesting that the nucleus is more intense.

For Cell #3, it appears that the cytoplasm has larger areas of stronger intensity than the nucleus, yet the measured values are $Nuc=65.55$ and $Cyto=67.26$, showing almost no difference.

For Cell #7, the nucleus seems to have areas of stronger intensity compared to the cytoplasm, but the values are $Nuc=56.38$ and $Cyto=53.52$, again indicating almost no difference.

The authors claimed that they used different regions for quantification and visual representation in order to display the overall cellular topology. However, I believe that the primary aim of Figures 2a and 4a should be to demonstrate a clear correlation between the visual impression and the numerical measurements. As such, it would be preferable to use the same images for both purposes. Moreover, while multiple EPI cells were depicted, only a single cell was shown for both Hypo and TE, which diminishes the need to represent their overall topology.

Additionally, given the heterogeneous staining patterns observed, I questioned whether examining only three planes would provide sufficient data for reliable quantification. While I recognize that 3D analysis may be challenging, wouldn't it be necessary for more accurate data?

Given the considerations outlined above, I am not sure whether the authors' approach to image processing and their accompanying explanations for this section are appropriate.

As the Reviewer #4 also claimed, I agree that the function of NODAL, BMP, and NOTCH pathway on AVE specification is important. However, the significance of NOTCH remains ambiguous, and other important data are not sufficiently clear as noted above. Therefore, I can't say for sure whether this manuscript is acceptable or appropriate for Nature Cell Biology.

Minor comment;

Regarding the quantification of pSMAD 1/5/9, the authors applied normalization with [Nuc/1+Background], but I have never seen this kind of correction for treatment of noise on images. I think the formula in this case is [observed signal = truth signal + background noise signal], so the calculation should be [observed value - background noise value]. Could the authors explain why they apply the normalization rather than the subtraction of the noise signal?

Reviewer #5 (Remarks to the Author):

I have been asked to comment on the remaining points by reviewer #1 in the last round of review (round #3):

I conclude that the images and quantifications presented are consistent. While I have not examined the paper in full, I do not find a discrepancy for the image analysis section highlighted by reviewer #1.

It is correct that quantifying fluorescence intensities within the cytoplasm of these cells is technically challenging. But by visual inspection, the images presented match the quantified results for both nuclear and cytoplasmic intensities. This is the case for the main Figure 2a, where they show a projected z stack throughout part of the cell, and for the Extended Figure 4, where they show unprocessed raw images. In general, the visual impression matches the quantified results.

It is true that in some Z sections the signal can vary between sub-cellular regions. But I do not see a reason to suspect that the quantification of the raw data (comprising several confocal scans per cell) has not been performed correctly.

Decision Letter, final checks:

Our ref: NCB-A50170B-Z

23rd November 2023

Dear Dr. Zernicka-Goetz,

Thank you for your patience as we've prepared the guidelines for final submission of your Nature Cell Biology manuscript, "Distinct pathways drive anterior hypoblast specification in the implanting human embryo" (NCB-A50170B-Z). Please carefully follow the step-by-step instructions provided in the attached file, and add a response in each row of the table to indicate the changes that you have made. Please also check and comment on any additional marked-up edits we have proposed within the text. Ensuring that each point is addressed will help to ensure that your revised manuscript can be swiftly handed over to our production team.

We would like to start working on your revised paper, with all of the requested files and forms, as

soon as possible (preferably within two weeks). Please get in contact with us if you anticipate delays.

In recognition of the time and expertise our reviewers provide to Nature Cell Biology's editorial process, we would like to formally acknowledge their contribution to the external peer review of your manuscript entitled "Distinct pathways drive anterior hypoblast specification in the implanting human embryo". For those reviewers who give their assent, we will be publishing their names alongside the published article.

Nature Cell Biology offers a Transparent Peer Review option for new original research manuscripts submitted after December 1st, 2019. As part of this initiative, we encourage our authors to support increased transparency into the peer review process by agreeing to have the reviewer comments, author rebuttal letters, and editorial decision letters published as a Supplementary item. When you submit your final files please clearly state in your cover letter whether or not you would like to participate in this initiative. Please note that failure to state your preference will result in delays in accepting your manuscript for publication.

Cover suggestions

COVER ARTWORK: We welcome submissions of artwork for consideration for our cover. For more information, please see our guide for cover artwork.

Nature Cell Biology has now transitioned to a unified Rights Collection system which will allow our Author Services team to quickly and easily collect the rights and permissions required to publish your work. Approximately 10 days after your paper is formally accepted, you will receive an email in providing you with a link to complete the grant of rights. If your paper is eligible for Open Access, our Author Services team will also be in touch regarding any additional information that may be required to arrange payment for your article.

Please note that *Nature Cell Biology* is a Transformative Journal (TJ). Authors may publish their research with us through the traditional subscription access route or make their paper immediately open access through payment of an article-processing charge (APC). Authors will not be required to make a final decision about access to their article until it has been accepted. Find out more about Transformative Journals

Authors may need to take specific actions to achieve compliance with funder and institutional open access mandates. If your research is supported by a funder that requires immediate open access (e.g. according to Plan S principles) then you should select the gold OA route, and we will direct you to the compliant route where possible. For authors selecting the subscription publication route, the journal's standard licensing terms will need to be accepted, including self-

archiving policies. Those licensing terms will supersede any other terms that the author or any third party may assert apply to any version of the manuscript.

Please use the following link for uploading these materials:
[Redacted]

Best regards,

Caroline Li
Staff
Nature Cell Biology

On behalf of

Stylios Lefkopoulos, PhD
He/him/his
Senior Editor, Nature Cell Biology
Springer Nature
Heidelberger Platz 3, 14197 Berlin, Germany

E-mail: stylios.lefkopoulos@springernature.com
Twitter: [@s_lefkopoulos](https://twitter.com/s_lefkopoulos)
LinkedIn: [linkedin.com/in/stylios-lefkopoulos-81b007a0](https://www.linkedin.com/in/stylios-lefkopoulos-81b007a0)

Reviewer #1:

Remarks to the Author:

In the re-revised manuscript, the authors addressed my concerns regarding image processing and representation. They clarified the differences in the methods used for capturing images for representation versus those for quantification.

Based on the detailed quantification information provided by the authors, I took a second look at the images. While some appear to align visually with the measurements, I found that others do not.

For example, in my view, in Fig 2a:

For Cell #1, visually, I don't perceive a difference in intensity between the nucleus and the cytoplasm; however, the measurements indicate nuc=77.67 and cyto=55.24, suggesting that the nucleus is more intense.

For Cell #3, it appears that the cytoplasm has larger areas of stronger intensity than the nucleus, yet the measured values are Nuc=65.55 and Cyto=67.26, showing almost no difference.

For Cell #7, the nucleus seems to have areas of stronger intensity compared to the cytoplasm, but the values are Nuc=56.38 and Cyto=53.52, again indicating almost no difference.

The authors claimed that they used different regions for quantification and visual representation in order to display the overall cellular topology. However, I believe that the primary aim of Figures 2a and 4a should be to demonstrate a clear correlation between the visual impression and the numerical measurements. As such, it would be preferable to use the same images for both purposes. Moreover, while multiple EPI cells were depicted, only a single cell was shown for both Hypo and TE, which diminishes the need to represent their overall topology.

Additionally, given the heterogeneous staining patterns observed, I questioned whether examining only three planes would provide sufficient data for reliable quantification. While I recognize that 3D analysis may be challenging, wouldn't it be necessary for more accurate data?

Given the considerations outlined above, I am not sure whether the authors' approach to image processing and their accompanying explanations for this section are appropriate.

As the Reviewer #4 also claimed, I agree that the function of NODAL, BMP, and NOTCH pathway on AVE specification is important. However, the significance of NOTCH remains ambiguous, and other important data are not sufficiently clear as noted above. Therefore, I can't say for sure whether this manuscript is acceptable or appropriate for Nature Cell Biology.

Minor comment;

Regarding the quantification of pSMAD 1/5/9, the authors applied normalization with $[\text{Nuc}/1 + \text{Background}]$, but I have never seen this kind of correction for treatment of noise on images. I think the formula in this case is $[\text{observed signal} = \text{truth signal} + \text{background noise signal}]$, so the calculation should be $[\text{observed value} - \text{background noise value}]$. Could the authors explain why they apply the normalization rather than the subtraction of the noise signal?

Reviewer #5:

Remarks to the Author:

I have been asked to comment on the remaining points by reviewer #1 in the last round of review (round #3):

I conclude that the images and quantifications presented are consistent. While I have not examined the paper in full, I do not find a discrepancy for the image analysis section highlighted by reviewer #1.

It is correct that quantifying fluorescence intensities within the cytoplasm of these cells is technically

challenging. But by visual inspection, the images presented match the quantified results for both nuclear and cytoplasmic intensities. This is the case for the main Figure 2a, where they show a projected z stack throughout part of the cell, and for the Extended Figure 4, where they show unprocessed raw images. In general, the visual impression matches the quantified results.

It is true that in some Z sections the signal can vary between sub-cellular regions. But I do not see a reason to suspect that the quantification of the raw data (comprising several confocal scans per cell) has not been performed correctly.

Author Rebuttal, Second Revision:

Reviewer #1:

Remarks to the Author:

In the re-revised manuscript, the authors addressed my concerns regarding image processing and representation. They clarified the differences in the methods used for capturing images for representation versus those for quantification.

We thank the reviewer for looking at our revised manuscript.

Based on the detailed quantification information provided by the authors, I took a second look at the images. While some appear to align visually with the measurements, I found that others do not.

For example, in my view, in Fig 2a:

For Cell #1, visually, I don't perceive a difference in intensity between the nucleus and the cytoplasm; however, the measurements indicate $nuc=77.67$ and $cyto=55.24$, suggesting that the nucleus is more intense.

For Cell #3, it appears that the cytoplasm has larger areas of stronger intensity than the nucleus, yet the measured values are $Nuc=65.55$ and $Cyto=67.26$, showing almost no difference.

For Cell #7, the nucleus seems to have areas of stronger intensity compared to the cytoplasm, but the values are $Nuc=56.38$ and $Cyto=53.52$, again indicating almost no difference.

We would once again reiterate that Fig. 2a included processed, not raw, images. Therefore, it is once again unsurprising that a subjective view of these images may yield differing impressions. Extended Data Fig. 4a-b provided in the previous round of revision instead showed unprocessed

images more representative on the intensities used for quantifications. We would also note, however, that as Reviewer 5 notes below, we do not agree that there are discrepancies between the quantifications and visual impression of the data.

The authors claimed that they used different regions for quantification and visual representation in order to display the overall cellular topology. However, I believe that the primary aim of Figures 2a and 4a should be to demonstrate a clear correlation between the visual impression and the numerical measurements. As such, it would be preferable to use the same images for both purposes. Moreover, while multiple EPI cells were depicted, only a single cell was shown for both Hypo and TE, which diminishes the need to represent their overall topology.

The previously provided Fig. 2a was a high magnification of the image of the segregated blastocyst presented in Fig. 2. Therefore, its presentation was influenced by the larger role of the image of the segregated blastocyst used in previous Fig. 2b – for which we indeed prioritized viewing embryo topology. We agree with the reviewer that this choice was less helpful than the way we have presented the data in Extended Data Fig. 4a where the whole, unprocessed imaging of the embryo is viewable with the individual cells mapped back to the whole embryo, in context.

Additionally, given the heterogeneous staining patterns observed, I questioned whether examining only three planes would provide sufficient data for reliable quantification. While I recognize that 3D analysis may be challenging, wouldn't it be necessary for more accurate data?

We are confident that our quantifications are biologically meaningful without the need for pursuing more complex imaging analysis techniques – we would again point the reviewer to the data provided in our previous rebuttal showing that the SMAD quantifications decrease significantly with the addition of signaling antagonists, which validates our approach.

Given the considerations outlined above, I am not sure whether the authors' approach to image processing and their accompanying explanations for this section are appropriate.

As the Reviewer #4 also claimed, I agree that the function of NODAL, BMP, and NOTCH pathway on AVE specification is important. However, the significance of NOTCH remains ambiguous, and other important data are not sufficiently clear as noted above. Therefore, I can't say for sure whether this manuscript is acceptable or appropriate for Nature Cell Biology.

Minor comment;

Regarding the quantification of pSMAD 1/5/9, the authors applied normalization with $[\text{Nuc}/1+\text{Background}]$, but I have never seen this kind of correction for treatment of noise on images. I think the formula in this case is $[\text{observed signal} = \text{truth signal} + \text{background noise}]$

signal], so the calculation should be [observed value - background noise value]. Could the authors explain why they apply the normalization rather than the subtraction of the noise signal?

It was particularly important for us to normalize using the background signal in this case because (1) laser penetration changes with depth and (2) we stained embryos from multiple experiments (e.g. with different iterations of making up primary and secondary antibodies). Therefore, the observed signal itself may change due to these differences within embryo and across experiments rather than based on biologically relevant differences in signaling. By dividing by background (with a pseudo count of 1 to avoid ever dividing by 0), we get a normalized pSMAD1.5.9 signal which is essentially a signal-to-noise ratio that will be more meaningful for comparisons. This is not an uncommon practice.

Reviewer #5:

Remarks to the Author:

I have been asked to comment on the remaining points by reviewer #1 in the last round of review (round #3):

I conclude that the images and quantifications presented are consistent. While I have not examined the paper in full, I do not find a discrepancy for the image analysis section highlighted by reviewer #1.

It is correct that quantifying fluorescence intensities within the cytoplasm of these cells is technically challenging. But by visual inspection, the images presented match the quantified results for both nuclear and cytoplasmic intensities. This is the case for the main Figure 2a, where they show a projected z stack throughout part of the cell, and for the Extended Figure 4, where they show unprocessed raw images. In general, the visual impression matches the quantified results.

It is true that in some Z sections the signal can vary between sub-cellular regions. But I do not see a reason to suspect that the quantification of the raw data (comprising several confocal scans per cell) has not been performed correctly.

We are extremely grateful to the reviewer for looking at our manuscript and for communicating the lack of discrepancy in our imaging analysis and approach.

Final Decision Letter:

Dear Magda,

I am pleased to inform you that your manuscript, "Distinct pathways drive anterior hypoblast specification in the implanting human embryo", has now been accepted for publication in Nature Cell Biology. Congratulations to you and the rest of the authors!

Please note that *Nature Cell Biology* is a Transformative Journal (TJ). Authors may publish their research with us through the traditional subscription access route or make their paper immediately open access through payment of an article-processing charge (APC). Authors will not be required to make a final decision about access to their article until it has been accepted. Find out more about Transformative Journals

If you have not already done so, we strongly recommend that you upload the step-by-step protocols used in this manuscript to the Protocol Exchange (www.nature.com/protocolexchange), an open online resource established by Nature Protocols that allows researchers to share their detailed experimental know-how. All uploaded protocols are made freely available, assigned DOIs for ease of citation and are fully searchable through nature.com. Protocols and Nature Portfolio journal papers in which they are used can be linked to one another, and this link is clearly and prominently visible in the online versions of both papers. Authors who performed the specific experiments can act as primary authors for the Protocol as they will be best placed to share the methodology details, but the Corresponding Author of the present research paper should be included as one of the authors. By uploading your Protocols to Protocol Exchange, you are enabling researchers to more readily reproduce or adapt the methodology you use, as well as increasing the visibility of your protocols and papers. You can also establish a dedicated page to collect your lab Protocols. Further information can be found at www.nature.com/protocolexchange/about

With kind regards,
Stelios

Stylios Lefkopoulos, PhD
He/him/his
Senior Editor, Nature Cell Biology
Springer Nature
Heidelberger Platz 3, 14197 Berlin, Germany

E-mail: stylios.lefkopoulos@springernature.com
Twitter: [@s_lefkopoulos](https://twitter.com/s_lefkopoulos)
LinkedIn: [linkedin.com/in/stylios-lefkopoulos-81b007a0](https://www.linkedin.com/in/stylios-lefkopoulos-81b007a0)